



# Global Nitrous Oxide Budget 1980-2020

Hanqin Tian[1*], Naiqing Pan[1,2], Rona L. Thompson[3], Josep G. Canadell[4], Parvadha Suntharalingam[5], Pierre Regnier[6], Eric A. Davidson[7], Michael Prather[8], Philippe Ciais[9], Marilena Muntean[10], Shufen Pan[11], Wilfried Winiwarter[12, 13], Sönke Zaehle[14], Feng Zhou[15], Robert B. Jackson[16], Hermann W. Bange[17], Sarah Berthet[18], Zihao Bian[19], Daniele Bianchi[20], Alexander F. Bouwman[21], Erik T. Buitenhuis[5], Geoffrey Dutton[22], Minpeng Hu[23], Akihiko Ito[24, 25], Atul K. Jain[26], Aurich Jeltsch-Thömmes[27], Fortunat Joos[27], Sian Kou-Giesbrecht[28, 29], Paul B. Krummel[30], Xin Lan[22,31], Angela Landolfi[32,17], Ronny Lauerwald[33], Ya Li[34], Chaoqun Lu[35], Taylor Maavara[36], Manfredi Manizza[37], Dylan B. Millet[38], Jens Mühle[37], Prabir K. Patra[39, 40, 41], Glen P. Peters[42], Xiaoyu Qin[34], Peter Raymond[43], Laure Resplandy[44], Judith A. Rosentreter[45,46], Hao Shi[34], Qing Sun[27], Daniele Tonina[47], Francesco N. Tubiello[48], Guido R. van der Werf[49], Nicolas Vuichard[9], Junjie Wang[21], Kelley C. Wells[38], Luke M. Western[22,50], Chris Wilson[51,52], Jia Yang[53], Yuanzhi Yao[54], Yongfa You[1,2], Qing Zhu[55]

[1]Schiller Institute for Integrated Science and Society, Department of Earth and Environmental Sciences, Boston College, Chestnut Hill, MA 02467, USA
[2]College of Forestry, Wildlife and Environment, Auburn University, Auburn, AL 36849, USA
[3]Norsk Institutt for Luftforskning, NILU, 2007 Kjeller, Norway
[4]Global Carbon Project, CSIRO Environment, Canberra, ACT 2101, Australia
[5]School of Environmental Sciences, University of East Anglia, Research Park, Norwich, NR4 7TJ, UK
[6]Department Geoscience, Environment & Society-BGEOSYS, Université Libre de Bruxelles, Brussels, Belgium
[7]Appalachian Laboratory, University of Maryland Center for Environmental Science, Frostburg, MD 21532, USA
[8]Department of Earth System Science, University of California, Irvine, CA 92697, USA
[9]Laboratoire des Sciences du Climat et de l'Environnement, LSCE, CEA CNRS, UVSQ UPSACLAY, 91198 Gif sur Yvette, France
[10]European Commission, Joint Research Centre (JRC), 21027 Ispra, Varese, Italy
[11]Department of Engineering and Environmental Studies Program, Boston College, Chestnut Hill, MA 02467, USA
[12]International Institute for Applied Systems Analysis, A-2361 Laxenburg, Austria
[13]Institute of Environmental Engineering, University of Zielona Góra, 65-417 Zielona Góra, Poland
[14]Max Planck Institute for Biogeochemistry, 07701 Jena, Germany
[15]Institute for Carbon Neutrality, Laboratory for Earth Surface Processes, Peking University, Beijing 100871, China
[16]Department of Earth System Science, Woods Institute for the Environment, and Precourt Institute for Energy, Stanford University, Stanford, CA 94305, USA
[17]GEOMAR Helmholtz Centre for Ocean Research Kiel, Wischhofstr. 1-3, 24148 Kiel, Germany
[18]Centre National de Recherches Météorologiques (CNRM), Université de Toulouse, Météo-France, CNRS, Toulouse, France
[19]School of Geography Science, Nanjing Normal University, Nanjing 210023, China
[20]Department of Atmospheric and Oceanic Sciences, University of California, Los Angeles, CA 90095, USA
[21]Department of Earth Sciences, Utrecht University, Princetonlaan 8a, Utrecht, The Netherlands, 3584CB
[22]Global Monitoring Laboratory, National Oceanic and Atmospheric Administration, Boulder, CO 80305, USA
[23]Department of Natural Resources and Environmental Science, University of Illinois Urbana-Champaign, Urbana, IL 61801, USA



[24]Graduate School of Life and Agricultural Sciences, University of Tokyo, Tokyo 113-8657, Japan
[25]Earth System Division, National Institute for Environmental Studies, Tsukuba 305-8506, Japan
[26]Department of Atmospheric Sciences, University of Illinois, Urbana-Champaign, Urbana, IL61801, USA
[27]Climate and Environmental Physics, Physics Institute and Oeschger Centre for Climate Change Research, University of Bern, 3012 Bern, Switzerland
[28]Canadian Centre for Climate Modelling and Analysis, Environment and Climate Change Canada, Victoria, BC, Canada V8W 324.
[29]Department of Earth and Environmental Sciences, Dalhousie University, Halifax, NS, Canada B3H 4R2.
[30]CSIRO Environment, Aspendale, Victoria 3195, Australia
[31] Cooperative Institute for Research in Environmental Sciences, University of Colorado Boulder, CO 80309 USA
[32]Institute of Marine Sciences, National Research Council (ISMAR-CNR), Via Fosso del Cavaliere 100, 00133, Rome, Italy;
[33]Université Paris-Saclay, INRAE, AgroParisTech, UMR ECOSYS, 91120 Palaiseau, France
[34]Research Center for Eco-Environmental Sciences, Chinese Academy of Sciences, Beijing 100085, China
[35]Department of Ecology, Evolution, and Organismal Biology, Iowa State University, Ames, IA 50011, USA
[36]School of Geography, University of Leeds, Leeds, UK
[37]Scripps Institution of Oceanography, University of California, San Diego, La Jolla, CA 92093, USA.
[38]Department of Soil, Water, and Climate, University of Minnesota, Saint Paul, MN 55108, USA
[39]Research Institute for Global Change, JAMSTEC, Yokohama 236 0001, Japan
[40]Research Institute for Humanity and Nature, Kyoto 603 8047, Japan
[41]Center for Environmental Remote Sensing, Chiba University, Chiba 263-8522, Japan
[42]CICERO Center for International Climate Research, Oslo 0349, Norway
[43]School of the Environment, Yale University, New Haven, CT 06511, USA
[44]Department of Geosciences, Princeton University, Princeton, NJ 08544, USA
[45]Centre for Coastal Biogeochemistry, Faculty of Science and Engineering, Southern Cross University, Lismore 2480, NSW, Australia
[46]Yale Institute for Biospheric Studies, Yale University, New Haven, CT 06520, USA
[47]Center for ecohydraulics Research, University of Idaho, Boise, Idaho 83702, USA
[48]Statistics Division, Food and Agriculture Organization of the United Nations, Via Terme di Caracalla, Rome 00153, Italy
[49]Meteorology and Air Quality Group, Wageningen University and Research Centre, 6708 PB Wageningen, the Netherlands
[50]School of Chemistry, University of Bristol, Bristol, BS8 1TS, UK
[51]School of Earth and Environment, University of Leeds, Leeds, LS2 9JT, UK
[52]National Centre for Earth Observation, University of Leeds, Leeds, LS2 9JT, UK
[53]Department of Natural Resource Ecology and Management, Oklahoma State University, Stillwater, OK 74078, USA
[54]School of Geographic Sciences, East China Normal University, Shanghai 200241, China
[55]Climate and Ecosystem Sciences Division, Lawrence Berkeley National Laboratory, 1 Cyclotron Road, Berkeley, CA 94720, USA

*Correspondence to: Hanqin Tian, (hanqin.tian@bc.edu)





**Abstract:** Nitrous oxide ($N_2O$) is a long-lived potent greenhouse gas and stratospheric ozone-depleting substance, which has been accumulating in the atmosphere since the pre-industrial period. The mole fraction of atmospheric $N_2O$ has increased by

nearly 25% from 270 parts per billion (ppb) in 1750 to 336 ppb in 2022, with the fastest annual growth rate since 1980 of more than 1.3 ppb $yr^{-1}$ in both 2020 and 2021. As a core component of our global greenhouse gas assessments coordinated by the Global Carbon Project (GCP), we present a global $N_2O$ budget that incorporates both natural and anthropogenic sources and sinks, and accounts for the interactions between nitrogen additions and the biochemical processes that control $N_2O$ emissions. We use Bottom-Up (BU: inventory, statistical extrapolation of flux measurements, process-based land and ocean modelling)

and Top-Down (TD: atmospheric measurement-based inversion) approaches. We provide a comprehensive quantification of global $N_2O$ sources and sinks in 21 natural and anthropogenic categories in 18 regions between 1980 and 2020. We estimate that total annual anthropogenic $N_2O$ emissions increased 40% (or 1.9 Tg N $yr^{-1}$) in the past four decades (1980-2020). Direct agricultural emissions in 2020, 3.9 Tg N $yr^{-1}$ (best estimate) represent the large majority of anthropogenic emissions, followed by other direct anthropogenic sources (including 'Fossil fuel and industry', 'Waste and wastewater', and 'Biomass burning'

(2.1 Tg N $yr^{-1}$), and indirect anthropogenic sources (1.3 Tg N $yr^{-1}$). For the year 2020, our best estimate of total BU emissions for natural and anthropogenic sources was 18.3 (lower-upper bounds: 10.5–27.0) Tg N $yr^{-1}$, close to our TD estimate of 17.0 (16.6–17.4) Tg N $yr^{-1}$. For the period 2010-2019, the annual BU decadal-average emissions for natural plus anthropogenic sources were 18.1 (10.4–25.9) Tg N $yr^{-1}$ and TD emissions were 17.4 (15.8–19.20 Tg N $yr^{-1}$. The once top emitter Europe has reduced its emissions since the 1980s by 31% while those of emerging economies have grown, making China the top emitter

since the 2010s. The observed atmospheric $N_2O$ concentrations in recent years have exceeded projected levels under all scenarios in the Coupled Model Intercomparison Project Phase 6 (CMIP6), underscoring the urgency to reduce anthropogenic $N_2O$ emissions. To evaluate mitigation efforts and contribute to the Global Stocktake of the United Nations Framework Convention on Climate Change, we propose establishing a global network for monitoring and modeling $N_2O$ from the surface through the stratosphere. The data presented in this work can be downloaded from https://doi.org/10.18160/RQ8P-2Z4R (Tian

et al. 2023).



**Executive summary**

The global $N_2O$ budget has been perturbed through direct and indirect anthropogenic emissions, but also through perturbations to the natural $N_2O$ sources and sinks through climate change, increasing atmospheric CO2 and land cover change. The tropospheric $N_2O$ mole fractions, precisely measured at a global network of stations, increased by more than 10% over the past

four decades, rising from 301 parts per billion (ppb) in 1980 to 333 ppb in 2020 and 336 ppb in 2022. It is higher than at any time in the last 800,000 years. The current growth rate of atmospheric $N_2O$ is unprecedented with respect to the ice core record covering the last deglacial transition (with decadal to centennial resolution) and likely unprecedented relative to the ice core records of the past 800,000 years with a substantially lower temporal resolution. The mean annual tropospheric growth rate increased from 0.76 (0.55-0.95) ppb $yr^{-1}$ in the decade of 2000-2009 to 0.96 (0.79-1.15) ppb $yr^{-1}$ in the decade of 2010-2019.

In 2020, the $N_2O$ tropospheric growth rate was 1.33 ppb $yr^{-1}$ (1.38 ppb $yr^{-1}$ in 2021), the highest observed rate since 1980 and over 30% higher than the average in the 2010s.

Global $N_2O$ emissions have significantly increased in the last four decades. The magnitudes of global $N_2O$ emissions estimated by the BU and TD approaches were comparable during the overlapping period 1997–2020, but TD estimates found a larger inter-annual variability and a faster rate of increase. BU approaches showed that global $N_2O$ emissions increased from 17.2 Tg

N $yr^{-1}$ (10.2-24.1 Tg N $yr^{-1}$) in 1997 to 18.3 Tg N $yr^{-1}$ (10.5-27.0 Tg N $yr^{-1}$) in 2020, with an average increase rate of 0.043 Tg N $yr^{-2}$ ($p < 0.05$). In contrast, according to TD estimates, global emissions increased from 15.4 Tg N $yr^{-1}$ (13.9-16.7 Tg N $yr^{-1}$) in 1997 to 17.0 Tg N $yr^{-1}$ (16.6-17.4 Tg N $yr^{-1}$) in 2020, implying a higher increase rate of 0.085 Tg N $yr^{-2}$ ($p < 0.05$).

The increase in global $N_2O$ emissions was primarily due to a 40% increase in anthropogenic emissions from 4.8 (3.1-7.3) Tg $yr^{-1}$ in 1980 to 6.7 (3.3-10.9) Tg $yr^{-1}$ in 2020. Among all anthropogenic sources, direct agricultural emissions made the largest

contribution, increasing from 2.2 (1.6-2.8) Tg N $yr^{-1}$ in 1980 to 3.9 (2.9-5.1) Tg N $yr^{-1}$ in 2020. The concurrent indirect agricultural $N_2O$ emissions also steadily increased from 0.9 (0.7-1.1) Tg N $yr^{-1}$ to 1.3 (0.9-1.6) Tg N $yr^{-1}$. In contrast, other direct anthropogenic emissions (including emissions from fossil fuel and biomass burning, industry and wastewater) did not show a significant trend, while fluxes induced by perturbations to climate, atmospheric $CO_2$, and land cover were negative and caused a reduction of $N_2O$ emissions which grew from -0.4 (-0.9-1.0) Tg $yr^{-1}$ in 1980 to -0.6 (-2.2-1.8) Tg $yr^{-1}$ in 2020. Unlike

anthropogenic emissions, global natural land and ocean $N_2O$ emissions were relatively stable, with values fluctuating between 11.5 and 11.9 Tg $yr^{-1}$.

During 2010-2019, similar estimates of global total $N_2O$ emissions were obtained using both BU and TD approaches, with decadal mean values of 18.1 (10.4–25.9) Tg N $yr^{-1}$ and 17.4 (15.8–19.2) Tg N $yr^{-1}$, respectively (Figure 1). According to the BU estimates, natural sources contributed 64% to the total emissions (11.6, 7.2–15.9 Tg N $yr^{-1}$). Specifically, natural soils

contributed the most, with a decadal average of 6.4 (3.9–8.6) Tg N $yr^{-1}$, followed by open oceans (3.5, 2.5–4.7 Tg N $yr^{-1}$), the natural source from shelves (1.2, 0.6–1.6 Tg N $yr^{-1}$), lightning and atmospheric production (0.4, 0.2–1.2 Tg N $yr^{-1}$), and inland waters, estuaries and coastal vegetation (0.1, 0.0–0.1 Tg N $yr^{-1}$). Anthropogenic sources contributed 36% to the total $N_2O$ emissions (6.5, 3.2–10.0 Tg N $yr^{-1}$). Direct agricultural emissions accounted for 56% of the total anthropogenic emissions (3.6, 2.7–4.8 Tg N $yr^{-1}$), followed by emissions from other direct anthropogenic sources ((2.1, 1.8–2.4 Tg N $yr^{-1}$), including



'Fossil fuel and industry' (1.1, 1.0-1.2 Tg N yr$^{-1}$), 'Waste and wastewater' (0.3, 0.3-0.3 Tg N yr$^{-1}$), and 'Biomass burning' (0.8, 0.5-1.0 Tg N yr$^{-1}$), and indirect anthropogenic emissions (1.2, 0.9–1.6 Tg N yr$^{-1}$). Perturbed fluxes from climate/CO2/land cover changes had a net negative effect (i.e., reduced) on $N_2O$ emissions (-0.6, -2.1–1.2 Tg N yr$^{-1}$). Increased $CO_2$ and land conversion from manure forest reduced $N_2O$ emissions, but climate change resulted in N2O emission of 0.7 (0.2-1.2) Tg N yr$^{-1}$.

Among the eighteen regions considered in this study, only Europe, Russia, and Japan and Korea had decreasing $N_2O$ emissions. Europe had the largest rate of decrease with an average of -11.4×10$^{-3}$ Tg N yr$^{-2}$ during 1980-2020 (31% reduction), largely resulting from reduced fossil fuel and industry emissions, which changed from 0.49 Tg N yr$^{-1}$ in 1980 to 0.14 Tg N yr$^{-1}$ in 2020. In addition to the large reduction of fossil fuel and industry emissions in Europe, direct and indirect agricultural emissions also declined during 1980-2020, however, the decreasing trend in direct agricultural emissions had leveled off since the 2000s.

China and South Asia had the largest increase in $N_2O$ emissions from 1980 to 2020. The rates of increase in anthropogenic emissions from China and South Asia were 18.1 x 10$^{-3}$ Tg N yr$^{-2}$ (81% increase) and 14.5 x 10$^{-3}$ Tg N yr$^{-2}$ (92% increase), respectively. In these two regions, direct nitrogen additions in agriculture made the largest contribution, while other direct and indirect emissions also steadily increased.

The atmospheric chemistry transport models used in this study show an increase in atmospheric $N_2O$ burden from 1527 (1504-
1545) Tg N in 2000-2009 to 1606 (1592-1621) Tg N in 2020, and proportional to this, a small increase in the atmospheric loss, from 12.1 (12.0-12.6) Tg N yr$^{-1}$ to 12.9 (12.5-13.2) Tg N yr$^{-1}$. The estimated increase in atmospheric $N_2O$ burden is comparable to estimates by satellite and photolysis models, showing an increase from 1528 Tg N in the 2000s to 1570 in the 2010s and 1592 Tg N in 2020. The atmospheric chemistry transport models, however, did not show any significant trend in the lifetime, which is in contrast to results based on satellite observations in the stratosphere; these observations indicate that the
atmospheric lifetime of $N_2O$ decreased from 119 years in the 2000s to 117 years in the 2010s. The reason for the discrepancy is not yet known and needs to be further investigated.

Several major uncertainties have been identified as follows: 1) inversion estimates are the most uncertain in the areas of South America, Africa, central and southern Asia, as well as Australasia, where the inversions are poorly constrained by observations. 2) Large uncertainties exist in the estimates of soil $N_2O$ emissions from tropical ecosystems in the Amazon Basin, the Congo
Basin, and Southeast Asia, as well as in regions with high fertilizer application rates and emissions, including Eastern China, Northern India, and the US Corn Belt. 3) The largest uncertainties in the estimates of ocean emissions are found in the equatorial Pacific, the Benguela upwelling region of the Atlantic, and the eastern equatorial Indian Ocean. The highest uncertainty in the equatorial upwelling and low-oxygen waters is associated with high sub-surface $N_2O$ production. 4) The $N_2O$ fluxes from atmospheric $CO_2$, manure forest conversion and biomass burning are poorly understood and quantified. The
relatively sparse distribution of current $N_2O$ observation sites underscores the necessity of establishing more sites and regular aircraft profiles, especially in tropical and subtropical regions, to better constrain emission estimates from inversion models. Based on this analysis and associated uncertainties, we propose the urgent development of a comprehensive Terrestrial and Ocean $N_2O$ Flux Monitoring and Analysis Network to better resolve spatio-temporal patterns and reduce uncertainties in $N_2O$

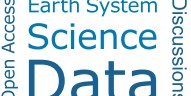

emissions. Such a development is a requirement to better constrain the future contribution of $N_2O$ to climate change and guide

policy choices to reduce $N_2O$ emissions.

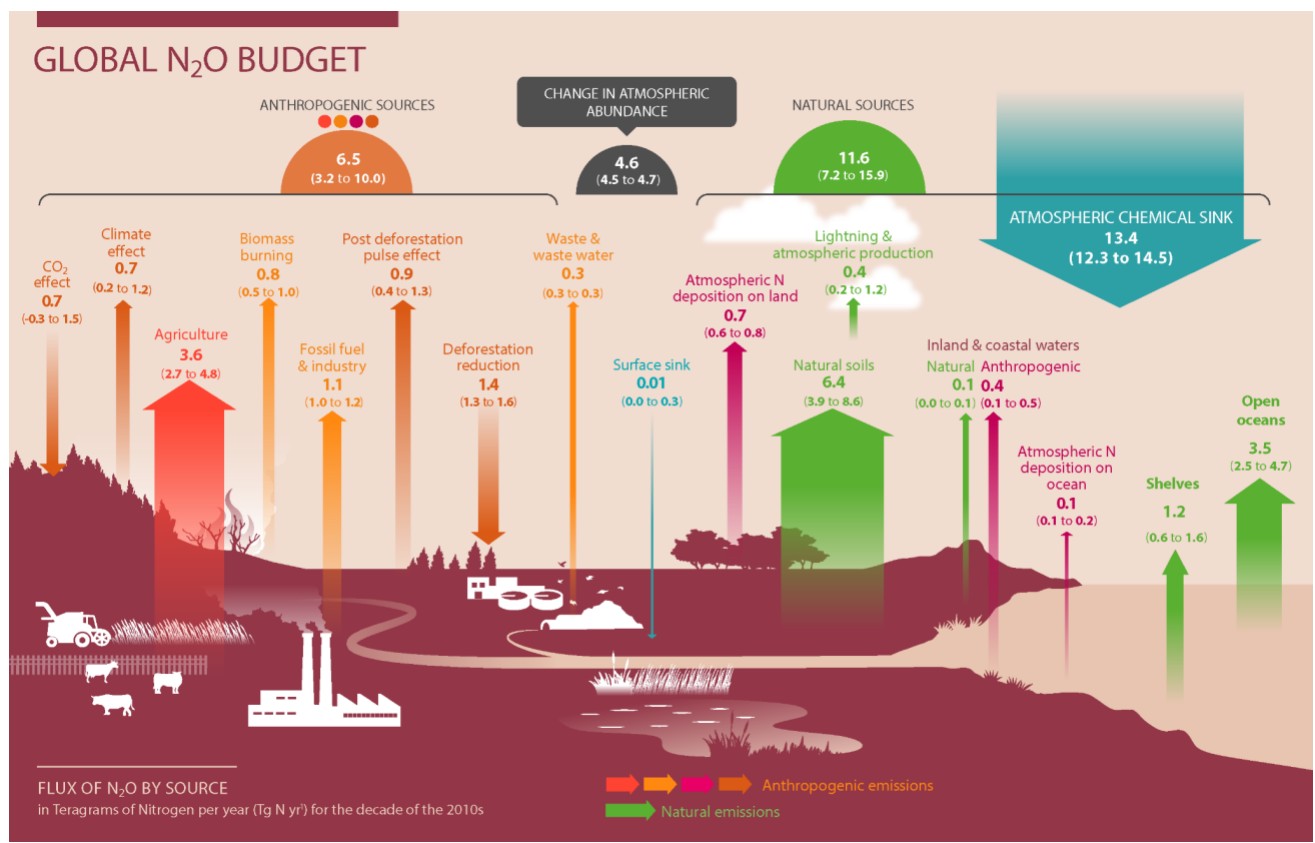

**Figure 1. Global $N_2O$ Budget during 2010-2019. The coloured arrows represent $N_2O$ fluxes (in Tg N yr$^{-1}$ for 2010–2019)**
**as follows: red, direct emissions from nitrogen additions in the agricultural sector (agriculture); orange, emissions from**
**other direct anthropogenic sources; maroon, indirect emissions from anthropogenic nitrogen additions; brown,**
**perturbed fluxes from changes in climate, $CO_2$ or land cover; green, emissions from natural sources. The anthropogenic**
**and natural $N_2O$ sources are derived from BU estimates. The blue arrows represent the surface sink and the observed**
**atmospheric chemical sink, of which about 1% occurs in the troposphere. The total budget (sources + sinks) does not**
**exactly match the observed atmospheric accumulation, because each of the terms has been derived independently and**
**we do not force TD agreement by rescaling the terms. This imbalance readily falls within the overall uncertainty in**
**closing the $N_2O$ budget, as reflected in each of the terms. The $N_2O$ sources and sinks are given in Tg N yr$^{-1}$. Copyright**
**the Global Carbon Project.**


## 1 Introduction

Nitrogen (N) is an essential element for the survival of all living organisms, required by numerous biological molecules such as nucleic acids, proteins, and chlorophyll (Galloway et al., 2021; Scheer et al., 2020). The addition of excess reactive N

compounds to terrestrial and oceanic ecosystems stimulates emissions of nitrous oxide ($N_2O$), which is currently the most emitted stratospheric ozone depleting substance (World Meteorological Organization, 2022) and a long-lived potent greenhouse gas with an atmospheric lifetime of more than 100 years (Myhre et al., 2013; Prather et al., 2015). Atmospheric $N_2O$ mole fractions have increased by more than 25% since the pre-industrial era, from 270 parts per billion (ppb) in 1750 to 336 ppb in 2022, and an increase of 35 ppb (10%) since 1980 (Figure 2). The current mole fraction is higher than at any time

in the last 800,000 years (Schilt et al. EPSL, 2010). The 20th century rate of increase in atmopsheric $N_2O$ is unprecedented over the past 20,000 years, covering the last glacial-interglacial transition, and likely unprecedented compared to the lower resolution ice core records of the past 800,000 years (Joos and Spahni, PNAS, 2007; Schilt et al., EPSL, 2010, Canadell et al., AR6, WGI, Chapter 5). The observation networks of AGAGE (Prinn et al., 2018), NOAA (Hall et al., 2007) and CSIRO (Francey et al., 2007) all show an overall increasing trend in the growth rate of atmospheric $N_2O$, the mean annual growth rate

increased from 0.76 (0.55-0.95) ppb yr$^{-1}$ in the 2000s to 0.96 (0.79-1.15) ppb yr$^{-1}$ in the 2010s, with significant seasonal and interannual variations. In 2020, the $N_2O$ atmospheric growth rate was 1.33 ppb yr$^{-1}$ (1.38 ppb yr$^{-1}$ in 2021), higher than any previous observed year since 1980, and more than 30% higher than the average value in the 2010s.

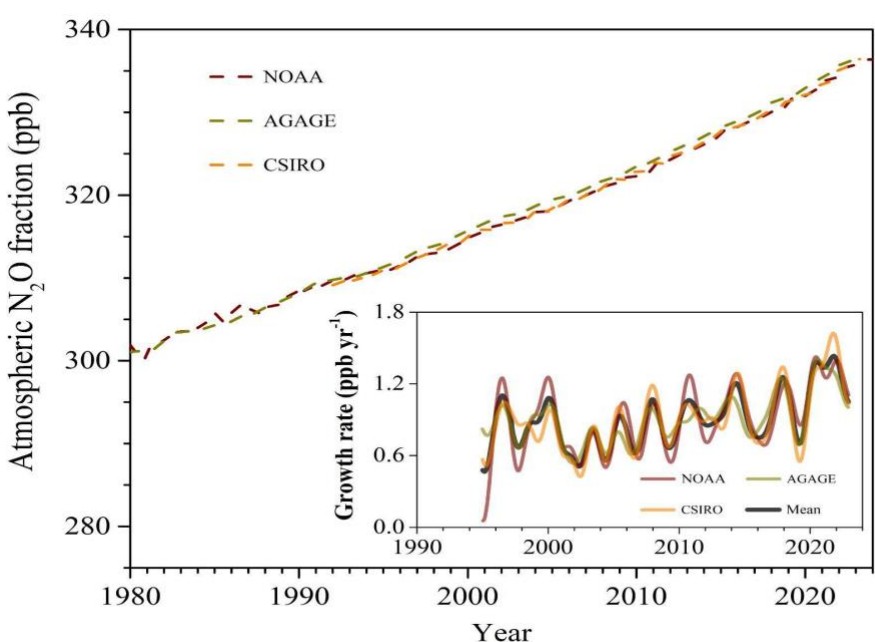

**Figure 2. Global mean atmospheric $N_2O$ dry mole fraction (atmospheric concentration) (1980-2022) and growth rate**
**(1995-2022) estimated by AGAGE, NOAA and CSIRO observing networks.**



Due to the rapid increase of global $N_2O$ emissions, observed atmospheric $N_2O$ mole fractions in recent years have begun to exceed the predicted levels under all scenarios in the Coupled Model Intercomparison Project Phase 6 (CMIP6) for the Sixth Assessment Report of the Intergovernmental Panel on Climate Change (IPCC 2021; Gidden et al., 2019; Tian et al., 2020).

$N_2O$ emissions are expected to continue increasing in the coming decades due to the growing demand for food, feed, fiber and energy, and a rising source from waste generation and industrial processes (Davidson & Kanter, 2014; Reay et al., 2012). Reducing $N_2O$ emissions is a required net-zero greenhouse gas (GHG) emissions and the recovery of stratospheric ozone (Jackson et al., 2019). $N_2O$ mitigation measures can potentially reduce GHG emissions equivalent to 5%–10% of the remaining carbon budget for holding global warming below 2 °C (Forster et al. 2021). In addition, $N_2O$ mitigation could reduce ozone

loss comparable to the depletion potential of the global chlorofluorocarbons (CFCs) stock in old air conditioners, refrigerators, insulation foams and other units (UNEP 2013). Implementing $N_2O$ mitigation will contribute to achieving the 2°C target of the Paris Agreement (Rogelj et al., 2016) and a set of United Nations Sustainable Development Goals (United Nations, 2016). Nitrification and denitrification are the two key microbial processes controlling $N_2O$ production, contributing 56-70% to global $N_2O$ emissions (Syakila & Kroeze, 2011; Tian et al., 2020); abiotic processes also play a role in the production of $N_2O$. We

categorize the processes governing $N_2O$ sources and sinks in 21 different categories (Figure 3): (1) $N_2O$ emissions from fossil fuel combustion, (2) $N_2O$ emissions from the chemical industry, (3) $N_2O$ emissions from wastewater treatment and discharge, (4) Natural $N_2O$ emissions from inland waters (rivers, lakes and reservoirs), estuaries and coastal vegetation, (5) Anthropogenic $N_2O$ from inland waters (rivers, lakes and reservoirs), estuaries and coastal vegetation, (6) Direct $N_2O$ emissions from agricultural soils, (7) $N_2O$ emissions from manure left on pasture, (8) $N_2O$ emission from manure management, (9) $N_2O$

emissions from coastal and freshwater aquaculture, (10) $N_2O$ emission/reduction due to agricultural land use and conservation, (11) Natural soil $N_2O$ emission, (12) $N_2O$ emissions from biomass burning, (13) Surface $N_2O$ uptake, (14) Indirect $N_2O$ emissions from anthropogenic nitrogen additions on land, (15) Perturbed $N_2O$ fluxes from climate/$CO_2$, (16) $N_2O$ emission/reduction due to land cover change/deforestation, (17), $N_2O$ emission from continental shelves, (18) $N_2O$ emission from open ocean, (19) $N_2O$ emissions from anthropogenic N deposition on oceans, (20) Lightning and atmospheric production

of $N_2O$, (21) Stratospheric $N_2O$ sink. There is also a small amount of $N_2O$ emission from termite mounds, but such an $N_2O$ flux is not quantified in the current budget analysis due to limited data.

Biogenic $N_2O$ emissions from land are regulated by multiple environmental factors, including soil moisture, temperature, oxygen status, pH, vegetation type, topography, atmospheric $CO_2$ concentration, and soil N and C availability (Butterbach-Bahl et al., 2013; Dijkstra et al., 2012; Li et al., 2020; Tian et al., 2019; Yin et al., 2022; H. Yu et al., 2022). The effects of

these environmental factors on $N_2O$ emissions have strong spatial and temporal heterogeneity, making up-scaling field $N_2O$ measurements to regional and global scales difficult. Studies using atmospheric $N_2O$ inverse modeling suggest a greater source of $N_2O$ from land and ocean in the colder and wetter La Nina conditions and vice versa in the warmer and drier El Niño conditions (Patra et al., 2022; Thompson et al., 2014). Ongoing environmental changes such as ocean warming (and associated changes in stratification and ice coverage), decreasing pH (i.e. increasing acidification), loss of dissolved oxygen (i.e.

deoxygenation), and eutrophication due to increasing anthropogenic inputs of nutrients via rivers and atmospheric deposition





of nitrogen aerosols, might significantly alter the production and consumption of $N_2O$ in the upper ocean, its distribution pattern and, ultimately, its release to the atmosphere (Bange et al., 2019, 2022; Wilson et al., 2019), exerting in the long term small but uncertain feedback on global warming (Battaglia and Joos, GBC, 2018, Forster et al., 2021) .

In this study, we construct a comprehensive global and regional $N_2O$ budget based on the processes and framework shown in
Figure 3 and following the framework of Tian et al. (2020). The figure summarizes the pathways of $N_2O$ formation, consumption, emission and absorption, and it helps to guide consistent estimations and comparisons of $N_2O$ budgets among regions and upscaling of regional budgets to the globe. $N_2O$ fluxes are grouped into two major categories based on the sources. The first category is natural $N_2O$ fluxes (blue arrows in Figure 3), which are $N_2O$ fluxes in the absence of climate change and anthropogenic disturbances, and include natural soil emissions, soil uptake, $N_2O$ emission from natural disturbances causing
wetland loss and degradation, lightning, and atmospheric production. This category also includes natural emissions from inland waters, estuaries, coastal zones and the ocean.

The second category is anthropogenic $N_2O$ fluxes (red arrows in Figure 3). The direct emissions from nitrogen additions in the agricultural sector ("agroecosystems" box in Figure 3) include emissions from direct application of synthetic nitrogen fertilizers and manure (henceforth "direct soil emissions"), manure left on pasture, manure management and aquaculture, while
other direct anthropogenic sources include fossil fuel combustion and industry, waste and wastewater, and biomass burning. Indirect $N_2O$ emissions derive from anthropogenic nitrogen additions such as atmospheric nitrogen deposition (NDEP) on land and ocean, and the effects of anthropogenic loads of reactive nitrogen in inland waters, estuaries, and coastal zones.

In the anthropogenic $N_2O$ fluxes category, we also consider $N_2O$ fluxes from the anthropogenic perturbations in climate, $CO_2$ and land-use/land-cover (from hereon perturbation fluxes). In terrestrial natural ecosystems, perturbation fluxes can be caused
by increasing $CO_2$ concentration, climate change (e.g., warming-induced thawing of permafrost), and land-use change (e.g., converting natural lands to lands for human uses, such as croplands, mining, logging, and the post-deforestation pulse effect, the long-term effect of reduced mature forest area). $N_2O$ emissions can either increase or decrease during land conversion depending on the type and phase of the land-use change. For example, when tropical forests are first converted to agriculture there is often a pulse of $N_2O$ emissions for the first year or for as long as five years, depending upon the circumstances;
following deforestation, emissions decline below those of the original forest if pastures degrade and if croplands are not fertilized, such as in slash-and-burn agriculture (Davidson and Artaxo, 2004, Meurer et al., 2016). When agriculture is abandoned and a secondary forest is allowed to regrow, $N_2O$ emissions gradually increase but usually remain lower than those of the original mature forest or from fertilized croplands (Davidson et al., 2007, Sullivan et al., 2019).





**Figure 3. N₂O sources and sinks and flux partitions contributing to the global N₂O budget. Upwards pointing arrows indicate a source to the atmosphere and downward pointing arrows represent a sink.**



Numerous efforts have estimated individual sources and sinks of $N_2O$ across global ecosystems. Prominently, anthropogenic $N_2O$ emissions have been annually reported for the past two decades by Annex I Parties (developed countries) to the United Nations Framework Convention on Climate Change (UNFCCC) (Reports | UNFCCC). As a result of the Paris Agreement, over 190 signatory countries are now required to report their national GHG inventory biannually, if not already reported

annually, with sufficient detail and transparency to track progress towards their Nationally Determined Contributions. However, national GHG inventories only provide a partial picture of the observed changes in atmospheric $N_2O$. They do not cover natural sources and have large uncertainties in the emission factors and activity data. Additionally, data are limited in many regions of the world, e.g., South America and Africa (Tian et al. 2020).

Tian et al. (2020) built the first comprehensive global $N_2O$ budget using multiple BU (BU) and TD (TD) methods as part of a

partnership between the Global Carbon Project (GCP) and the International Nitrogen Initiative (INI). Based on Tian et al. (2020) and the budget framework established in Figure 3, our study presents an improved and updated global $N_2O$ budget and its regional attribution to 18 land regions and the global ocean. The budgets cover the decades of 1980-89, 1990-99, 2000-09, 2010-2019, with a complete budget extension to 2020 and atmospheric $N_2O$ changes in 2021 and 2022. The work allows us to explore the relative temporal and spatial importance of multiple sources and sinks that drive the atmospheric burden of $N_2O$,

their uncertainties, and interactions between anthropogenic and natural forcings. This study also consolidates the international scientific capacity and networks that contribute to this assessment with the aim to provide improved and updated $N_2O$ budgets at regular intervals.

This global effort builds from and contributes to the set of global GHG assessments that the GCP has established including regular updates of the carbon ($CO_2$-C), methane ($CH_4$), and now $N_2O$ budgets, and other biogeochemical budgets of global

significance. The budgets have been designed to: a) support global and national scientific assessments (e.g., IPCC, WCRP annual reports), b) align scientific research and data products to support climate mitigation and sustainability policy needs, and c) contribute to the global stocktake of the Paris Agreement to track progress towards national determined contributions and the ultimate goal of achieving net-zero GHG emissions. Integration of all GHGs in robust and shared methodological approaches and data delivery platforms are central goals of GCP.

**2 Methodology and Data**

**2.1 Definitions, terminology and unit of $N_2O$ sources and sinks**

This study provides an estimation of the global $N_2O$ budget considering all possible sources, sinks and perturbations, a total of 21 $N_2O$ fluxes. To simplify our analysis, we further grouped these fluxes into six major categories: (1) 'natural baseline fluxes': this is the source in the absence of climate change and anthropogenic disturbances and includes emissions from soils,

surface uptake, shelf and ocean emissions, lightning and atmospheric production, and emissions from inland waters, estuaries, and coastal vegetation; (2) direct emissions from nitrogen additions in the agricultural sector ('agriculture'), which includes emissions from direct application of nitrogen fertilizers and manure (henceforth 'direct soil emissions'), manure left on pasture,



manure management and aquaculture; (3) 'perturbed fluxes from climate/$CO_2$/land cover change' which includes the effects of $CO_2$, climate, the post-deforestation pulse, and the long-term effect of reduced mature forest area; (4) indirect emissions

from anthropogenic nitrogen additions including atmospheric nitrogen deposition (NDEP) on the land, atmospheric NDEP on the ocean, and effects of anthropogenic loads of reactive nitrogen in inland waters, estuaries and coastal zones; (5) other direct anthropogenic sources including fossil fuel and industry, waste and wastewater, and biomass burning; and (6) the atmospheric sink in the stratosphere (via photolysis and oxidation by $O^1D$). Our anthropogenic $N_2O$ emission categories are aligned with those compiled by the national greenhouse gas inventories using IPCC 2006 methodologies and reported to the UNFCCC

(Table A1).

In this study, $N_2O$ fluxes are expressed in teragrams of $N_2O$-N per year: $1\,Tg\,N_2O$-N $yr^{-1}$ ($1\,Tg\,N\,yr^{-1}$) $=10^{12}\,g\,N_2O$-N $yr^{-1}=1.57\times10^{12}\,g\,N_2O\,yr^{-1}$, with change rates in $N_2O$ fluxes expressed in the unit of $Tg\,N_2O$-N $yr^{-2}$ ($Tg\,N\,yr^{-2}$). Atmospheric $N_2O$ is expressed as dry air mole fractions, in parts per billion (ppb), with atmospheric $N_2O$ annual increases expressed in parts per billion per year. Unless specified, uncertainties are reported in brackets as minimum and maximum values of all estimates,

following Tian et al., (2020).

**2.2 Definition of Regions**

As anthropogenic emissions are often reported at the country level, we divide global land into 18 regions and define these regions based on a country list (Table A2). This approach is compatible with all TD and BU approaches considered here. The number of regions was close to the widely used TransCom inter-comparison map (Gurney et al., 2004), but with subdivisions

to separate the contribution of important countries or regions to the global $N_2O$ budget (such as China, South Asia and the United States). This regionalization is also compatible with the REgional Carbon Cycle Assessment and Processes (Poulter et al. 2022) after aggregation into ten regions. The 18 regions are United States (USA), Canada (CAN), Central America (CAM), Northern South America (NSA), Brazil (BRA), Southwest South America (SSA), Europe (EU), Northern Africa (NAF), Equatorial Africa (EQAF), Southern Africa (SAF), Russia (RUS), Central Asia (CAS), Middle East (MIDE), China (CHN),

Korea and Japan (KAJ), South Asia (SAS), Southeast Asia (SEAS), and Australasia (AUS). The region definition is the same as that used for the GCP methane and $N_2O$ budgets (Saunois et al., 2020; Stavert et al., 2022; Tian et al., 2019).

**2.3 Overview of methods used for global $N_2O$ budget synthesis**

Four major methods are available to estimate large-scale $N_2O$ emissions: atmospheric inversion models (method 1), activity and emission factor-based inventories (method 2), empirical-based algorithms and machine learning algorithms (method 3),

and process-based ecosystem models (method 4). Atmospheric inversion models (method 1), a TD approach, utilizes measurements of atmospheric $N_2O$ mixing ratios combined with atmospheric transport models, driven by meteorological fields, to estimate the emissions of $N_2O$ (Thompson et al., 2014). Atmospheric inversion models usually use Bayesian statistics, which starting from a prior emission estimate, find the optimal $N_2O$ emissions, that is those that best agree with observed



atmospheric N$_2$O mixing ratios, while at the same time being guided by the prior emission and observation uncertainties
(Nevison et al., 2018; Thompson et al., 2019).

TD approaches generally only estimate the total N$_2$O emission, which is spatially and temporally resolved, but do not constrain the contributions from different sources. The other three methods belong to BU approaches, which are capable of quantifying N$_2$O emissions from different sources. Emission activity and factor-based inventories (method 2) use a prescribed emission factor (EF) to calculate N$_2$O emissions. This approach has been widely used in national emission inventories and global studies
(Davidson, 2009; Oreggioni et al., 2021; Crippa et al., 2021; Winiwarter et al., 2018). Nevertheless, the fixed EFs cannot capture the nonlinear response of agricultural soil N$_2$O emissions to N inputs (Gerber et al., 2016), and also cannot fully reflect the dependence of EFs on climate, management practices, soil physical and biochemical conditions (e.g., Marzadri et al 2022). Therefore, a spatially referenced nonlinear model (SRNM) was developed to simulate N$_2$O emissions in response to fertilizer application under various environmental and management conditions, which outperformed the default EF method (Zhou et al.,
2015). In recent years, machine learning algorithms (method 3) have been applied to estimate soil N$_2$O emissions. A random forest model was used to estimate global terrestrial background N$_2$O emissions (Yin et al., 2022) and N$_2$O emissions from intensively managed cropping systems (Saha et al., 2021). Moreover, a machine-learning-based stochastic gradient boosting model was developed to predict global terrestrial nitrification and its fraction in N$_2$O emissions (Pan et al., 2021).

Compared with the three above-mentioned methods, process-based ecosystem models (method 4) have two notable advantages
(Xu et al. 2020; Tian et al. 2019): (1) they are capable of modeling the key processes affecting N$_2$O production and emission such as autotrophic nitrification, denitrification, plant nitrogen uptake, ammonia volatilization, nitrate leaching, soil thermal and hydrological processes; and (2) they integrate various driving factors controlling soil N$_2$O emissions, such as fertilizer use, atmospheric N deposition, land use change, climate change, and atmospheric CO$_2$ concentration change and thus can disentangle the effects of different driving factors. Although multiple process-based models estimated global soil N$_2$O
emissions, large discrepancies exist in these estimates due to the diverse parameterizations of biogeochemical processes in different models, our limited understanding of the mechanisms responsible for N$_2$O emissions, and the uncertainties in input data. The N$_2$O Model Intercomparison Project (NMIP) was launched (Tian et al., 2018; Tian et al., 2019) to develop a multi-model ensemble estimation of global soil N$_2$O emissions during 1861-2016 and quantified the contributions of different driving factors.

We consider global N$_2$O emissions from land and ocean including natural fluxes and anthropogenic emissions based on BU and TD approaches (Figure 4). The BU methods considered include eight process-based terrestrial biosphere models from NMIP2 (global Nitrogen/N$_2$O Model Inter-comparison Project phase 2), six ocean models (Battaglia and Joos, 2018; Berthet et al., 2023; Buitenhuis et al., 2018; Carroll et al., 2020; Landolfi et al., 2017) and one machine-learning based observational shelf product (Yang et al., 2020), a mix of five approaches relying on meta-analysis, statistical and process-based models for
inland waters and coastal ecosystems (Hu et al., 2016; Lauerwald et al., 2019; Maavara et al., 2019, Yao et al., 2020; Marzadri et al., 2021; Marzadri et al., 2022; Rosentreter et al., 2023); four GHG emission databases - Emissions Database for Global Atmospheric Research EDGAR v7.0 (Crippa et al., 2021, https://edgar.jrc.ec.europa.eu/dataset_ghg70), FAOSTAT (Tubiello



et al., 2015), UNFCCC (https://unfccc.int/reports), GFED4s (van der Werf et al., 2017) (only for biomass burning) - and one
statistical model (SRNM) only for cropland soils (Wang et al., 2020). The TD approach consisted of four independent
atmospheric inversion frameworks, namely INVICAT (Wilson et al., 2014), PyVAR-CAMS (Thompson et al., 2014),
MIROC4-ACTM (Patra et al., 2022), and GEOS-Chem (Wells et al., 2018).

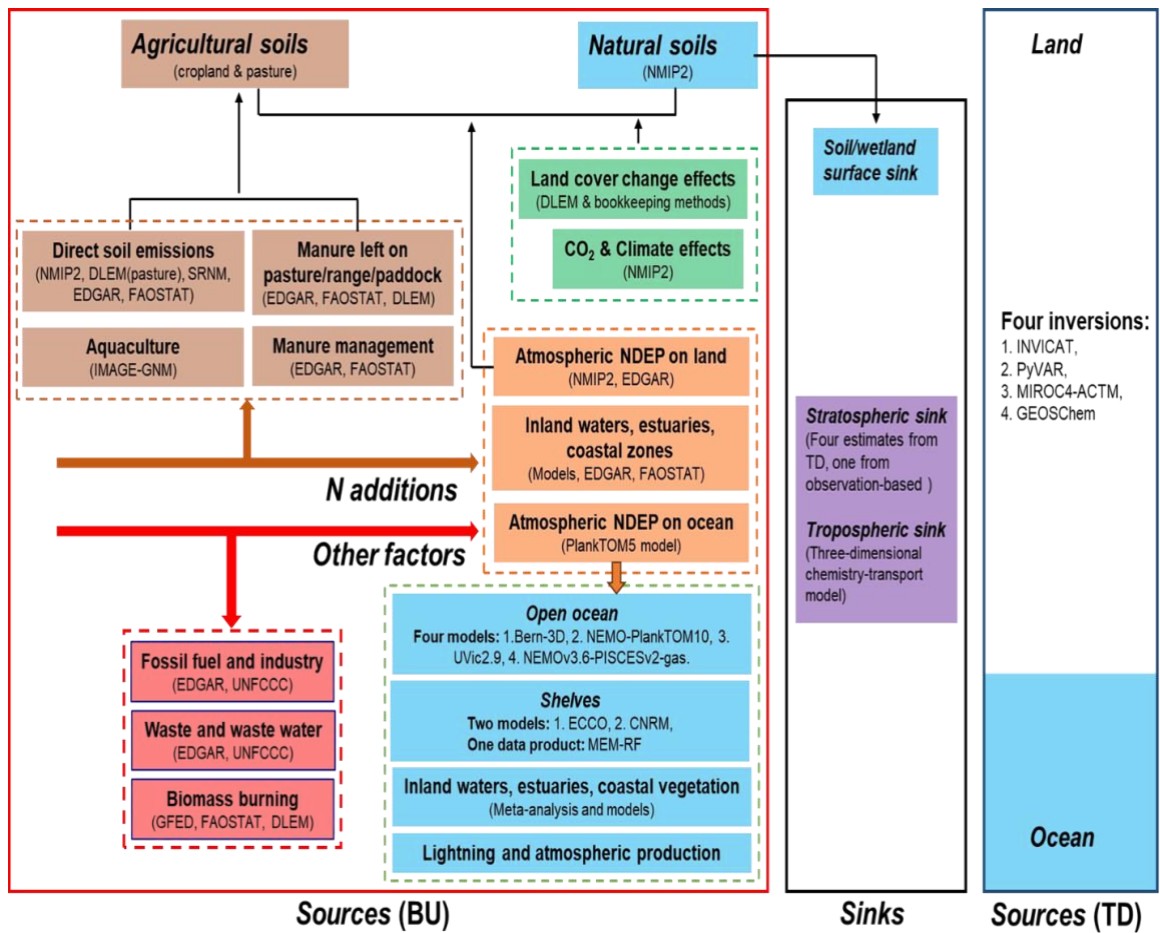

**Figure 4. Methodologies used to estimate each of the main flux categories contributing to the global $N_2O$ budget. BU
and TD represent BU and TD methods, respectively. We use both approaches, including 20 BU and four TD estimates
of $N_2O$ fluxes from land and oceans. For sources estimated by the BU approach, we include eight process-based
terrestrial biosphere modelling studies; six process-based ocean biogeochemical models and one shelf observational
product; one nutrient budget model; five inland and coastal water modelling or meta-analysis studies; one statistical
model SRNM based on spatial extrapolation of field measurements; and four greenhouse-gas inventories: EDGAR
v7.0, FAOSTAT, UNFCCC, and GFED. Previous estimates of surface sink, lightning and atmospheric production,
model-based tropospheric sink and observed stratospheric sink are included in the current synthesis. The nutrient
budget model provides nitrogen flows in global freshwater and marine aquaculture over the period 1980–2020. Model-
based estimates of $N_2O$ emissions from inland and coastal waters include rivers and reservoirs, lakes, estuaries, coastal
zones (that is, seagrasses, mangroves, saltmarsh and intertidal saltmarsh) and coastal upwelling.**





**Table 1. Methods, spatial and temporal resolution and data sources for the synthesis of the global N₂O budget**

| Model/Data name | Spatial resolution | Time period | References |
|---|---|---|---|
| **Inventories** | | | |
| EDGAR v7 | 0.1°×0.1° | 1980-2020 | Crippa et al. (2021), |
| GFED4s | 0.25°×0.25° | 1997-2020 | Van Der Werf et al. (2017) |
| FAOSTAT | Country-level | 1980-2020 | Tubiello et al. (2022) |
| UNFCCC | Country-level | 1990-2020 | https://di.unfccc.int/time_series |
| **Terrestrial Biosphere models participated in NMIP2** | | | |
| CLASSIC | 0.5°×0.5° | 1980-2020 | Asaadi and Arora (2021) Kou-Giesbrecht and Arora (2022) |
| DLEM | 0.5°×0.5° | 1980-2020 | Tian et al. (2015), Xu et al. (2017) |
| ELM | 0.5°×0.5° | 1980-2020 | Zhu et al. (2019) |
| ISAM | 0.5°×0.5° | 1980-2020 | Shu et al. (2020); Xu et al. (2021) |
| LPX-Bern | 0.5°×0.5° | 1980-2020 | Xu and Prentice (2008), Stocker et al. (2013) |
| O-CN | 1°×1° | 1980-2020 | Zaehle et al. (2011) |
| ORCHIDEE | 0.5°×0.5° | 1980-2020 | Vuichard, N., et al. (2019) |
| VISIT | 0.5°×0.5° | 1980-2020 | Ito et al. (2018) |
| **Ocean Biogeochemical Models** | | | |
| Bern-3D | 9° · 4.5° · 32 levels | 1980-2019 | Battaglia and Joos (2018) |
| NEMOv3.6-PISCESv2-gas | 1° · 1° · 75 levels | 1980-2020 | Berthet et al. (2023); Seferian et al. (2019) |
| NEMO-PlankTOM10.2 | 2° · (0.5°−2°) · 30 levels | 1980-2016 | Buitenhuis et al. (2018) |
| UVic2.9 | 3.6° · 1.8° · 19 levels | 1980-2020 | Landolfi et al. (2017) |
| **Shelf products** | | | |
| MEM-RF | 0.25°×0.25° | 1988-2017 mean | Yang et al. (2020) |
| CNRM-0.25° | 0.25°×0.25° | 1998-2018 mean | Berthet et al. (2019) |
| ECCO2-Darwin & ECCO-Darwin | 1/3° (ECCO-Darwin) - 1/6°(ECCO2-Darwin) | 1998-2013 mean (ECCO-Darwin), | Ganesan et al. (2020) Carroll et al. (2020) |





| | | 2006-2013 mean (ECCO2-Darwin) | |
|---|---|---|---|
| **Inland waters, estuaries and coastal vegetation** | | | |
| DLEM-TAC | 0.5°×0.5° | 1980-2019 | Yao et al. (2020), Tian et al. (2020) |
| Mechanistic Stochastic Model | 0.5°×0.5° | 2000 | Lauerwald et al. (2019); Maavara et al. (2019) |
| Meta analysis- based upscaling | watershed-level 18 regions | 2000 1975-2020 | Hu et al. (2016) Rosentreter et al. (2023) |
| Integrated ML & Physical model | 0.5°×0.5° | 2000 | Marzadri et al. (2021) |
| **Atmospheric inversion models** | | | |
| INVICAT | 5.625°×5.625° | 1997-2020 | Wilson et al. (2014) |
| PyVAR-CAMS | 3.75°×1.875° | 1997-2020 | Thompson et al. (2014) |
| MIROC4-ACTM | ~2.8°×2.8° | 1997-2019 | Patra et al. (2018,2022) |
| GEOS-Chem | 5°×4° | 1997-2019 | Wells et al. (2018) |
| **Other models and datasets** | | | |
| | | | |
| SRNM | 1/12°×1/12° | 1980-2020 | Wang et al. (2020) |
| Bookkeeping method | 0.25°×0.25° | 1980-2020 | Tian et al. (2020), Keller and Reiners (1994) |
| IMAGE-GNM | Country-level | 1980-2020 | Bouwman et al. (2011), Bouwman et al. (2013a) |

## 2.4 Model and inventory data synthesis

### 2.4.1 Natural N$_2$O fluxes

'Natural soil baseline' emissions were obtained from the ensemble mean of the eight terrestrial biosphere models participated in NMIP-2 that run with pre-industrial land cover (Table 1) : (1) Canadian Land Surface Scheme including Biogeochemical Cycles (CLASSIC) (Asaadi & Arora, 2021; Melton et al., 2020; Kou-Giesbrecht & Arora, 2022), (2) the Dynamic Land Ecosystem Model (DLEM) (Tian, et al., 2015; Xu et al., 2017; You et al., 2022), (3) E3SM Land Model (ELM) (Zhu et al., 2019), (4) the Integrated Science Assessment Model (ISAM) (Shu et al., 2020; Xu et al., 2021), (5) Land Processes and

eXchanges model - Bern (LPX-Bern v1.4) (Lienert and Joos, 2018; Joos et al., 2020), (6) O-CN (Zaehle et al.,2011), (7)





Organising Carbon and Hydrology In Dynamic Ecosystems (ORCHIDEE) (Goll et al., 2017), and (8) Vegetation Integrated SImulator for Trace gases (VISIT) (Ito et al., 2018).

Natural emission from 'Inland water, estuaries, coastal vegetation' including inland and coastal waters were obtained from models by Yao et al. (2020), Maavara et al. (2019), Lauerwald et al. (2019), Marzadri et al. (2021), and the meta-analyses by

Hu et al. (2016), Rosentreter et al. (2023). Since the data (rivers, lakes, reservoirs, and estuaries) provided by Hu et al. (2016), Maavara et al. (2019), Lauerwald et al. (2019), and Marzadri et al. (2021) are for the year 2000, we assumed that these values are constant during 1980−2020. Yao et al. (2020) provided annual riverine $N_2O$ emissions using DLEM during 1980-2019. Here, we averaged riverine estimates from Yao et al. (2020), Maavara et al. (2019), Hu et al. (2016), and Marzadri et al. (2021), assuming that estimates of Maavara et al. (2019) and Hu et al. (2016) represent emissions from larger rivers only, while Yao

et al. (2020) and Marzadri et al. (2021) also account for emissions from streams and small rivers. Note further that the estimate by Marzadri et al. (2021) is not fully global as it excludes river systems North of 60°N. Therefore, we did not use this assessment for the regions of Canada, US, Russia and Europe. DLEM also estimated annual $N_2O$ emissions from global reservoirs, and we averaged these estimates with those from Maavara et al. (2019) to represent emissions from reservoirs during 1980−2020. The estimate for global and regional lakes was based on the long-term averaged values provided by

Lauerwald et al. (2019) and an estimate by the DLEM-TAC model (Li et al. submitted). For estuaries, we combined the estimate by Maavara et al. (2019) which relies on a process-based modeling approach with a new meta-data analysis by Rosentreter et al. (2023). The observation-based analysis includes the contribution of coastal vegetated ecosystems, a contribution not accounted for in Maavara et al. (2019). Estuaries and coastal vegetation data are from studies published between 1975-2020 and we assume fluxes are constant during 1980-2020 (Rosentreter et al. 2023).To disentangle natural and

anthropogenic fluxes, we considered the emissions in the year 1900 simulated by DLEM (Yao et al., 2020) as equivalent to the natural emission, assuming that the N load from land was negligible in that period (Kroeze et al., 1999). Using this approach, we quantified the contribution of natural sources to total emission from reservoirs, lakes, estuaries and coastal vegetation to be 44% (36%−52%), taking into account all N inputs (i.e., inorganic, organic, dissolved, and particulate forms).

$N_2O$ emissions from continental shelves were calculated using one data-driven estimate and three high-resolution model

estimates for various time periods (Resplandy et al., 2023, also see Supplementary Information SI-7), namely an observation-based estimate that relied on a random-forest (RF) algorithm to interpolate $N_2O$ data (Yang et al., 2020), based on a synthesis of over 158,000 observations of $N_2O$ mixing ratio, partial pressure, and concentration in the surface ocean from the MEMENTO database (MEM-RF) (Kock and Bange, 2015), an estimate relying on the high-resolution configuration (Berthet et al., 2019) of the global ocean-biogeochemical component of CNRM-ESM2-1 (CNRM-0.25°), and two estimates relying on

the ECCO-Darwin model run at 1/3° (ECCO-Darwin1) and 1/6° (ECCO-Darwin2), respectively. Considering that ECCO-Darwin1 and Darwin2 relied on the same model, their mean $N_2O$ fluxes were used.

Estimates of natural $N_2O$ emissions from open oceans are derived from four global ocean biogeochemistry models including Bern-3D (Battaglia and Joos, 2018), NEMOv3.6-PISCESv2-gas (Berthet et al., 2023), NEMO-PlankTOM10 (Buitenhuis et



al., 2018), and UVic2.9 (Landolfi et al., 2017). Towards the $N_2O$ budget synthesis, modeling groups reported gridded monthly fluxes at a $1^\circ \times 1^\circ$ resolution for the period 1980-2020. Specific details on ocean model configurations and $N_2O$ parameterizations are reported in the individual model publications.

We combined the estimate from lightning with that from atmospheric production into an integrated category 'Lightning and atmospheric production' (Kolhmann and Poppe, 1999; Dentener and Crutzen, 1994). We simplified the 'Lightning and

atmospheric production' category as purely natural, although atmospheric production is affected to some extent by anthropogenic activities such as enhancement of the concentrations of the reactive species $NH_3$ and $NO_2$. This category is in any case very small and the anthropogenic enhancement effect is uncertain. The estimate of 'Surface sink' was obtained from Schlesinger (2013) and Syakila et al. (2010).

### 2.4.2 Direct emissions from nitrogen additions (agriculture)

Agriculture $N_2O$ emissions consist of four components: 'Direct soil emissions', 'Manure left on pasture', 'Manure management', and 'Aquaculture'. Data for 'Direct soil emissions' were obtained as the ensemble mean of $N_2O$ emissions from the average of two inventories (EDGAR v7.0 and FAOSTAT), the SRNM/DLEM models, and the NMIP2/DLEM models. The statistical model SRNM only covers cropland $N_2O$ emissions. Thus, we added the DLEM-based estimate of pasture $N_2O$ emissions into the two estimates of cropland to represent direct agricultural soil emissions (i.e., SRNM/DLEM or

NMIP2/DLEM). 'Manure left on pasture' is the ensemble mean of EDGAR v7.0, FAOSTAT, and DLEM. 'Manure management' emissions are the mean of EDGAR v7.0 and FAOSTAT. Global N flows (i.e., fish feed intake, fish harvest, and waste) in freshwater and marine aquaculture were obtained from Bouwman et al. (2011), Bouwman et al. (2013a) and Beusen et al. (2016) and based on IMAGE-GNM aquaculture nutrient budget model for the period 1980−2020. We then calculated global aquaculture $N_2O$ emissions as an 1.8% loss of N waste in aquaculture, i.e., the same EF used in Hu et al. (2012) and

MacLeod et al. (2019). The uncertainty range of the EF is from 0.5% (Eggleston et al., 2006) to 5% (Williams and Crutzen, 2010), the same range used in the UNEP report (Bouwman et al., 2013b).

### 2.4.3 Emissions from other direct anthropogenic sources.

This category includes 'Fossil fuel and industry', 'Waste and wastewater', and 'Biomass burning'. Both emissions from 'Fossil fuel and industry' and 'Waste and wastewater' were calculated as the ensemble means of EDGAR v7.0 and UNFCCC

databases. The 'Biomass burning' emission is the ensemble mean of FAOSTAT, DLEM, and GFED4s databases. In EDGAR v7.0, 'Waste and wastewater' includes 'Waste incineration' and 'Wastewater handling'. We merged 'Transportation', 'Energy', 'Industry', and 'Residential and other sectors' to represent the total emission from 'Fossil fuel and industry'. In addition to the IPCC agriculture burning categories 'Burning crop residues' and 'Burning savannah', we included FAOSTAT estimates for $N_2O$ emissions from deforestation fires, forest fires and peatland fires (Prosperi et al., 2020). The FAOSTAT

emissions database of the Food and Agriculture Organization of the United Nations (FAO) covers emissions of $N_2O$ from agriculture and land use by country and globally, from 1961 to 2020 for agriculture, and from 1990 for relevant land use



categories, i.e., cultivation of histosols, biomass burning, etc., applying only Tier-1 coefficients (Tubiello et al., 2022; 2021; Conchedda and Tubiello, 2020; Prosperi et al., 2020).

### 2.4.4 Indirect emissions from anthropogenic N additions

This category considers N deposition on land and ocean ('N deposition on land' and 'N deposition on ocean'), as well as the N leaching and runoff from upstream ('Inland and coastal waters'). The emission from 'N deposition on ocean' was provided by Suntharalingam et al. (2012), while emission from 'N deposition on land' was the average of two estimates by NMIP2/EDGAR v7.0 and NMIP2. EDGAR v7.0 provided estimates of indirect emissions from both agricultural and non-agricultural sectors, however, here, we sum the ensemble mean of NMIP2 estimates of indirect emissions from agricultural

sectors with indirect emissions from non-agricultural sector of EDGAR v7.0 (i.e., NMIP2/EDGAR v7.0) to represent N deposition induced soil emissions from both agricultural and non-agricultural sectors. The $N_2O$ emissions from 'Inland and coastal waters' consist of rivers, reservoirs, lakes, estuaries, and continental shelves, which is the ensemble mean of an average of two inventories (EDGAR v7.0 Indirect $N_2O$ emissions - leaching and runoff - and FAOSTAT), and the mean of meta-analysis and models. The anthropogenic emission from inland freshwaters estimated by Yao et al. (2020) considered annual

N inputs and other environmental factors (i.e., climate, elevated $CO_2$, and land cover change). For the long-term average in rivers, reservoirs, estuaries and lakes estimated by empirical methods, we applied a mean of 56% (based on the ratio of anthropogenic to total N additions from land) to calculate anthropogenic emissions. Seagrass, mangrove, saltmarsh and intertidal $N_2O$ emissions were updated from Rosentreter et al., (2023). Coastal wetlands with low disturbance generally either have low $N_2O$ emissions or act as a sink for $N_2O$ (Erler et al., 2015; Murray et al., 2020).

### 2.4.5 Perturbation of N2O fluxes from climate/CO2/land cover change


The estimate of climate and $CO_2$ effects on emissions was based on eight NMIP2 models, and we used SH1−SH7 and SH1−SH8 to model the effects of $CO_2$ and climate on global terrestrial soil $N_2O$ emissions, respectively. The effect of land cover change on $N_2O$ dynamics includes the reduction due to 'Long-term effect of reduced mature forest area' and the additional emissions due to 'Post-deforestation pulse effect'. The two estimates were based on the book-keeping approach and

the DLEM model simulation. The book-keeping method is developed by (Houghton et al., 1983) for accounting for carbon flows due to land use. A similar book-keeping method was developed to account for $N_2O$ emission due to deforestation (see Supplementary Information SI-9).

### 2.4.6 Atmospheric production of reactive nitrogen

$N_2O$ production in the atmosphere is a relatively small component of the global budget. $N_2O$ is produced by the gaseous phase oxidation of $NH_3$ in the troposphere, however, there are few published estimates of this source and it remains poorly constrained. In this paper, we refer to the two known published estimates, which are 0.4 Tg N $yr^{-1}$ (Kolhmann and Poppe,



1999) and 0.6 (0.3-1.1) Tg N yr$^{-1}$ (Dentener and Crutzen, 1994), that are derived using global models of atmospheric chemistry and transport. Since human activities have greatly affected the atmospheric abundance of $NH_3$ a significant portion of this source may be considered anthropogenic. Lightning production of $NO_x$ indirectly leads to $N_2O$ emission through its oxidation and subsequent deposition on land and ocean. A recent study estimated the global lightning production of $NO_x$ to be 9 Tg N yr$^{-1}$ (Nault et al. 2017), which is larger than previous estimates of 5 (2-8) Tg N yr$^{-1}$ (Schumann and Huntrieser et al. 2007). In this study, we assume an effective emission factor of 1% (de Klein et al. 2006) and using the median estimate of 5 Tg N yr$^{-1}$ we estimate a global source of $N_2O$ of 0.05 (0.02-0.09) Tg N yr$^{-1}$. There is also N2O production from N2 +O(1D), about 2% of atmospheric source (Estupiñán et al. 2005).

## 2.5 Atmospheric observation data synthesis

### 2.5.1 Atmospheric burden and trends from tropospheric observations

The monthly tropospheric $N_2O$ mole fraction and their growth rates are derived from three different atmospheric observational networks: The Advanced Global Atmospheric Gases Experiment (AGAGE, Prinn et al. 2018), The Commonwealth Scientific and Industrial Research Organization (CSIRO, Francey et al. 2003) and the National Ocean and Atmospheric Administration (NOAA, Dutton et al. 2023; Lan et al. 2022). Further information on the three networks' stations, instruments, calibration, uncertainties and access to data are provided in the Supplementary Information, SI-12 Atmospheric $N_2O$ Observation Networks.

The atmospheric burden and its rate of change during 1980−2020 were derived from mean maritime surface abundance (mole fraction) of $N_2O$ (Prather et al., 2023) with a conversion factor of 4.79 Tg N ppb$^{-1}$ (Prather, et al., 2012). Combining uncertainties in measuring the mean surface mole fractions (Dlugokencky et al., 1994) and that of converting surface mole fractions to a global mean abundance (Prather et al., 2012), we estimate a ±1.4% uncertainty in the burden. Annual change in atmospheric abundance is calculated from the combined NOAA and AGAGE record of surface $N_2O$ and uncertainty on atmospheric abundance estimates is taken from the IPCC AR5.

### 2.5.2 Atmospheric loss rates and trends from stratospheric observations

The NASA Aura MLS satellite instrument has provided consistent global measurements of stratospheric $N_2O$, $O_3$ and temperature (T) since August 2004. These have been used with simple stratospheric chemistry models to calculate the monthly mean stratospheric loss of $N_2O$ due to photolysis and oxidation by $O(^1D)$ (Prather et al., 2015; 2023; Minschwaner et al. 1998). Tropospheric chemical loss also occurs, but at a very low rate (<1% of the total) and is thus not included in the calculations.



### 2.5.3 Atmospheric inversion estimates of N₂O emissions and losses

For the TD constraints on both land and ocean $N_2O$ fluxes for the period 1998−2020, we used estimates from four independent
atmospheric inversion frameworks (INVICAT, PyVAR-CAMS, MIROC4-ACTM, and GEOS-Chem), all of which used a
Bayesian inversion method (see supplementary information for details on the inversion frameworks).

The inversion frameworks INVICAT and PyVAR-CAMS used the transport models TOMCAT and LMDz5, respectively,
which were both driven by ECMWF ERA5 meteorology, while MIROC4-ACTM used the transport model ACTM, which as
driven by JRA-55 meteorology, and GEOS-Chem used the transport model of the same name, which was driven by MERRA-
2 meteorology. All inversion frameworks assumed that the prior distribution of emissions followed a normal distribution, with
the multivariate mean taken from different models and data products, with standard deviations detailed in the supplement.
Specifically, GEOS-Chem, INVICAT and PyVAR-CAMS built prior flux distributions for natural soil emissions from the
terrestrial biospheric model O-CN (Zaehle et al., 2011) and for biomass burning emissions from GFED-v4s (van der Werf et
al., 2017). For anthropogenic emissions from agricultural and non-agricultural sectors (excluding biomass burning), estimates
from EDGAR v5 were used to build the prior for the period 2005-2020 (since these estimates were only available up to 2015,
the emissions for 2016-2020 were estimated based on those of the year 2015) and for the period 1997-2004, the estimates from
EDGAR-v4.32 were used. On the other hand, MIROC4-ACTM used the estimate from the terrestrial biospheric model VISIT
for natural soils emissions and EDGAR v4.2 estimates for all anthropogenic emissions.

The inversion frameworks used atmospheric observations from ground-based networks, specifically NOAA, AGAGE and
CSIRO (see supplementary information for details).

The atmospheric transport models also calculate the loss of $N_2O$ in the stratosphere by photolysis and oxidation by $O(^1D)$
radicals (Minschwaner et al. 1998). The TD mean posterior estimates for the 18 land regions were calculated by integrating
the gridded fluxes at $1° \times 1°$ over each region (the fluxes were interpolated from the original model resolution to $1° \times 1°$).

## 3. Results

### 3.1 Trends in atmospheric mole fractions and implied emissions

### 3.1.1 Trends in atmospheric N₂O mole fractions

The three observation networks AGAGE, NOAA and CSIRO show consistent growth in atmospheric $N_2O$ mole fractions from
315.8 (315.5-316.2) ppb in 2000 to 335.9 (335.6-336.1) ppb in 2022. The mean annual growth rate increased from 0.76 (0.55-
0.95) ppb $yr^{-1}$ in the 2000s to 0.96 (0.79-1.15) ppb $yr^{-1}$ in 2010s with significant seasonal and interannual variations. In 2020
and 2021, the $N_2O$ atmospheric growth rate was 1.33 ppb $yr^{-1}$ and 1.38 ppb $yr^{-1}$, respectively, both higher than any previous
observed year (since 1980), and was more than 30% higher than the average value in the decade of the 2010s (Figure 2). As is
shown in Figure 5, the observed $N_2O$ mole fraction in 2020 (mean: 333.2, 332.7-333.5 ppb) has exceeded predicted levels
across the four illustrative Representative Concentration Pathways (RCPs) (329.2-331.5 ppb) used in CMIP5 (Meinshausen et

al. 2011) and the seven illustrative Socioeconomic Pathways (SSPs) (330.5-331.9 ppb) used in CMIP6 (Meinshausen et al.
560    2020).

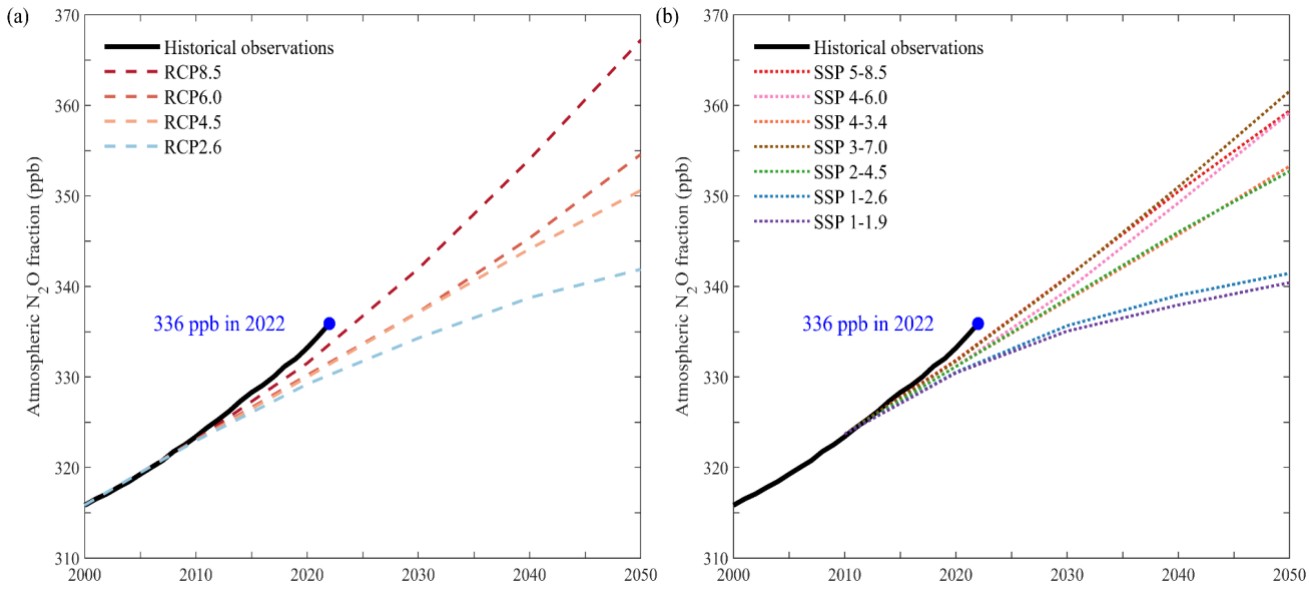

**Figure 5. Comparison between measured global N$_2$O mole fractions from the three GHG observing networks and the projected mole fractions from (a) the four illustrative Representative Concentration Pathways (RCPs) in the IPCC**
**Fifth Assessment Report, and (b) the seven illustrative Socioeconomic Pathways (SSPs) used in CMIP6.**

### 3.2 N$_2$O sources and sinks: BU estimates

### 3.2.1 Anthropogenic sources

### 3.2.1.1 Global anthropogenic emissions during 1980-2020

Global total anthropogenic emissions increased in the last four decades, from 4.8 (3.1-7.3) TgN yr$^{-1}$ in 1980 to 6.7 (3.3-10.9) TgN yr$^{-1}$ in 2020, with large uncertainties (Figure 6). Among all anthropogenic sources, direct emissions from nitrogen additions in the agricultural sector made the largest contribution to the increase, which grew from 2.2 (1.6-2.8) TgN yr$^{-1}$ in 1980 to 3.9 (2.9-5.1) TgN yr$^{-1}$ in 2020. Indirect N$_2$O emissions also steadily increased during the study period, from 0.9 (0.7-1.1) TgN yr$^{-1}$ in 1980 to 1.3 (0.9-1.6) TgN yr$^{-1}$ in 2020. In contrast, other direct anthropogenic emissions did not have a trend,
and the total amount fluctuated around 2.1 TgN yr$^{-1}$. Perturbed fluxes from climate/CO$_2$/land cover change led to a small increase in N$_2$O sink, from -0.4 (-0.9-1.0) TgN yr$^{-1}$ in 1980 to -0.6 (-2.2-1.8) TgN yr$^{-1}$ in 2020.

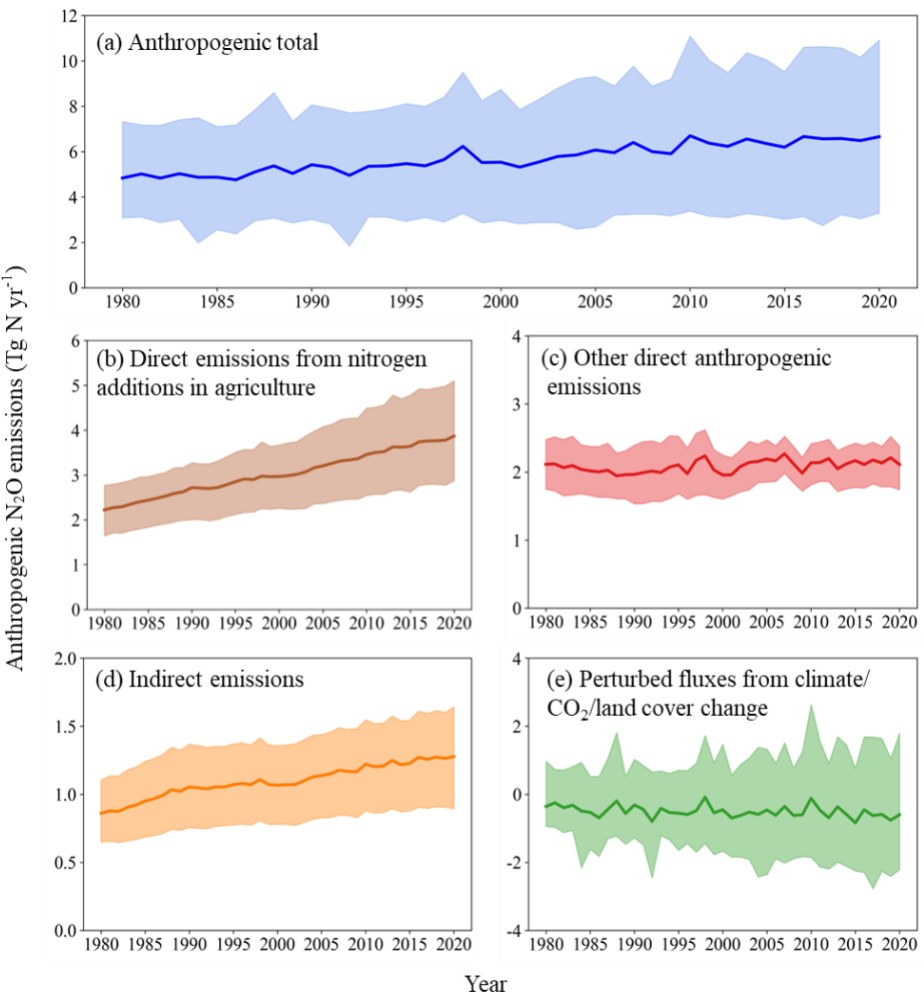

**Figure 6. Changes in global anthropogenic N₂O emissions (a) and N₂O emissions from different sectors (b-e) during**
**1980-2020.**

### 3.2.1.2 Direct emissions from nitrogen additions in the agricultural sector (Agriculture)

In the past four decades, N₂O emissions from all the four sources within the agricultural sector significantly increased (Figure

7), with the largest contribution from direct soil emissions (from 1.1 TgN yr⁻¹ in 1980 to 2.1 TgN yr⁻¹ in 2020), followed by

manure left on pasture (from 0.9 TgN yr⁻¹ in 1980 to 1.4 TgN yr⁻¹ in 2020), aquaculture (from 0.01 TgN yr⁻¹ in 1980 to 0.12

TgN yr⁻¹ in 2020), and manure management (from 0.24 TgN yr⁻¹ in 1980 to 0.26 TgN yr⁻¹ in 2020).

Direct soil emissions accounted for the largest proportion of emissions from the agriculture sector. All four estimates show a

steady increase in direct soil emissions since 1980 (Figure 7a). Among them, NMIP2/DLEM exhibited the largest magnitude

and the fastest increase rate, from 1.1 TgN yr⁻¹ in 1980 to 2.6 TgN yr⁻¹ in 2020. By contrast, SRNM/DLEM suggested the





slowest increase rate, from 1.0 TgN yr$^{-1}$ in 1980 to 1.7 Tg yr$^{-1}$ in 2020. The estimates of the two inventories (FAOSTAT and EDGARv7.0) exhibited similar magnitudes and trends, especially after 1990. All three estimates suggested a significant increasing trend for N$_2$O emissions from manure left on pasture over the period 1980-2020. Although all methods showed an increasing trend, they had significant differences in magnitude and increase rate (Figure 7b). FAOSTAT showed the largest

magnitude and increase rate, from 1.2 TgN yr$^{-1}$ in 1980 to 1.9 TgN yr$^{-1}$ in 2020. However, DLEM showed a smaller magnitude and a slower increase rate, from 0.5 TgN yr$^{-1}$ in 1980 to 0.9 TgN yr$^{-1}$ in 2020. Although the two inventory estimates for emissions from manure management showed similar temporal variations, FAOSTAT has a larger magnitude than EDGARv7.0 (Figure 7c). According to the IMAGE-GNM aquaculture nutrient budget model, N$_2$O emissions from aquaculture increased more than tenfold, from 0.01 TgN yr$^{-1}$ in 1980 to 0.12 TgN yr$^{-1}$ in 2020 (Figure 7d).

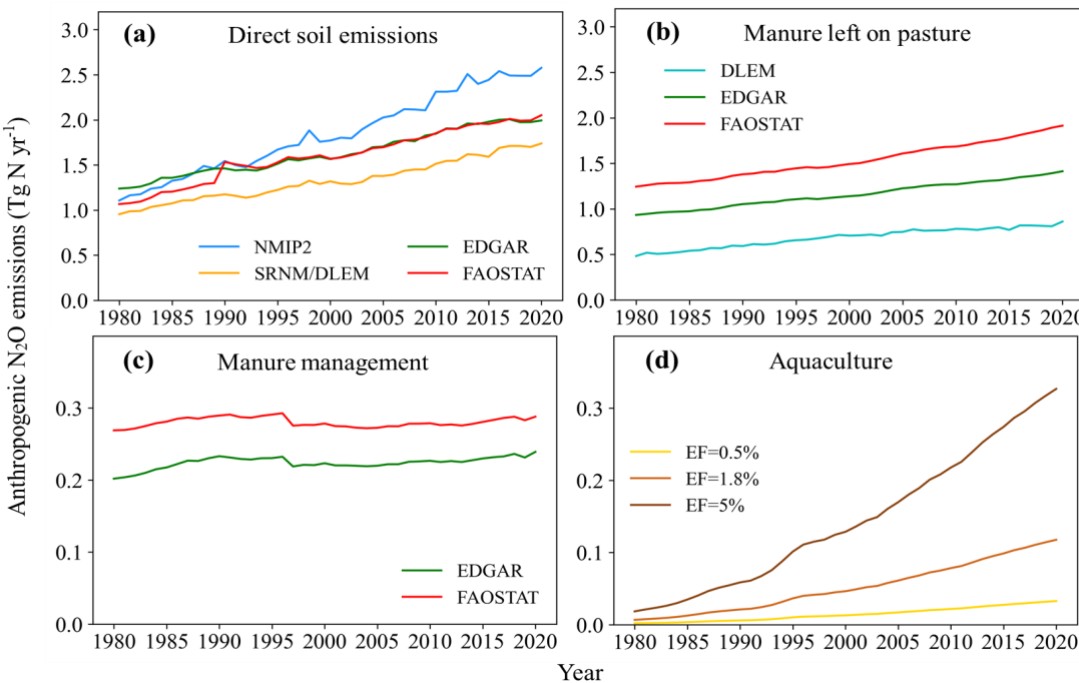


**Figure 7. Changes in global direct N$_2$O emissions from fertilizer and manure applied on agricultural soils (a), manure left on pasture (b), manure management (c), and aquaculture (d) during 1980-2020.**

### 3.2.1.3 Other direct anthropogenic sources

Fossil fuel and industry emissions accounted for the largest proportion of N$_2$O emissions from other direct anthropogenic sources. Estimates from two approaches showed different trends during their overlapping period: EGDARv7.0 had an increasing trend from 0.9 TgN yr$^{-1}$ in 1990 to 1.1 TgN yr$^{-1}$ in 2020, while EDGAR/UNFCCC did not show a trend with 1.0 TgN yr$^{-1}$ in 1990 and 1.0 TgN yr$^{-1}$ in 2020 (Figure 8a). These inventories, however, do not capture a strong increase in emissions from adipic acid production since 2010 (Davidson and Winiwarter, 2023). Both EDGARv7.0 and EDGAR/UNFCCC show a

steady and significant increase in N$_2$O emissions from waste and wastewater. Although EDGAR/UNFCCC shows a larger
magnitude than EGDARv7.0, these two inventory estimates show similar growth rates (Figure 8b). There are large
uncertainties in the magnitude and temporal trend of N$_2$O emissions from biomass burning (Figure 8c). DLEM and GFED
show a larger magnitude of emissions than FAOSTAT. Both DLEM and GFED have a decreasing trend over the overlapping
period of 1997-2020, however, FAOSTAT shows no significant trend during this period.


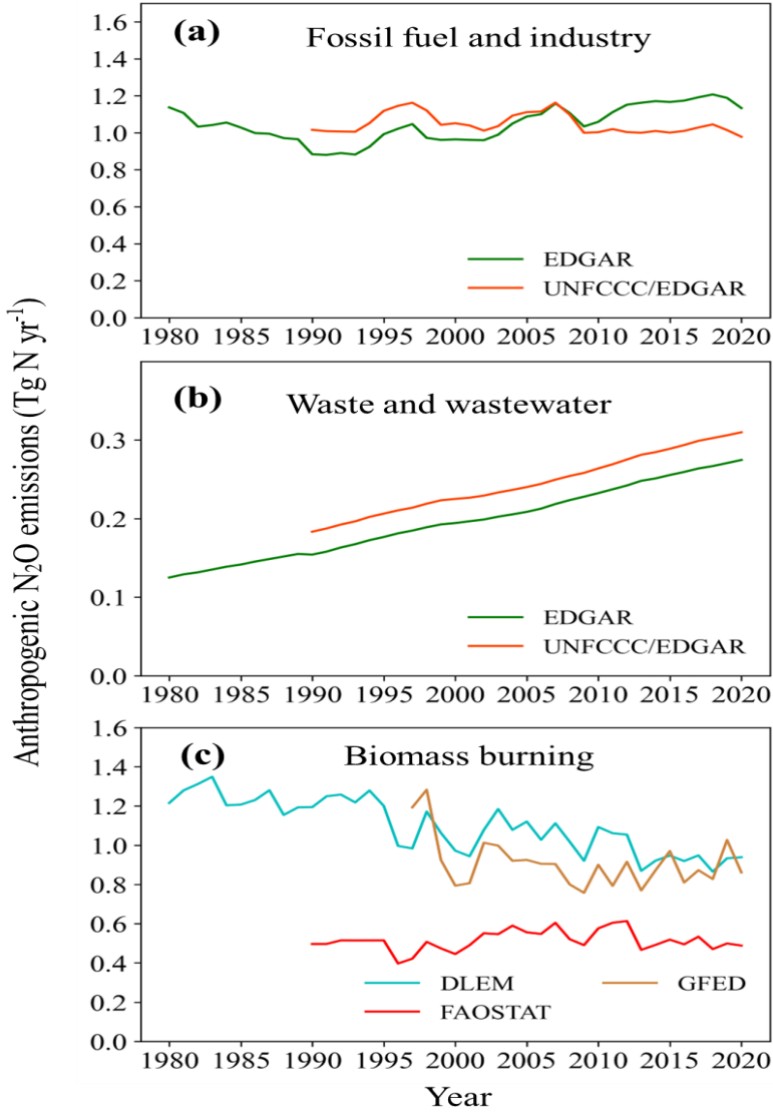

**Figure 8. Changes in N$_2$O emissions from other direct anthropogenic sources: fossil fuel (a), waste and wastewater (b), and biomass burning (c) during 1980-2020.**

**3.2.1.4 Indirect emissions from anthropogenic nitrogen additions**

Global anthropogenic $N_2O$ emissions from inland waters, estuaries and coastal vegetation continuously increased during 1980-2020 (Figure 9a). Although all methods revealed an overall increasing trend in emissions, process-based models show a much smaller magnitude and increase rate than the two inventories. According to meta-analysis and models, anthropogenic emissions from inland and coastal waters increased from 0.11 TgN yr$^{-1}$ in 1980 to 0.15 TgN yr$^{-1}$ in 2020. In contrast, EGDARv7.0 and

FAOSTAT showed emissions increased from 0.33 and 0.35 TgN yr$^{-1}$ in 1980 to 0.53 and 0.57 TgN yr$^{-1}$ in 2020, respectively. Emissions from N deposition on land also continued to increase during 1980-2020 (Figure 9b). NMIP2 and EDGAR/NMIP2 show emissions increasing from 0.6 and 0.4 TgN yr$^{-1}$ in 1980 to 0.9 and 0.6 TgN yr$^{-1}$ in 2020, respectively.

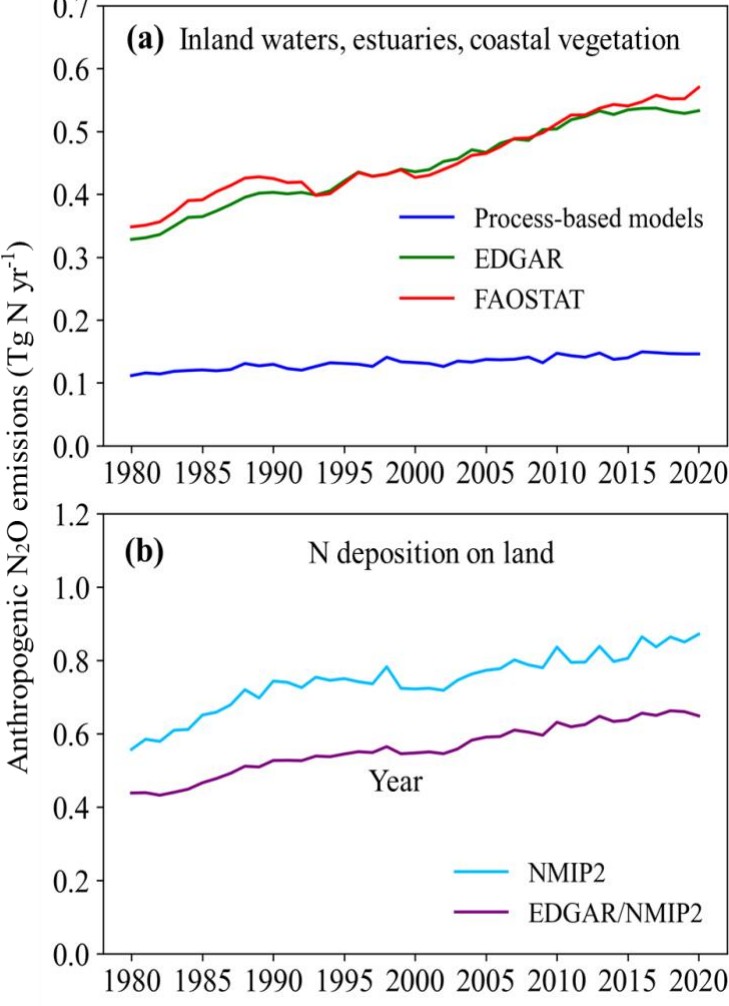

**Figure 9. Changes in indirect $N_2O$ emissions from anthropogenic nitrogen additions to inland waters (river, lake and reservoir), estuaries and coastal vegetation, and N deposition on land during 1980-2020.**





### 3.2.1.5 Perturbation fluxes from climate/CO₂/land cover change

Simulations with both DLEM and book-keeping approach suggested increasing uncertainties in post-deforestation pulse effect during 1980-2020. The post-deforestation pulse effect was 0.8 (0.6-1.1) Tg N yr$^{-1}$ in 1980 and 0.8 (0.4-1.3) Tg N yr$^{-1}$ in 2020

(Figure 10a). In contrast, DLEM and empirical approaches are comparable in terms of the magnitude and temporal changes in long-term reduction effect of deforestation, both approaches suggested a strong long-term reduction effect, which grew from -1.2 (-1.0, -1.4) Tg N yr$^{-1}$ in 1980 to -1.4 (-1.3, -1.6) Tg N yr$^{-1}$ in 2020 (Figure 10b). In general, deforestation had a negative effect on global soil $N_2O$ emissions. However, most NMIP2 models suggested a positive effect of climate change on soil $N_2O$ emissions, although with large uncertainty and significant interannual variations; this positive climate feedback significantly

increased during the past four decades (Figure 10c). In contrast to climatic effects, most NMIP2 models suggested a negative effect of rising atmospheric $CO_2$ concentration on soil $N_2O$ emissions through increasing nitrogen use efficiency and hence reducing soil N availability (Figure 10d). However, NMIP2 models have large discrepancies in the $CO_2$ fertilization effect on $N_2O$ emissions; ELM and ISAM suggested a positive effect, while all the other models suggest a negative effect.

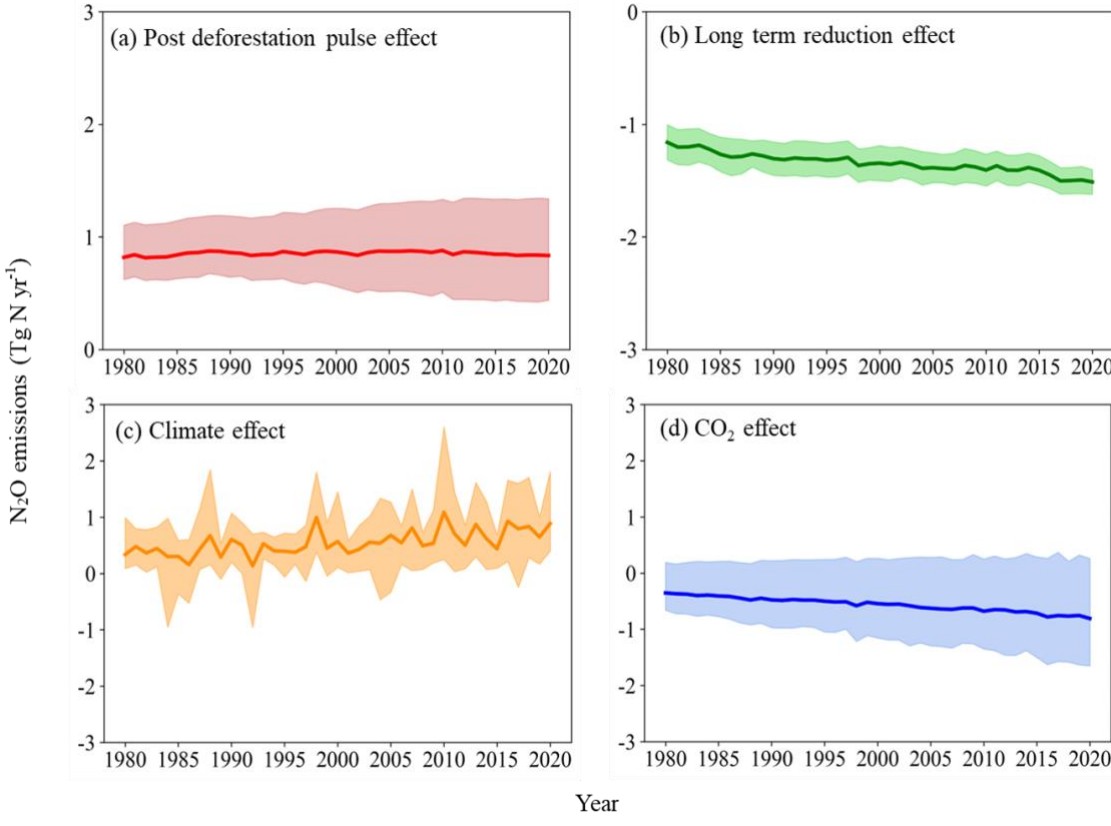

**Figure 10. Changes in perturbed $N_2O$ fluxes from changes in climate, $CO_2$, and land cover during 1980-2020.**



### 3.2.2 Natural N$_2$O sources

Global total N$_2$O emissions from all-natural sources, including natural emissions from soils, open oceans, shelves, lightning, inland waters, estuaries, and coastal vegetation, natural soils, and open oceans remained relatively steady throughout the study period 1980-2020. The mean value of global natural N$_2$O emissions fluctuated between 11.5-11.9 TgN yr$^{-1}$, with an average of 11.7 TgN yr$^{-1}$. Global natural N$_2$O emissions also have a large uncertainty, with the maximum estimates (15.8-16.6 TgN yr$^{-1}$) roughly double the minimum estimates (7.0-8.0 TgN yr$^{-1}$).

### 3.2.2.1 Natural soil N$_2$O emission baseline

The Natural soil N$_2$O emission baseline represents the preindustrial soil N$_2$O emissions derived from NMIP2 simulations, driven by potential vegetation/land cover and other environmental factors in the pre-industrial period (1850). Global natural soil N$_2$O emissions are estimated to be 6.4 TgN yr$^{-1}$, and account for 55% of the total natural emissions. However, N$_2$O emissions from natural soils estimated by the NMIP2 showed large divergences among eight models. Among the NMIP2 models, ELM had the highest estimate with an average of 8.6 TgN yr$^{-1}$, which was more than double the estimate from the CLASSIC model (3.9 TgN yr$^{-1}$).

### 3.2.2.2 Natural N$_2$O emission baseline from open ocean and continental shelves

We also estimated N$_2$O emissions from the open oceans and continental shelves. Open ocean is the second largest source of natural N$_2$O emissions with a global mean value fluctuating between 3.4 and 3.8 TgN yr$^{-1}$ during 1980-2020. Open ocean N$_2$O emissions were estimated by four ocean models. Among these models, NEMOv3.6-PISCESv2-gas had the highest estimate, with an average of 4.6 TgN yr$^{-1}$, while NEMO-PlankTOM10 had the lowest estimate with an average of 2.8 TgN yr$^{-1}$. The four ocean models show different trends in open ocean emissions. NEMOv3.6-PISCESv2-gas shows a slight increasing trend, while the other three models show consistent decreasing trends. In addition to open oceans, shelves are an important source of N$_2$O emissions, which was not quantified in the previous global N$_2$O budget (Tian et al., 2020). Global shelf N$_2$O emissions were estimated by two high-resolution models (CNRM and ECCO) and one data product (MEM-RF). The average of the three estimates is 1.2 TgN yr$^{-1}$, ranging from 0.6 TgN yr$^{-1}$ (ECCO) to 1.6 TgN yr$^{-1}$ (MEM-RF).

### 3.2.2.3 Natural N$_2$O emission from inland waters, estuaries & coastal vegetation

Natural N$_2$O emissions from inland waters and estuaries were much smaller than emissions from the soils, oceans and shelves. It has an average value of 0.08 TgN yr$^{-1}$, ranging from 0.05 TgN yr$^{-1}$ to 0.14 TgN yr$^{-1}$. Rivers are the largest source emitting 0.04 (0.01-0.08) TgN yr$^{-1}$ of N$_2$O, and account for 48% of the natural emissions from inland waters and estuaries. The global natural N$_2$O emissions from lakes and estuaries were 0.02 (0.01-0.03) TgN yr$^{-1}$ and 0.02 (0.02-0.03) TgN yr$^{-1}$, respectively.

#### 3.2.2.4 Lightning, atmospheric production and natural sinks

The source of reactive N from lightning, and its contribution to $N_2O$, and the direct production of $N_2O$ from $NH_3$ in the atmosphere are relatively small, and we have no new estimates in this work. However, synthesizing the available estimates in
the scientific literature, we estimate lightning to contribute 0.05 (0.02-0.09) TgN $yr^{-1}$ (median and range) (Nault et al. 2017; Schumann and Huntrieser et al. 2007) and atmospheric production to contribute 0.5 (0.3-1.1) TgN $yr^{-1}$ (Kolhmann and Poppe, 1999; Dentener and Crutzen, 1994).

Similarly, the surface sink of $N_2O$ is small and we do not produce a new estimate in this budget but only synthesize available estimates from the literature. We estimate the global surface sink to be 0.01 (0.0 – 0.3) TgN $yr^{-1}$.

### 3.3 $N_2O$ sources and sinks: TD estimates

#### 3.3.1 TD total source

Ensemble estimates across the four atmospheric inversions show that the long-term average global $N_2O$ emissions during 1997-2020 was 16.6 TgN $yr^{-1}$ (minimum: 15.5 TgN $yr^{-1}$; maximum: 18.2 TgN $yr^{-1}$). All four inversions show a significant increasing trend in global $N_2O$ emissions ($p<0.05$) with a mean rate of increase of 0.10 TgN $yr^{-2}$ (0.08 - 0.12 TgN $yr^{-2}$) (Figure
11a).

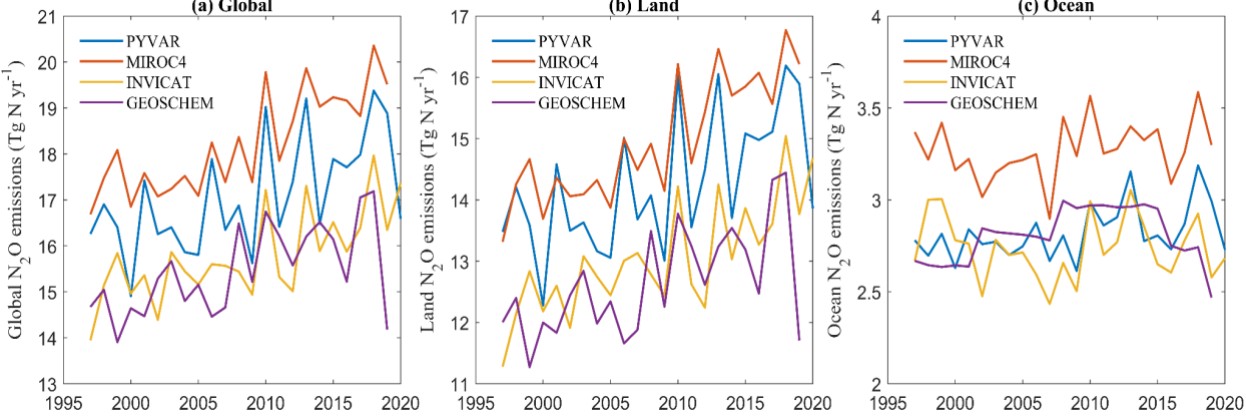

**Figure 11. Annual global $N_2O$ emissions during 1997-2020 estimated by four atmospheric inversions (TD model). (a) Total global emission, (b) Land emission and (c) ocean emission.**

#### 3.3.1.1 TD land emission

The estimates derived from the four inversions show that the land-based emission is the dominant source of $N_2O$ emissions, over ocean sources, and the long-term average land $N_2O$ emission during 1997-2020 was 13.7 TgN $yr^{-1}$ (minimum: 12.6 TgN $yr^{-1}$; maximum: 15.0 Tg N $yr^{-1}$), contributing 80-85% of the global $N_2O$ emissions. Land sources dominated the interannual





variability of global N₂O emissions and the trend (Figure 11b). All TD models suggested a significant increasing trend in land
N₂O emissions during the study period 1997-2020 ($p<0.05$), with increase rate ranging from 0.09 TgN yr⁻¹ to 0.13 TgN yr⁻¹.

### 3.3.1.2 TD ocean emission

The magnitude of N₂O emissions from oceans is much smaller than that from land (Figure 11c). The mean ocean N₂O emission during 1997-2020 derived from four inversion models was 2.9 TgN yr⁻¹, ranging from a minimum of 2.7 TgN yr⁻¹ to a maximum of 3.3 TgN yr⁻¹. The estimates of MIROC4 were much higher than the estimates of other models. The four inversions
show divergent interannual variability, and none suggested a significant trend. The TD estimates on ocean N₂O emission is much smaller than that estimated by four ocean biogeochemical models, with a global mean value fluctuating between 3.4 and 3.8 TgN yr⁻¹ during 1980-2020.

### 3.3.2 TD stratospheric sink

The four inversions have comparable magnitudes of global stratospheric N₂O sink (via photolysis and oxidation by the
electronically excited atomic oxygen, O(¹D), in the stratosphere), with an average value of 12.4 TgN yr⁻¹ (min, max of 12.2, 12.7 TgN yr⁻¹) for 2000-2020 (Figure 12). All four inversions found that the global stratospheric N₂O sink increased during 1997-2020 (Figure 13) in proportion to the growing atmospheric N₂O abundance, with an average rate of increase of 0.05 TgN yr⁻² (0.03 - 0.07 TgN yr⁻²). Differences among the estimates decreased after 2000 likely due to improvements in observation coverage and accuracy, but possibly also due to decreasing influence of the initial mixing ratio fields, which differed among
the inversion frameworks. Although the inversions show comparable trends in the sink, they differ in their inter-annual variability.

We also provide an independent estimate for the stratospheric sink based on satellite observations and a photolysis model. This estimate likewise showed that the sink increased, from 12.8 Tg N yr⁻¹ in the 1990s to 14.0 Tg N yr⁻¹ in the 2010s (Table 2), with higher annual loss rates than estimated by the inversions, and an average loss of 13.4 TgN yr⁻¹ for 2005-2021. This
estimate also showed large quasi-biennial interannual variability with an amplitude of 7 %. More interestingly, over this time period the abundance of N₂O in the middle stratosphere, where the greatest loss of N₂O occurs, was increasing at a rate of 5.0+-1.2 %/decade, which is faster than the increase in the tropospheric abundance of 2.9+-0.0 %/decade. This resulted in a greater loss of N₂O (i.e., more than proportionate to the mean atmospheric increase) and thus a decrease of the mean atmospheric lifetime (burden divided by loss) of 2.1 ± 0.7% per decade, from 119.3 years in the 2000s to 117 years in the
2010s (Prather et al. 2023, also see Table 2). These changes are thought to be a result of an increase in the intensity of Brewer-Dobson Circulation (BDC), which would transport N₂O more rapidly from the troposphere into the mid-stratosphere. An increase in the intensity of BDC is predicted by climate models (Oberlander-Hayn et al. 2016). However, we note that none of the atmospheric inversions found a significant trend in the atmospheric lifetime (although the total loss increased, Figure 12) and more research is needed to identify why there is this discrepancy.




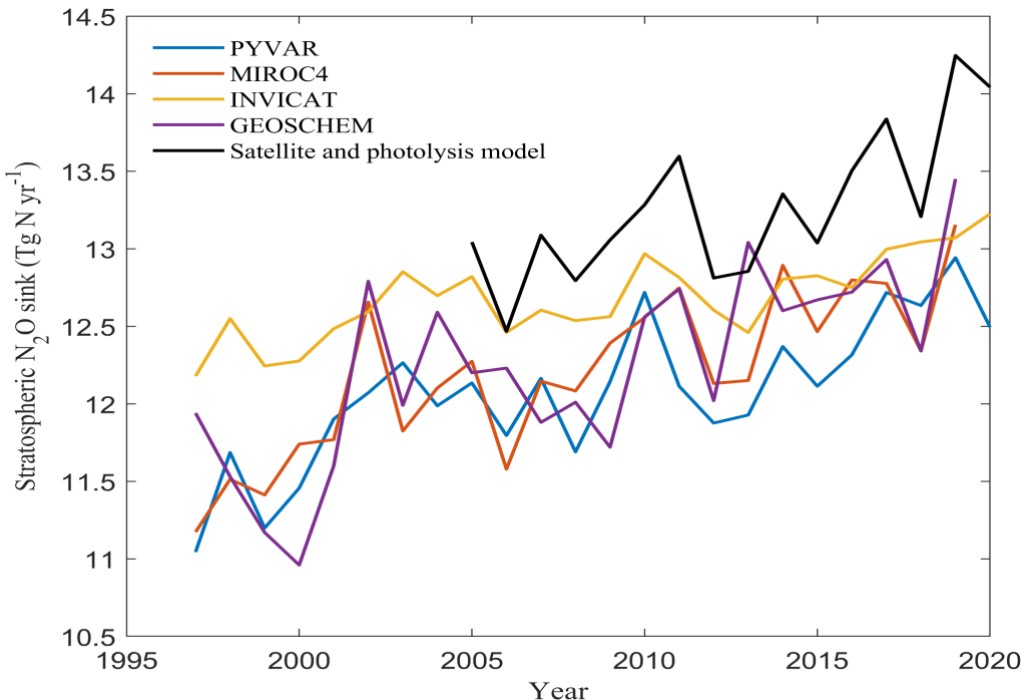

**Figure 12. Global stratospheric N₂O sink estimated by atmospheric inversions, satellite and photolysis model during 1997-2020.**

### 3.4 Decadal patterns and trend of the global N₂O budget: Comparisons between BU and TD approaches

BU approaches provide estimates of N₂O fluxes for the identified sources and sinks during 1980-2020, while TD approaches only provide the total net flux during 1997-2020. In the following analyses of the decadal global N₂O budget, the comparison between BU and TD approaches is only for total N₂O estimates. We rely on BU approaches to quantify all identified sources and sinks (Table 2, Figure 1).

### 3.4.1 Global N₂O budget in recent decade (2010-2019)

The BU and TD approaches give remarkably consistent estimates of global total N₂O emissions in the 2010s, with values of 18.1 (minimum–maximum: 10.4–25.9) Tg N yr$^{-1}$ and 17.4 (minimum–maximum: 15.8–19.2) Tg N yr$^{-1}$ (Fig 1, Table 2), respectively. However, the BU estimate shows a large uncertainty range in part because of the spread of estimates from process-based models. TD approaches estimate that the stratospheric sink (i.e., N₂O losses via photolysis and reaction with O($^1$D) in the stratosphere) for the 2010s was 12.6 (12.3 - 12.9) Tg N yr$^{-1}$. However, the atmospheric sink estimate based on satellite

observations and a photolysis model for the 2010s was 13.4 (12.3 - 14.5) Tg N yr$^{-1}$. The imbalance of sources and sinks of N₂O derived from the averaged BU and TD estimates is 4.7 Tg N yr$^{-1}$. This imbalance agrees well
with the observed increase in atmospheric abundance of N₂O between 2010 and 2019 of 4.6 (4.5–4.7) Tg N yr$^{-1}$. Based on the BU-based estimates, natural sources contributed 64% of total emissions (mean: 11.6; min–max: 7.2–15.9 Tg N yr$^{-1}$) during



this period. Specifically, the natural soil flux contributed the most, with the decadal mean of 6.4 (3.9–8.6) Tg N yr$^{-1}$, followed
by the open ocean emissions (mean: 3.5, 2.5–4.7 Tg N yr$^{-1}$), shelf emissions (mean: 1.2, 0.6–1.6 Tg N yr$^{-1}$), lightning and
atmospheric production (mean: 0.4, 0.2–1.2 Tg N yr$^{-1}$), and natural emissions from inland waters and estuaries (mean: 0.1,
0.0–0.1 Tg N yr$^{-1}$) (Figure 1).

Anthropogenic sources contributed, on average, 36% to the total $N_2O$ emissions (mean: 6.5; minimum–maximum: 3.2–10.0
Tg N yr$^{-1}$) in the 2010s. Direct emissions from nitrogen additions in agriculture were 3.6 (2.7–4.8) Tg N yr$^{-1}$, contributing to
56% of the total anthropogenic emissions (Table 2). Emissions from other direct anthropogenic sources made the second
largest contribution, with a decadal mean of 2.1 (1.8–2.4) Tg N yr$^{-1}$. Indirect emissions from anthropogenic nitrogen additions
contributed to 19% of the total anthropogenic emissions, with a decadal mean of 1.2 (0.9–1.6) Tg N yr$^{-1}$. Changes in climate,
$CO_2$ and land cover had an overall negative effect on $N_2O$ emissions (mean: -0.6, -2.1–1.2 Tg N yr$^{-1}$), mainly because of the
negative effects of reduced mature forest area (mean: -1.4, -1.6– -1.3 Tg N yr$^{-1}$) and increasing $CO_2$ concentration (mean: -
0.7, -1.5–0.3 Tg N yr$^{-1}$).

**3.4.2 Decadal trend of the global $N_2O$ budget**

Global $N_2O$ emissions estimated by the BU and TD approaches were comparable in magnitude during the overlapping period
1997–2020, but TD estimates implied a larger inter-annual variability and a faster rate of increase (Figure 13a). BU and TD
approaches diverge when estimating the magnitude of land emissions compared with ocean emissions, although they are
consistent with respect to trends (Figure 13b). According to the BU approaches, global $N_2O$ emissions increased from 17.2 Tg
N yr$^{-1}$ (10.2-24.1 Tg N yr$^{-1}$) in 1997 to 18.3 Tg N yr$^{-1}$ (10.5-27.0 Tg N yr$^{-1}$) in 2020, with an average increase rate of 0.043
Tg N yr$^{-2}$ ($p<0.05$). In contrast, TD approaches suggested global emissions increased from 15.4 Tg N yr$^{-1}$ (13.9-16.7 Tg N
yr$^{-1}$) in 1997 to 17.0 Tg N yr$^{-1}$ (16.6-17.4 Tg N yr$^{-1}$) in 2020, implying a higher increase rate of 0.085 Tg N yr$^{-2}$ ($p<0.05$). The
BU estimate during 1997–2010 was on average 1.2 Tg N yr$^{-1}$ higher than the TD estimate. However, after 2010, the difference
in the magnitude of emissions between the two approaches is smaller, because of the rapid increase in the TD estimates. Since
the year 1980, BU approaches suggested a significant increase in global $N_2O$ emissions that was primarily driven by
anthropogenic sources (Table 2). Satellite and photolysis model estimate that the atmospheric $N_2O$ burden increased from 1528
Tg N in the 2000s to 1570 in the 2010s and 1592 Tg N in 2020, which is comparable to estimates by atmospheric chemistry
transport models, showing an increase in atmospheric $N_2O$ burden from 1527 (1504-1545) Tg N in the 2000s to 1606 (1592-
1621) Tg N in 2020.

Earth System
Science
Data

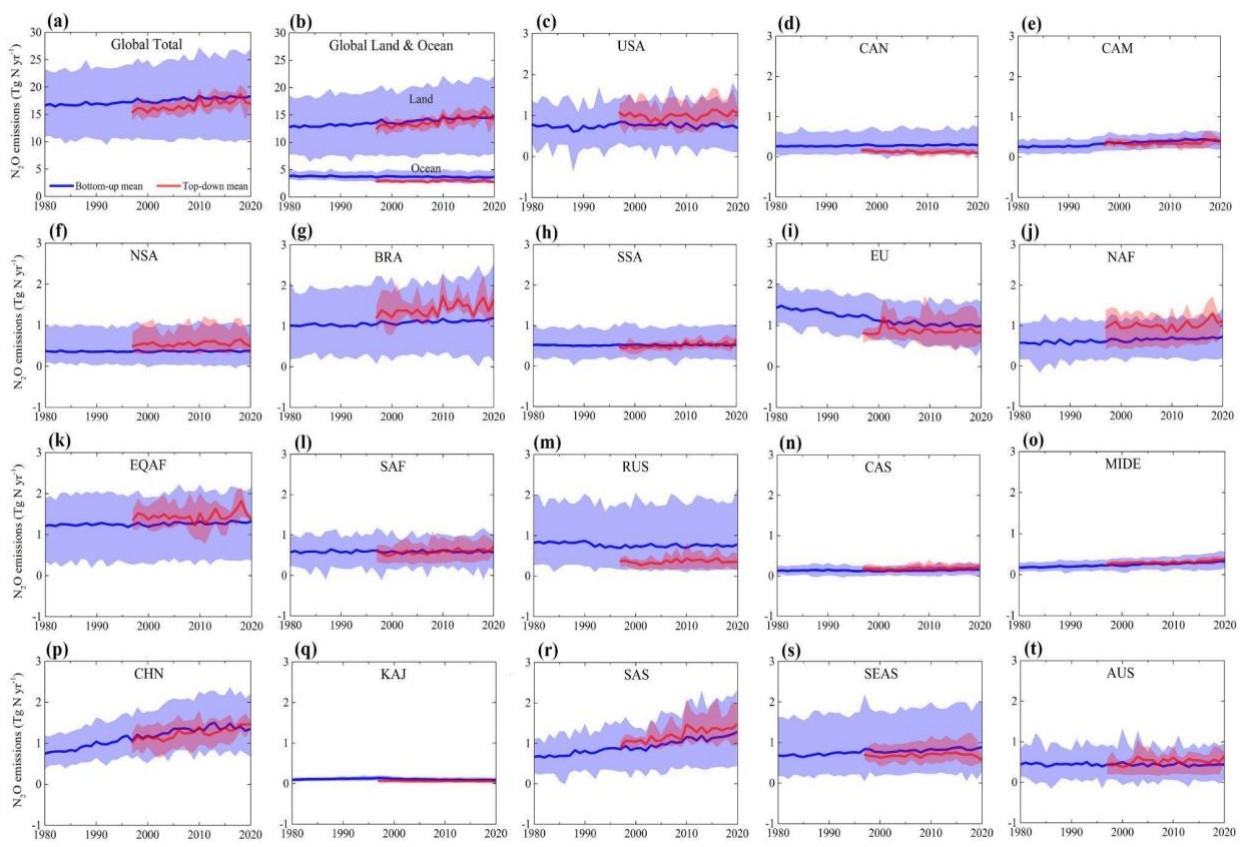

**Figure 13. Comparison of global and regional N₂O emissions estimated by BU and TD approaches. The 18 regions include United States (USA), Canada (CAN), Central America (CAM), Northern South America (NSA), Brazil (BRA), Southwest South America (SSA), Europe (EU), Northern Africa (NAF), Equatorial Africa (EQAF), Southern Africa (SAF), Russia (RUS), Central Asia (CAS), Middle East (MIDE), China (CHN), Korea and Japan (KAJ), South Asia (SAS), Southeast Asia (SEAS), and Australasia (AUS).**

**Table 2. The global N₂O budget for the decades of the1980s, 1990s, 2000s, 2010s, and year 2020** (Tg N yr$^{-1}$)

| | | 1980-1989 | 1990-1999 | 2000-2009 | 2010-2019 | 2020 |
|---|---|---|---|---|---|---|
| **Anthropogenic sources (BU)** | | Mean (Min, Max) | Mean (Min, Max) | Mean (Min, Max) | Mean (Min, Max) | Mean (Min, Max) |
| Agricultural | Direct soil emissions | 1.2 (1.1, 1.3) | 1.5 (1.2, 1.6) | 1.7 (1.4, 2.0) | 2.0 (1.6, 2.4) | 2.1 (1.7, 2.6) |
| | Manure left on pasture | 0.9 (0.5, 1.3) | 1.1 (0.6, 1.4) | 1.2 (0.7, 1.6) | 1.3 (0.8, 1.8) | 1.4 (0.9, 1.9) |
| | Manure management | 0.2 (0.2, 0.3) | 0.3 (0.2, 0.3) | 0.2 (0.2, 0.3) | 0.3 (0.2, 0.3) | 0.3 (0.2, 0.3) |
| | Aquaculture | 0.0 (0.0, 0.0) | 0.0 (0.0, 0.1) | 0.1 (0.0, 0.2) | 0.1 (0.0, 0.3) | 0.1 (0.0, 0.3) |
| | Subtotal | 2.4 (1.8, 3.0) | 2.8 (2.1, 3.4) | 3.2 (2.3, 4.0) | 3.6 (2.7, 4.8) | 3.9 (2.9, 5.1) |



| Other direct anthropogenic sources | Fossil fuels and industry | 1.0 (1.0, 1.0) | 1.0 (0.9, 1.1) | 1.1 (1.0, 1.1) | 1.1 (1.0, 1.2) | 1.1 (1.0, 1.1) |
|---|---|---|---|---|---|---|
| | Waste and wastewater | 0.1 (0.1, 0.1) | 0.2 (0.2, 0.2) | 0.2 (0.2, 0.2) | 0.3 (0.3, 0.3) | 0.3 (0.3, 0.3) |
| | Biomass burning | 0.9 (0.5, 1.2) | 0.9 (0.5, 1.2) | 0.8 (0.5, 1.0) | 0.8 (0.5, 1.0) | 0.8 (0.5, 0.9) |
| | Subtotal | 2.0 (1.7, 2.4) | 2.1 (1.6, 2.4) | 2.1 (1.8, 2.4) | 2.1 (1.8, 2.4) | 2.1 (1.7, 2.4) |
| Indirect emissions from anthropogenic nitrogen additions | Inland waters, estuaries, coastal vegetation | 0.3 (0.1, 0.4) | 0.3 (0.1, 0.4) | 0.4 (0.1, 0.5) | 0.4 (0.1, 0.5) | 0.4 (0.1, 0.6) |
| | Atmospheric nitrogen deposition on land | 0.5 (0.5, 0.6) | 0.6 (0.5, 0.7) | 0.7 (0.6, 0.8) | 0.7 (0.6, 0.8) | 0.8 (0.6, 0.9) |
| | Atmospheric nitrogen deposition on ocean | 0.1 (0.1, 0.2) | 0.1 (0.1, 0.2) | 0.1 (0.1, 0.2) | 0.1 (0.1, 0.2) | 0.1 (0.1, 0.2) |
| | Subtotal | 0.9 (0.7, 1.2) | 1.0 (0.8, 1.4) | 1.2 (0.8, 1.4) | 1.2 (0.9, 1.6) | 1.3 (0.9, 1.6) |
| Perturbed fluxes from climate/CO2/land cover change | CO2 effect | -0.4 (-0.8, 0.2) | -0.5 (-1.0, 0.2) | -0.6 (-1.3, 0.3) | -0.7 (-1.5, 0.3) | -0.8 (-1.6, 0.3) |
| | Climate effect | 0.4 (0.1, 0.8) | 0.5 (0.2, 0.7) | 0.6 (0.1, 0.8) | 0.7 (0.2, 1.2) | 0.9 (0.4, 1.8) |
| | Post-deforestation pulse effect | 0.8 (0.6, 1.1) | 0.9 (0.6, 1.2) | 0.9 (0.5, 1.3) | 0.9 (0.4, 1.3) | 0.8 (0.4, 1.3) |
| | Long-term effect of reduced mature forest area | -1.2 (-1.1, -1.4) | -1.3 (-1.2, -1.5) | -1.4 (-1.2, -1.5) | -1.4 (-1.3, -1.6) | -1.5 (-1.4, -1.6) |
| | Subtotal | -0.4 (-1.1, 0.7) | -0.5 (-1.4, 0.6) | -0.6 (-1.9, 0.8) | -0.6 (-2.1, 1.2) | -0.6 (-2.2, 1.8) |
| **Anthropogenic total** | | 5.0 (3.0, 7.3) | 5.5 (3.1, 7.9) | 5.8 (3.1, 8.6) | 6.5 (3.2, 10.0) | 6.7 (3.3, 10.9) |
| **Natural fluxes (BU)** | | | | | | |
| Natural soils baseline | | 6.4 (3.9, 8.5) | 6.4 (3.8, 8.6) | 6.4 (3.9, 8.5) | 6.4 (3.9, 8.6) | 6.4 (3.8, 8.7) |
| Open ocean baseline | | 3.7 (3.0, 4.6) | 3.6 (2.8, 4.5) | 3.6 (2.7, 4.7) | 3.5 (2.5, 4.7) | 3.5 (2.5, 4.7) |
| Continental shelves | | 1.2 (0.6, 1.6) | 1.2 (0.6, 1.6) | 1.2 (0.6, 1.6) | 1.2 (0.6, 1.6) | 1.2 (0.6, 1.6) |
| Natural (inland waters, estuaries, coastal vegetation) | | 0.1 (0.0, 0.1) | 0.1 (0.0, 0.1) | 0.1 (0.0, 0.1) | 0.1 (0.0, 0.1) | 0.1 (0.0, 0.1) |
| Lightning and atmospheric production | | 0.4 (0.2, 1.2) | 0.4 (0.2, 1.2) | 0.4 (0.2, 1.2) | 0.4 (0.2, 1.2) | 0.4 (0.2, 1.2) |
| Surface sink (soils/wetlands) | | 0.0 (0.0, -0.3) | 0.0 (0.0, -0.3) | 0.0 (0.0, -0.3) | 0.0 (0.0, -0.3) | 0.0 (0.0, -0.3) |
| **Natural total** | | 11.8 (7.8, 15.8) | 11.7 (7.5, 15.8) | 11.7 (7.4, 15.9) | 11.6 (7.2, 15.9) | 11.7 (7.2, 16.1) |
| **BU Total Net Flux (source)** | | 16.8 (10.8, 23.1) | 17.2 (10.6, 23.7) | 17.5 (10.5, 24.5) | 18.1 (10.4, 25.9) | 18.3 (10.5, 27.0) |
| TD ocean | | | | 2.8 (2.6, 3.2) | 3.0 (2.7, 3.3) | 2.7 (2.7, 2.7) |
| TD land | | | | 13.2 (12.1, 14.3) | 14.5 (13.0, 15.9) | 14.3 (13.9, 14.7) |
| **TD Total Net Flux (source)** | | | | 16.0 (14.9, 17.5) | 17.4 (15.8, 19.2) | 17.0 (16.6, 17.4) |
| TD stratospheric sink | | | | 12.2 (11.7, 12.6) | 12.6 (12.3, 12.9) | 12.9 (12.5, 13.2) |





| | | | 12.8 (11.7, 13.8) | 13.4 (12.3, 14.5) | 14.0 (12.8, 15.2) |
|---|---|---|---|---|---|
| Atmospheric Chemical sink (a) | | | 12.8 (11.7, 13.8) | 13.4 (12.3, 14.5) | 14.0 (12.8, 15.2) |
| Change in atmospheric abundance (b) | | | 3.6 (3.6, 3.7) | 4.6 (4.5, 4.7) | 6.4 (6.2, 6.5) |
| Atmospheric burden | | | 1528 | 1570 | 1592 |
| Lifetime ('obs' from MLS) | | | 119.3 | 117 | |

*Notes: BU estimates include four categories of anthropogenic source and one category for natural sources and sinks. The sources and sinks of $N_2O$ are given in Tg N $yr^{-1}$. The atmospheric burden is given in Tg N. (a) Calculated from satellite observations with a photolysis model (about 1% of this sink occurs in the troposphere). (b) Calculated from the combined NOAA and AGAGE record of surface $N_2O$ and adopting the uncertainty of the IPCC Assessment Report 5 (Chapter 6)2. Detailed information on calculating each sub-category is shown in Supplementary Tables 1–13.*

### 3.5 Regional BU and TD estimates and their trends

To assess regional $N_2O$ budgets, we divide the global land into 18 regions as described in the method section. Our regional analyses include: 1) trends and variations of regional total $N_2O$ emissions from all sources derived from available estimates of TD (1997-2020) and BU (1980-2020) (Figure 13); 2) trends and variations of region anthropogenic $N_2O$ emissions from all identified sources during 1980-2020 derived from BU approach (Figure 14); and 3) Decadal regional $N_2O$ budget (2010-2019)
derived from both BU and TD approaches (Figure 15). The sections followed provide detailed estimates for each of the 18 regions.

#### 3.5.1 United States of America (USA)

For the USA, the TD estimates show higher total $N_2O$ emissions than the BU estimates over the period 1997-2020 (Figure 13c), with 1.00 (0.69–1.39) Tg N $yr^{-1}$ and 0.77 (0.19–1.42) Tg N $yr^{-1}$, respectively. Both approaches suggest that the total $N_2O$
emissions from the USA remained relatively stable during 1997-2020. Based on the BU estimates, changes in climate, $CO_2$, and land cover caused emission decline over 1980-2020, with the average of -0.30 Tg N $yr^{-1}$. The flux fluctuated between -0.39 Tg N $yr^{-1}$ and -0.20 Tg N $yr^{-1}$. Indirect emissions from anthropogenic nitrogen additions increased from 0.11 Tg N $yr^{-1}$ in 1980 to 0.13 Tg N $yr^{-1}$ in 1995 and then decreased to 0.10 Tg N $yr^{-1}$ in 2020. Direct emissions from nitrogen additions in agriculture increased from 0.25 Tg N $yr^{-1}$ in 1980 to 0.30 Tg N $yr^{-1}$ in 2020. However, the increase in direct agricultural
emissions was offset by the trend in emissions from other direct anthropogenic sources, which decreased from 0.26 Tg N $yr^{-1}$ in 1980 to 0.19 Tg N $yr^{-1}$ in 2020. The total anthropogenic $N_2O$ emissions slightly increased during 1980-2020, at the average rate of $1.3\times10^{-3}$ Tg N $yr^{-2}$. This increase primarily occurred during 1980-1997 (Figure 14).

In the 2010s, the BU estimates (0.75, 0.24–1.33 Tg N $yr^{-1}$) were on average 0.28 Tg N $yr^{-1}$ lower than the TD estimates (1.03, 0.71–1.45 Tg N $yr^{-1}$) (Figure 15). According to the BU results, natural sources contributed 52% of total emissions (0.39, 0.22–
0.65 Tg N $yr^{-1}$) during this period. Direct emissions from nitrogen additions in agriculture were 0.30 (0.18–0.38) Tg N $yr^{-1}$, contributing 39% of the total emissions. Emissions from other direct anthropogenic sources made the second largest contribution to anthropogenic emissions, with the decadal mean of 0.21 (0.18–0.23) Tg N $yr^{-1}$. Indirect emissions from anthropogenic nitrogen additions contributed 15% of the total anthropogenic emissions, with a decadal mean of 0.11 (0.07–





0.14) Tg N yr$^{-1}$. Changes in climate, $CO_2$ and land cover had an overall negative effect on $N_2O$ emissions with the mean value

of -0.25 Tg N yr$^{-1}$, ranging from -0.42 Tg N yr$^{-1}$ to -0.07 Tg N yr$^{-1}$. Recent study indicated that $N_2O$ emissions could be
increased by freeze-thaw cycles (Del Grosso et al. 2022) and tillage practices (Lu et al. 2022). Our BU estimates did not take
into consideration of freeze-thaw and tillage practice, which may have underestimated $N_2O$ emissions.

### 3.5.2 Canada (CAN)

BU approaches suggested a larger magnitude of total $N_2O$ emissions from Canada than TD approaches over the period 1997-

2020 (Figure 13d), with values of 0.29 (0.05–0.69) Tg N yr$^{-1}$ and 0.12 (0.06–0.19) Tg N yr$^{-1}$, respectively. BU and TD
estimates also showed divergent trends. TD estimates decreased at the rate of -1.5×10$^{-3}$ Tg N yr$^{-2}$, however, BU estimates
increased at the rate of 0.8×10$^{-3}$ Tg N yr$^{-2}$. According to the BU results, the increase in total $N_2O$ emissions from Canada was
mainly driven by the direct emissions from nitrogen additions in agriculture, which increased from 0.02 Tg N yr$^{-1}$ in 1980 to
0.05 Tg N yr$^{-1}$ in 2020. Perturbed fluxes from changes in climate, $CO_2$ and land cover showed an overall increase from 0.00

Tg N yr$^{-1}$ in 1980 to 0.02 Tg N yr$^{-1}$ in 2020. Indirect $N_2O$ emissions from Canada were relatively stable during the study
period, while emissions from other direct anthropogenic sources had large interannual variabilities (Figure 14).
In the 2010s, the BU estimates of Canada's total $N_2O$ emissions (0.29, 0.07–0.67 Tg N yr$^{-1}$) were over two times higher than
the TD estimates (0.12, 0.06–0.20 Tg N yr$^{-1}$) (Figure 15). According to the BU results, natural sources contributed 59% of
total emissions (0.17, 0.04–0.43 Tg N yr$^{-1}$) during this period. Direct emissions from nitrogen additions in agriculture were

0.05 (0.03–0.06) Tg N yr$^{-1}$, contributing to 15% of the total emissions. Emissions from other direct anthropogenic sources and
indirect emissions from anthropogenic nitrogen additions were 0.04 (0.02–0.08) Tg N yr$^{-1}$ and 0.02 (0.02–0.03) Tg N yr$^{-1}$,
respectively. Changes in climate, $CO_2$ and land cover had an overall positive effect on $N_2O$ emissions with the mean value of
0.01 Tg N yr$^{-1}$, ranging from -0.03 Tg N yr$^{-1}$ to 0.07 Tg N yr$^{-1}$.

### 3.5.3 Central America (CAM)

TD and BU estimates agreed well regarding the magnitudes and trends of $N_2O$ emissions from Central America (Figure 13e),
with mean values of 0.39 (0.20–0.59) Tg N yr$^{-1}$ and 0.35 (0.25–0.47) Tg N yr$^{-1}$ for BU and TD approaches, respectively.
During 1997-2020, the rate of increase of the BU estimates (4.2 ×10$^{-3}$ Tg N yr$^{-2}$) was higher than that of TD estimates (2.5
×10$^{-3}$ Tg N yr$^{-2}$). Emissions from other direct anthropogenic sources increased from 0.03 Tg N yr$^{-1}$ in 1980 to 0.15 Tg N yr$^{-1}$
in 2020 and were the major driver of the increase in $N_2O$ emissions from Central America. Direct agricultural emissions

increased during the study period, from 0.09 Tg N yr$^{-1}$ in 1980 to 0.11 Tg N yr$^{-1}$ in 2020. Indirect emissions and perturbed
fluxes from changes in climate, $CO_2$ and land cover were relatively stable during this period (Figure 14).
The BU and TD approaches gave comparable estimates of total $N_2O$ emissions from Central America in the 2010s, with values
of 0.42 (0.24–0.60) Tg N yr$^{-1}$ and 0.36 (0.24–0.48) Tg N yr$^{-1}$ for BU and TD approaches (Figure 15), respectively. Natural
sources contributed 40% of total emissions (mean: 0.17, 0.07–0.25 Tg N yr$^{-1}$) during this period. Emissions from other direct

anthropogenic sources contributed to 48% of the total emissions (mean: 0.18, 0.17–0.18 Tg N yr$^{-1}$). Direct and indirect





emissions were 0.11 (0.07–0.14) Tg N yr$^{-1}$ and 0.02 (0.02–0.03) Tg N yr$^{-1}$, respectively. Changes in climate, $CO_2$ and land cover had an overall negative effect on $N_2O$ emissions with the mean value of -0.05 Tg N yr$^{-1}$, ranging from -0.10 Tg N yr$^{-1}$ to 0.00 Tg N yr$^{-1}$.

### 3.5.4 Northern South America (NSA)

TD approaches suggested a larger magnitude of total $N_2O$ emissions from Northern South America than BU approaches over the period 1997-2020 (Figure 13f), with 0.55 (0.34–0.98) Tg N yr$^{-1}$ and 0.37 (0.03–1.00) Tg N yr$^{-1}$, respectively for each approach. During 1997-2020, the increase rate of the TD estimates (2.2 ×10$^{-3}$ Tg N yr$^{-2}$) was higher than that of BU estimates (0.5 ×10$^{-3}$ Tg N yr$^{-2}$). Direct agricultural emissions made the largest contribution to the increase in $N_2O$ emissions from Northern South America, increasing from 0.04 Tg N yr$^{-1}$ in 1980 to 0.07 Tg N yr$^{-1}$ in 2020 (Figure 14). $N_2O$ emissions from
the other three anthropogenic sectors did not have a significant trend during 1980-2020.

The BU estimates in the 2010s (0.37, 0.04–1.02 Tg N yr$^{-1}$) were on average 0.20 Tg N yr$^{-1}$ lower than the TD estimates (0.58, 0.35–1.06 Tg N yr$^{-1}$) (Figure 15). The average natural emission was 0.35 Tg N yr$^{-1}$ in the 2010s, contributing 93% of total emissions. Direct agricultural emissions, other direct emissions, and indirect emissions were 0.07 (0.05–0.09) Tg N yr$^{-1}$, 0.02 (0.01–0.02) Tg N yr$^{-1}$ and 0.01 (0.01–0.02) Tg N yr$^{-1}$, respectively. Changes in climate, $CO_2$ and land cover had an overall
negative effect on $N_2O$ emissions with the mean value of -0.07 Tg N yr$^{-1}$, ranging from -0.18 Tg N yr$^{-1}$ to 0.03 Tg N yr$^{-1}$.

### 3.5.5 Brazil (BRA)

The average total $N_2O$ emissions from Brazil estimated by BU approaches was 1.12 Tg N yr$^{-1}$, ranging from 0.25 Tg N yr$^{-1}$ to 2.16 Tg N yr$^{-1}$ (Figure 13g), which was lower than the TD estimates (mean: 1.42, 1.18–1.75 Tg N yr$^{-1}$). Both approaches detected a notable increasing trend in total $N_2O$ emissions during 1997-2020. TD approaches suggested a higher increase rate
(11.6 ×10$^{-3}$ Tg N yr$^{-2}$) than BU approaches (4.2 ×10$^{-3}$ Tg N yr$^{-2}$). Direct agricultural emissions, which increased from 0.13 Tg N yr$^{-1}$ in 1980 to 0.32 Tg N yr$^{-1}$ in 2020, made the largest contribution to the increase in $N_2O$ emissions from Brazil (Figure 14). Indirect emissions also show an increase from 0.03 Tg N yr$^{-1}$ in 1980 to 0.06 Tg N yr$^{-1}$ in 2020. Emissions from other anthropogenic sources and perturbed fluxes from changes in climate, $CO_2$, and land cover did not have an obvious trend during the study period.
The TD estimates in the 2010s (1.51, 1.40–1.79 Tg N yr$^{-1}$) were on average 0.38 Tg N yr$^{-1}$ higher than the BU estimates (1.14, 0.28–2.21 Tg N yr$^{-1}$) (Figure 15). According to the BU results, the average natural emission was 0.95 Tg N yr$^{-1}$ in the 2010s, contributing to 84% of total emissions. Direct agricultural emissions, other direct emissions, and indirect emissions were 0.28 (0.22–0.35) Tg N yr$^{-1}$, 0.09 (0.06–0.11) Tg N yr$^{-1}$ and 0.05 (0.02–0.07) Tg N yr$^{-1}$, respectively. Changes in climate, $CO_2$ and land cover had an overall negative effect on $N_2O$ emissions with the mean value of -0.23 Tg N yr$^{-1}$, ranging from -0.45 Tg N
yr$^{-1}$ to 0.05 Tg N yr$^{-1}$.





### 3.5.6 Southwest South America (SSA)

BU and TD estimates are consistent in the magnitude of the total $N_2O$ emissions from Southwest South America during 1980-2020, with values of 0.52 (0.14–0.99) Tg N yr$^{-1}$ and 0.51 (0.40–0.63) Tg N yr$^{-1}$ (Figure 13h), respectively. TD estimates increased at the rate of $5.3 \times 10^{-3}$ Tg N yr$^{-2}$ over 1997-2020, however, BU estimates did not have an obvious trend during this period. Among the four anthropogenic sectors, direct agricultural emissions had the largest increase, from 0.10 Tg N yr$^{-1}$ in 1980 to 0.15 Tg N yr$^{-1}$ in 2020 (Figure 14). Indirect emissions also increased from 0.02 Tg N yr$^{-1}$ in 1980 to 0.03 Tg N yr$^{-1}$ in 2020. Perturbed fluxes from changes in climate, $CO_2$ and land cover had a decreasing trend, while emissions from other sectors fluctuated over the study period.

The BU and TD approaches gave similar estimates of total $N_2O$ emissions from Southwest South America in the 2010s, with values of 0.52 (0.20–0.97) Tg N yr$^{-1}$ and 0.55 (0.44–0.67) Tg N yr$^{-1}$ (Figure 15), respectively. The mean natural emission was 0.39 Tg N yr$^{-1}$ in the 2010s, accounting for 75% of total emissions. Direct agricultural emissions, other direct emissions, and indirect emissions were 0.14 (0.09–0.19) Tg N yr$^{-1}$, 0.05 (0.03–0.06) Tg N yr$^{-1}$ and 0.03 (0.01–0.03) Tg N yr$^{-1}$, respectively. Changes in climate, $CO_2$ and land cover had an overall negative effect on $N_2O$ emissions with the mean value of -0.08 Tg N yr$^{-1}$, ranging from -0.16 Tg N yr$^{-1}$ to 0.01 Tg N yr$^{-1}$.

### 3.5.7 Europe (EU)

The BU estimates suggest that Europe had the largest decrease rate of regional $N_2O$ emissions among the 18 regions, and the average decrease rate during 1980-2020 was $-11.4 \times 10^{-3}$ Tg N yr$^{-2}$ (Figure 13i). For the period 1997-2020, this decreasing trend slowed-down as estimated by BU approaches ($-7.5 \times 10^{-3}$ Tg N yr$^{-2}$), while the TD approach suggests a small increase of ($1.6 \times 10^{-3}$ Tg N yr$^{-2}$) (Figure 13i). Emissions from other direct anthropogenic sources (including 'Fossil fuel and industry', 'Waste and wastewater', and 'Biomass burning'), which decreased from 0.51 Tg N yr$^{-1}$ in 1980 to 0.18 Tg N yr$^{-1}$ in 2020, made the largest contribution to the decreasing trend in $N_2O$ emissions from Europe. Direct agricultural emissions and indirect emissions show overall decrease trends from 0.46 and 0.16 Tg N yr$^{-1}$ in 1980 to 0.38 and 0.12 Tg N yr$^{-1}$ in 2020, respectively. However, the decreasing trend in direct agricultural emissions has leveled off since the 2000s. Perturbed fluxes from changes in climate, $CO_2$ and land cover decreased during 1980-1985, then slowly increased (Figure 14).

The BU and TD approaches gave comparable estimates of European $N_2O$ emissions in the 2010s, with values of 0.99 (0.48–1.54) Tg N yr$^{-1}$ and 0.86 (0.49–1.36) Tg N yr$^{-1}$ (Figure 15), respectively. According to the BU results, natural sources only contributed to 27% of total emissions (mean: 0.26, 0.11–0.52 Tg N yr$^{-1}$) during this period. Direct agricultural emissions, other direct emissions, and indirect emissions were 0.37 (0.28–0.44) Tg N yr$^{-1}$, 0.19 (0.15–0.24) Tg N yr$^{-1}$ and 0.13 (0.08–0.16) Tg N yr$^{-1}$, respectively. Changes in climate, $CO_2$ and land cover had an overall positive effect on $N_2O$ emissions with the mean value of 0.04 Tg N yr$^{-1}$, ranging from -0.14 Tg N yr$^{-1}$ to 0.18 Tg N yr$^{-1}$.



### 3.5.8 Northern Africa (NAF)

For Northern Africa, TD approaches suggested a larger magnitude of the total $N_2O$ emissions than BU approaches over the
period 1997-2020 (Figure 13j), with the values of 1.01 (0.52–1.32) Tg N yr$^{-1}$ and 0.66 (0.18–1.21) Tg N yr$^{-1}$, respectively.
Both approaches suggest that $N_2O$ emissions from Northern Africa significantly increased during 1997-2020, and the increase
rates estimated by the BU and TD approaches were $3.6 \times 10^{-3}$ Tg N yr$^{-2}$ and $4.7 \times 10^{-3}$ Tg N yr$^{-2}$, respectively. Direct emissions
increased from 0.10 Tg N yr$^{-1}$ in 1980 to 0.27 Tg N yr$^{-1}$ in 2020, making the largest contribution to the increase in $N_2O$
emissions from Northern Africa (Figure 14). Indirect emissions also significantly increased from 0.02 Tg N yr$^{-1}$ in 1980 to
0.04 Tg N yr$^{-1}$ in 2020. In contrast, other anthropogenic emissions decreased from 0.12 Tg N yr$^{-1}$ in 1980 to 0.11 Tg N yr$^{-1}$ in
2020. $N_2O$ Fluxes caused by changes in climate, $CO_2$ and land cover remained relatively stable during 1980-2020.
In the 2010s, the BU estimates (0.68, 0.18–1.20 Tg N yr$^{-1}$) were on average 0.36 Tg N yr$^{-1}$ lower than the TD estimates (1.03,
0.71–1.45 Tg N yr$^{-1}$) (Figure 15). Natural sources accounted for 46% of total emissions (0.32, 0.08–0.59 Tg N yr$^{-1}$) during
this period. Direct emissions from nitrogen additions in agriculture were 0.23 (0.09-0.34) Tg N yr$^{-1}$, contributing to 34% of
the total emissions. Emissions from other direct anthropogenic sources made the second largest contribution to anthropogenic
emissions, with the decadal mean of 0.11 (0.08-0.14) Tg N yr$^{-1}$. Indirect emissions and perturbed fluxes from changes in
climate, $CO_2$ and land cover were 0.04 (0.02-0.06) Tg N yr$^{-1}$ and -0.01 (-0.09-0.06), respectively.

### 3.5.9 Equatorial Africa (EQAF)

Similar to Northern Africa, TD approaches suggested a larger magnitude of total $N_2O$ emissions from Equatorial Africa than
BU approaches over the period 1997-2020 (Figure 13k), with values of 1.45 (1.15–1.78) Tg N yr$^{-1}$ and 1.27 (0.35–2.05) Tg N
yr$^{-1}$, respectively. Both approaches suggested that $N_2O$ emissions from Equatorial Africa significantly increased during 1997-
2020, and the increase rates estimated by the BU and TD approaches were $3.1 \times 10^{-3}$ Tg N yr$^{-1}$ and $2.1 \times 10^{-3}$ Tg N yr$^{-1}$,
respectively. Direct emissions more than tripled during the study period, from 0.07 Tg N yr$^{-1}$ in 1980 to 0.22 Tg N yr$^{-1}$ in 2020,
dominating the increase in $N_2O$ emissions from Equatorial Africa (Figure 14). Indirect emissions also steadily increased from
0.04 Tg N yr$^{-1}$ in 1980 to 0.06 Tg N yr$^{-1}$ in 2020. On the contrary, perturbed fluxes from changes in climate, $CO_2$ and land
cover showed an overall decreasing trend with large interannual variabilities. Emissions from other anthropogenic sources
show relatively stable.
The BU and TD approaches gave comparable estimates of $N_2O$ emissions from Equatorial Africa in the 2010s, with values of
1.29 (0.40–2.04) Tg N yr$^{-1}$ and 1.50 (1.15–1.80) Tg N yr$^{-1}$ (Figure 15), respectively. According to the BU results, natural
emissions were the dominant component, accounting for 76% of total emissions (mean: 0.98, 0.42–1.31 Tg N yr$^{-1}$) during this
period. Direct agricultural emissions, other direct emissions, and indirect emissions were 0.18 (0.13–0.25) Tg N yr$^{-1}$, 0.26
(0.19–0.34) Tg N yr$^{-1}$ and 0.05 (0.03–0.08) Tg N yr$^{-1}$, respectively. Changes in climate, $CO_2$ and land cover had an overall
negative effect on $N_2O$ emissions with the mean value of -0.18 Tg N yr$^{-1}$, ranging from -0.37 Tg N yr$^{-1}$ to 0.06 Tg N yr$^{-1}$.





### 3.5.10 Southern Africa (SAF)

BU and TD estimates are consistent in the magnitude of the total $N_2O$ emissions from Southern Africa during 1997-2020, at 0.59 (0.14–1.02) Tg N yr$^{-1}$ and 0.58 (0.33–0.86) Tg N yr$^{-1}$ (Figure 13l), respectively. TD estimates increased at the rate of $4.5 \times 10^{-3}$ Tg N yr$^{-2}$ over 1997-2020, however, BU estimates did not show an obvious trend during this period. According to the BU results, direct agricultural emissions increased from 0.05 Tg N yr$^{-1}$ in 1980 to 0.08 Tg N yr$^{-1}$ in 2020, while emissions from other anthropogenic sources slightly decreased from 0.19 Tg N yr$^{-1}$ in 1980 to 0.17 Tg N yr$^{-1}$ in 2020. Both indirect emissions and perturbed fluxes from changes in climate, $CO_2$ and land cover had no significant trend (Figure 14).

BU and TD approaches gave consistent estimates of total $N_2O$ emissions from Southern Africa in the 2010s, with values of 0.59 (0.21–0.97) Tg N yr$^{-1}$ and 0.61 (0.35–0.87) Tg N yr$^{-1}$ for BU and TD approaches (Figure 15), respectively. Natural emissions were the dominant components, accounting for 64% of total emissions (mean: 0.38, 0.13–0.59 Tg N yr$^{-1}$) during this period. Direct agricultural emissions, other direct emissions, and indirect emissions were 0.07 (0.05–0.09) Tg N yr$^{-1}$, 0.19 (0.17–0.23) Tg N yr$^{-1}$ and 0.02 (0.01–0.03) Tg N yr$^{-1}$, respectively. Changes in climate, $CO_2$ and land cover had an overall negative effect on $N_2O$ emissions with the mean value of -0.07 Tg N yr$^{-1}$, ranging from -0.16 Tg N yr$^{-1}$ to 0.03 Tg N yr$^{-1}$.

### 3.5.11 Russia (RUS)

The average total $N_2O$ emissions from Russia estimated by BU approaches was 0.74 Tg N yr$^{-1}$, ranging from 0.15 Tg N yr$^{-1}$ to 1.85 Tg N yr$^{-1}$ (Figure 13m), which was much higher than the estimates of TD approaches (mean: 0.36, 0.18–0.52 Tg N yr$^{-1}$). Both approaches suggested that Russia's total $N_2O$ emissions increased during 1997-2020, and the increase rates estimated by the BU and TD approaches were $1.4 \times 10^{-3}$ Tg N yr$^{-2}$ and $1.7 \times 10^{-3}$ Tg N yr$^{-2}$, respectively. Direct agricultural emissions, other direct emissions, and indirect emissions had divergent trends before and after 1997. From 1980 to 1997, $N_2O$ emissions from all these three sectors decreased. After 1997, direct agricultural emissions and other direct emissions had an overall increasing trend, while indirect emissions remained relatively stable. Perturbed fluxes from changes in climate, $CO_2$ and land cover showed relatively stable with large interannual variabilities (Figure 14).

In the 2010s, the BU estimates (0.75, 0.17–1.84 Tg N yr$^{-1}$) were on average 0.36 Tg N yr$^{-1}$ higher than the TD estimates (0.38, 0.18–0.59 Tg N yr$^{-1}$) (Figure 15). Natural sources accounted for 63% of total emissions (0.47, 0.12–1.22 Tg N yr$^{-1}$) during this period. Direct agricultural emissions, other direct emissions, and indirect emissions were 0.06 (0.05–0.07) Tg N yr$^{-1}$, 0.10 (0.04–0.18) Tg N yr$^{-1}$ and 0.05 (0.03–0.07) Tg N yr$^{-1}$, respectively. Changes in climate, $CO_2$ and land cover had an overall positive effect on $N_2O$ emissions with the mean value of 0.06 Tg N yr$^{-1}$, ranging from -0.08 Tg N yr$^{-1}$ to 0.30 Tg N yr$^{-1}$.

### 3.5.12 Central Asia (CAS)

TD approaches suggested a larger magnitude of total $N_2O$ emissions from Central Asia than BU approaches over the period 1997-2020 (Figure 13n), with values of 0.19 (0.10–0.29) Tg N yr$^{-1}$ and 0.14 (0.01–0.27) Tg N yr$^{-1}$, respectively. BU and TD estimates were consistent in the trend of total $N_2O$ emissions during 1997-2020, with increase rates of $1.9 \times 10^{-3}$ Tg N yr$^{-2}$ and





$2.0 \times 10^{-3}$ Tg N $yr^{-2}$, respectively. Direct emissions increased from 0.05 Tg N $yr^{-1}$ in 1980 to 0.07 Tg N $yr^{-1}$ in 2020, making the largest contribution to the increase in $N_2O$ emissions from Central Asia. Other direct emissions and indirect emissions had no significant trend. Fluxes from changes in climate, $CO_2$ and land cover showed an overall increasing trend with large interannual variability (Figure 14).

In the 2010s, the TD estimates (0.20, 0.10–0.32 Tg N $yr^{-1}$) were on average 0.05 Tg N $yr^{-1}$ higher than the BU estimates (0.15, 0.03–0.28 Tg N $yr^{-1}$) (Figure 15). Natural sources accounted for 30% of total emissions (0.04, 0.01–0.10 Tg N $yr^{-1}$) during this period. Direct agricultural emissions, other direct emissions, and indirect emissions were 0.06 (0.02–0.08) Tg N $yr^{-1}$, 0.02 (0.01–0.02) Tg N $yr^{-1}$ and 0.02 (0.01–0.03) Tg N $yr^{-1}$, respectively. Changes in climate, $CO_2$ and land cover had an overall positive effect on $N_2O$ emissions with the mean value of 0.02 Tg N $yr^{-1}$, ranging from -0.03 Tg N $yr^{-1}$ to 0.05 Tg N $yr^{-1}$.

### 3.5.13 Middle East (MIDE)

BU and TD estimates are comparable for the magnitude of the total $N_2O$ emissions from the Middle East during 1997-2020, with values of 0.27 (0.11–0.45) Tg N $yr^{-1}$ and 0.30 (0.25–0.36) Tg N $yr^{-1}$ (Figure 13o), respectively. BU and TD estimates were consistent in the trend of total $N_2O$ emissions during 1997-2020, with increase rates of $4.4 \times 10^{-3}$ Tg N $yr^{-2}$ and $3.9 \times 10^{-3}$ Tg N $yr^{-2}$, respectively. According to the BU results, direct agricultural emissions increased from 0.07 Tg N $yr^{-1}$ in 1980 to 0.13 Tg N $yr^{-1}$ in 2020. Emissions from other anthropogenic sources (Fossil fuel and industry particularly) had the largest increase, from 0.03 Tg N $yr^{-1}$ in 1980 to 0.10 Tg N $yr^{-1}$ in 2020. Indirect emissions also continuously increased from 0.02 Tg N $yr^{-1}$ in 1980 to 0.04 Tg N $yr^{-1}$ in 2020. Perturbed fluxes from changes in climate, $CO_2$ and land cover had no significant trend (Figure 14).

BU and TD approaches gave consistent estimates of total $N_2O$ emissions from the Middle East in the 2010s, with values of 0.28 (0.13–0.49) Tg N $yr^{-1}$ and 0.32 (0.26–0.39) Tg N $yr^{-1}$ for BU and TD approaches (Figure 15), respectively. Natural emissions were 0.04 (0.02–0.08 Tg N $yr^{-1}$), accounting for 15% of total emissions during this period. Direct agricultural emissions, other direct emissions, and indirect emissions were 0.11 (0.05–0.21) Tg N $yr^{-1}$, 0.09 (0.07–0.10) Tg N $yr^{-1}$ and 0.03 (0.02–0.04) Tg N $yr^{-1}$, respectively. Changes in climate, $CO_2$ and land cover had an overall positive effect on $N_2O$ emissions with the mean value of 0.01 Tg N $yr^{-1}$, ranging from -0.02 Tg N $yr^{-1}$ to 0.05 Tg N $yr^{-1}$.

### 3.5.14 China (CHN)

BU and TD approaches agreed very well regarding the magnitudes and trends of $N_2O$ emissions from China. Both approaches suggested that China's total $N_2O$ emissions significantly increased during 1997-2020, and the increase rates estimated by the BU and TD approaches were $12.6 \times 10^{-3}$ Tg N $yr^{-1}$ and $16.5 \times 10^{-3}$ Tg N $yr^{-1}$, respectively (Figure 13p). According to the BU results, China's total $N_2O$ emissions increased from 0.75 Tg N $yr^{-1}$ in 1980 to 1.35 Tg N $yr^{-1}$ in 2020. Direct emissions from N additions in agriculture made the largest contribution to the increase in China's $N_2O$ emissions, which increased from 0.29 Tg N $yr^{-1}$ in 1980 to 0.71 Tg N $yr^{-1}$ in 2016 and then decreased to 0.64 Tg N $yr^{-1}$ in 2020 due to decreased N fertilizer application (Figure 14). Both indirect emissions and other direct emissions continuously increased, from 0.09 and 0.11 Tg N





yr$^{-1}$ in 1980 to 0.24 and 0.27 Tg N yr$^{-1}$ in 2020, respectively. The total anthropogenic N$_2$O emissions from China increased at the average rate of 18.1×10$^{-3}$ Tg N yr$^{-2}$ during 1980-2020, which was the largest among the 18 regions and contributed to 42.3% of the increase in global anthropogenic N$_2$O emissions.

The BU and TD approaches gave consistent estimates of China's total N$_2$O emissions in the 2010s, with values of 1.39 (0.83–2.13) Tg N yr$^{-1}$ and 1.33 (1.06–1.60) Tg N yr$^{-1}$ for BU and TD approaches (Figure 15), respectively. According to the BU results, natural sources only contributed 21% of total emissions (0.29, 0.20–0.51 Tg N yr$^{-1}$) during this period. Nitrogen additions in agriculture were the dominant source of N$_2$O emissions, contributing to 49% of the total emissions (0.68, 0.48–1.03 Tg N yr$^{-1}$). Emissions from other direct anthropogenic sources and indirect emissions from anthropogenic nitrogen additions were 0.23 (0.23–0.23) Tg N yr$^{-1}$ and 0.24 (0.17–0.28) Tg N yr$^{-1}$, respectively. Changes in climate, CO$_2$ and land cover had an overall negative effect on N$_2$O emissions with the mean value of -0.05 Tg N yr$^{-1}$, ranging from -0.24 Tg N yr$^{-1}$ to 0.08 Tg N yr$^{-1}$.

### 3.5.15 Korea and Japan (KAJ)

TD approaches suggested a smaller magnitude of total N$_2$O emissions from Korea and Japan than BU approaches over the period 1997-2020 (Figure 13q), with the values of 0.06 (0.03–0.11) Tg N yr$^{-1}$ and 0.10 (0.05–0.16) Tg N yr$^{-1}$, respectively. Both approaches suggested that total N$_2$O emissions from Korea and Japan decreased during 1997-2020, and the decrease rates estimated by the BU and TD approaches were -1.4 ×10$^{-3}$ Tg N yr$^{-2}$ and -0.5×10$^{-3}$ Tg N yr$^{-2}$, respectively. Other direct emissions (fossil fuel and industry, particularly) dominated the temporal variations of N$_2$O emissions from Korea and Japan, which increased from 0.04 Tg N yr$^{-1}$ in 1980 to 0.08 Tg N yr$^{-1}$ in 1997 and then decreased to 0.04 Tg N yr$^{-1}$ in 2020. Emissions from agriculture, indirect sources and perturbed fluxes remained relatively stable during 1997-2020 (Figure 14).

In the 2010s, BU estimates (mean: 0.10, 0.05–0.15 Tg N yr$^{-1}$) of total N$_2$O emissions were on average 0.04 Tg N yr$^{-1}$ higher than the TD estimate (0.06, 0.04–0.11 Tg N yr$^{-1}$) (Figure 15). Natural sources accounted for 27% of total emissions (0.03, 0.00–0.05 Tg N yr$^{-1}$) during this period. Direct agricultural emissions, other direct emissions, and indirect emissions were 0.03 (0.02–0.04) Tg N yr$^{-1}$, 0.04 (0.04–0.04) Tg N yr$^{-1}$ and 0.01 (0.01–0.02) Tg N yr$^{-1}$, respectively. Changes in climate, CO$_2$ and land cover had an overall negative effect on N$_2$O emissions with the mean value of -0.01 Tg N yr$^{-1}$, ranging from -0.02 Tg N yr$^{-1}$ to 0.00 Tg N yr$^{-1}$.

### 3.5.16 South Asia (SAS)

BU and TD estimates are comparable in terms of both the magnitude and trend of the total N$_2$O emissions from South Asia (Figure 13r). The magnitudes of total N$_2$O emissions estimated by BU and TD approaches were 1.03 (0.35–1.80) Tg N yr$^{-1}$ and 1.21 (0.96–1.56) Tg N yr$^{-1}$, respectively. Both approaches suggested that the total N$_2$O emissions from South Asia significantly increased during 1997-2020, and the increase rates estimated by BU and TD approaches were 17.6 ×10$^{-3}$ Tg N yr$^{-2}$ and 20.2×10$^{-3}$ Tg N yr$^{-2}$, respectively. Direct emissions from nitrogen additions in agriculture made the largest contribution to the increase in N$_2$O emissions in South Asia, which increased from 0.19 Tg N yr$^{-1}$ in 1980 to 0.55 Tg N yr$^{-1}$ in 2020 due to





increased N fertilizer application (Figure 14). Other direct emissions and indirect emissions also significantly increased, from 0.06 and 0.06 Tg N yr$^{-1}$ in 1980 to 0.14 and 0.17 Tg N yr$^{-1}$ in 2020, respectively. Fluxes from changes in climate, $CO_2$ and land cover showed an overall increasing trend with large interannual variabilities.

BU estimates (1.15, 0.44–2.02 Tg N yr$^{-1}$) were on average 0.21 Tg N yr$^{-1}$ lower than the TD estimate in the 2010s (1.36, 1.05–1.84 Tg N yr$^{-1}$) (Figure 15). Natural sources accounted for 28% of total emissions (0.32, 0.12–0.55 Tg N yr$^{-1}$) during this

period. Direct agricultural emissions, other direct emissions, and indirect emissions were 0.49 (0.25–0.75) Tg N yr$^{-1}$, 0.13 (0.13–0.13) Tg N yr$^{-1}$ and 0.15 (0.10–0.19) Tg N yr$^{-1}$, respectively. Changes in climate, $CO_2$ and land cover had an overall positive effect on $N_2O$ emissions with the mean value of 0.05 Tg N yr$^{-1}$, ranging from -0.15 Tg N yr$^{-1}$ to 0.39 Tg N yr$^{-1}$.

### 3.5.17 Southeast Asia (SEAS)

TD approaches suggested a smaller magnitude of the total $N_2O$ emissions from Southeast Asia than BU approaches over the

period 1997-2020 (Figure 13s), with values of 0.69 (0.50–1.02) Tg N yr$^{-1}$ and 0.82 (0.21–1.87) Tg N yr$^{-1}$, respectively. Both approaches suggested that total $N_2O$ emissions from Southeast Asia increased during 1997-2020, and the rates of increase estimated by the BU and TD approaches were $4.3 \times 10^{-3}$ Tg N yr$^{-2}$ and $2.3 \times 10^{-3}$ Tg N yr$^{-2}$, respectively. Direct agricultural emissions, other direct emissions, and indirect emissions significantly increased during the study period, from 0.09, 0.08 and 0.04 Tg N yr$^{-1}$ in 1980 to 0.29, 0.11 and 0.12 Tg N yr$^{-1}$ in 2020, respectively. Meanwhile, perturbed fluxes from changes in

climate, $CO_2$ and land cover significantly decreased from -0.20 Tg N yr$^{-1}$ in 1980 to -0.27 Tg N yr$^{-1}$ in 2020 (Figure 14).

The BU and TD approaches gave comparable estimates of the total $N_2O$ emissions from Southeast Asia in the 2010s, with values of 0.84 (0.28–1.85) Tg N yr$^{-1}$ and 0.72 (0.51–1.12) Tg N yr$^{-1}$ for BU and TD approaches (Figure 15), respectively. Natural sources accounted for 70% of total emissions (mean: 0.59, 0.24–1.30 Tg N yr$^{-1}$) during this period. Direct agricultural emissions, other direct emissions, and indirect emissions were 0.26 (0.20–0.34) Tg N yr$^{-1}$, 0.11 (0.09–0.13) Tg N yr$^{-1}$ and 0.10

(0.06–0.14) Tg N yr$^{-1}$, respectively. Changes in climate, $CO_2$ and land cover had an overall negative effect on $N_2O$ emissions with the mean value of -0.22 Tg N yr$^{-1}$, ranging from -0.30 Tg N yr$^{-1}$ to -0.07 Tg N yr$^{-1}$.

### 3.5.18 Australasia (AUS)

BU and TD estimates are comparable in terms of magnitude of the total $N_2O$ emissions from Australasia (Figure 13t). The magnitudes of total $N_2O$ emissions estimated by BU and TD approaches were 0.44 (0.02–0.93) Tg N yr$^{-1}$ and 0.52 (0.21–0.72)

Tg N yr$^{-1}$, respectively. TD estimates increased at the rate of $4.4 \times 10^{-3}$ Tg N yr$^{-2}$ over 1997-2020; however, BU estimates did not show a notable trend during this period (Figure 13t). According to the BU results, direct agricultural emissions increased from 0.08 Tg N yr$^{-1}$ in 1980 to 0.09 Tg N yr$^{-1}$ in 2020, while emissions from all the other three anthropogenic sectors remained stable (Figure 14).

In the 2010s, the magnitudes of total $N_2O$ emissions estimated by BU and TD approaches were 0.42 (0.03–0.82) Tg N yr$^{-1}$ and

0.53 (0.20–0.71) Tg N yr$^{-1}$, respectively. Natural sources accounted for 60% of total emissions (0.25, 0.05–0.47 Tg N yr$^{-1}$) during this period. Direct agricultural emissions, other direct emissions, and indirect emissions were 0.08 (0.02–0.12) Tg N


yr$^{-1}$, 0.08 (0.06–0.11) Tg N yr$^{-1}$ and 0.12 (0.07–0.16) Tg N yr$^{-1}$, respectively. Changes in climate, CO$_2$ and land cover had an overall negative effect on N$_2$O emissions with the mean value of -0.01 Tg N yr$^{-1}$, ranging from -0.12 Tg N yr$^{-1}$ to -0.10 Tg N yr$^{-1}$ (Figure 15).

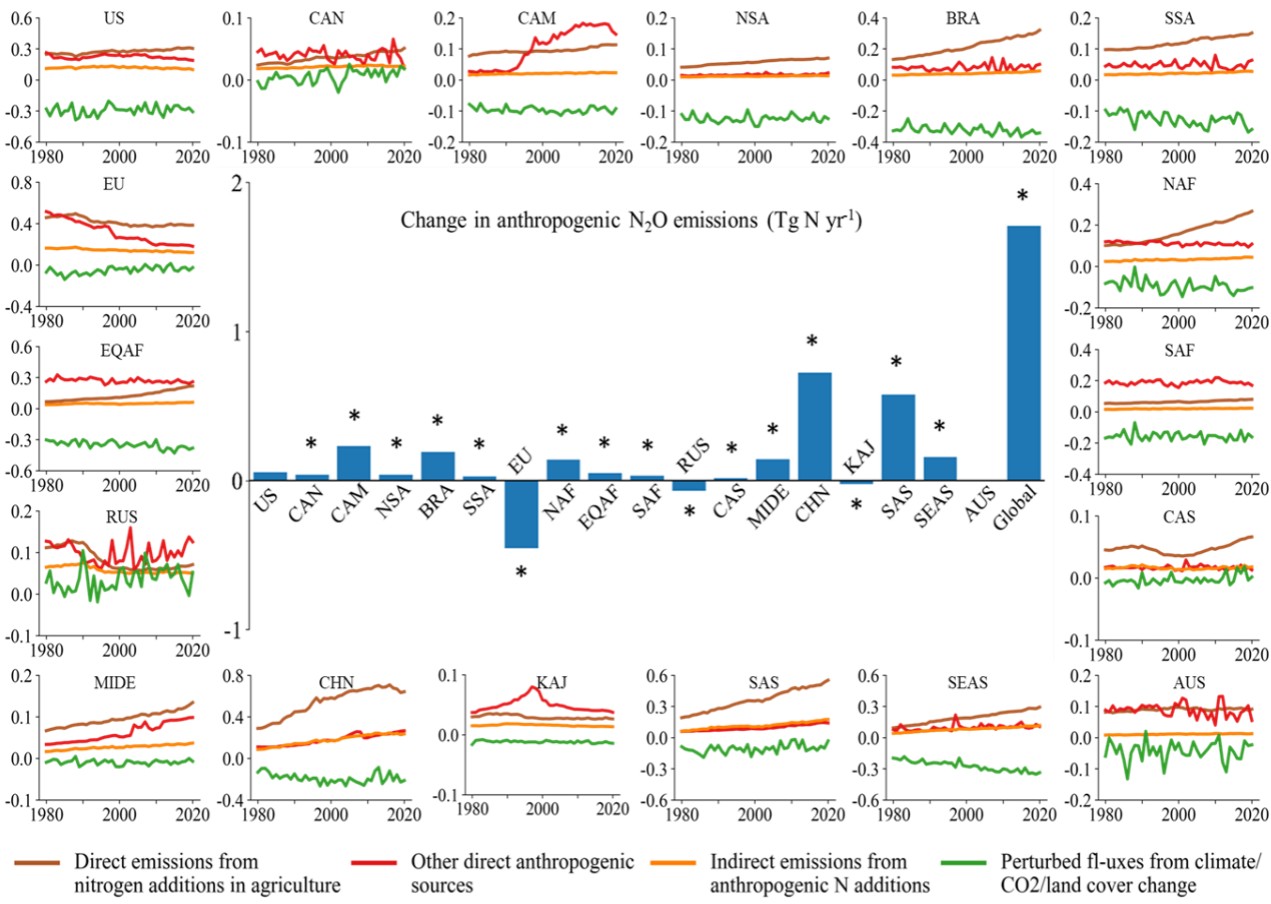


**Figure 14. Ensembles of regional anthropogenic N$_2$O emissions over the period 1980–2020. The bar chart in the centre shows the total changes in regional and global N$_2$O emissions during the study period of 1980–2020. Error bars indicate the 95% confidence interval for the average of the changes. The Mann–Kendall test was performed to establish any trends globally and for each region over the period 1980–2020. The changes were calculated from the annual change rate (Tg N yr$^{-2}$), determined from a linear regression, multiplied by 40 years. All regions except Australasia and the USA show a significant increasing or decreasing trend in the estimated ensemble N$_2$O emissions during 1980-2020. \*P < 0.05.**



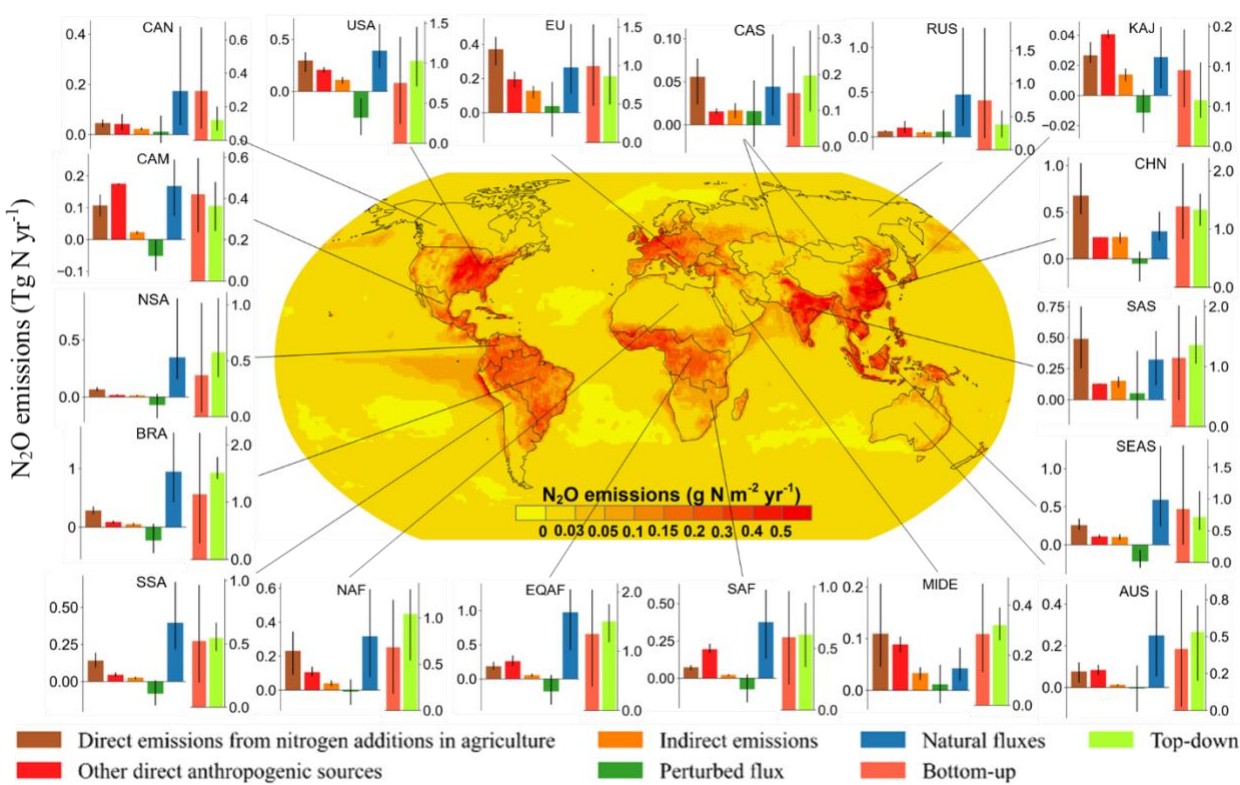

**Figure 15. Regional N₂O emissions during 2010-2019. Each subplot shows the emissions from five sub-sectors using BU approaches, followed by the sum of these five categories using BU approaches (blue) and the estimates from TD approaches (yellow). Error bars indicate the spread between the minimum and the maximum values. The centre map shows the spatial distribution of 10-year average N₂O emissions from land and ocean based on the land and ocean models.**

## 4. Discussion

### 4.1 Emission sources and comparison with previous estimates of the global N₂O budget

In comparing the global N₂O budget estimates with previous studies, the definitions and terminology used in this study for N₂O sources and sinks are consistent with those in Tian et al. (2020). In this new synthesis, we have also included a new emission source, namely "continental shelves", corresponding to the shallow portion of the ocean overlying continental shelves (Laruelle et al., 2013), which was not explicitly reported in the previous global N₂O budget (Tian et al. 2020). Thus, a total of 18 sources and 3 sinks are quantified in the global N₂O budget reported here. We utilized a similar methodology to synthesize multiple TD and BU estimates. The TD estimates of global total emissions in this study are consistent with Tian et al. (2020). However, the TD estimates of emissions from the ocean are about 2.3 Tg N yr⁻¹ lower than the previous estimate in the 2000s, while the TD estimates of land emissions are about 2.4 Tg N yr⁻¹ higher than the previous estimate for the decade 2007-2016 (Tian et al. 2020). Global BU estimates in this study are about 1.1 Tg N yr⁻¹ higher than the previous estimate, primarily due





to the inclusion of emissions from continental shelves (mean: 1.2 Tg N yr$^{-1}$) and 0.8 Tg N yr$^{-1}$ higher than the previous estimate for the natural soils baseline.

According to our analysis, natural soils contributed to more than half of terrestrial $N_2O$ emissions (Table 2), consistent with previous studies (Denman et al., 2007, Tian et al., 2020). The global natural soil emissions derived from this study are estimated

to be 6.4 Tg N yr$^{-1}$, with a large uncertainty ranging from 3.9 to 8.6 Tg N yr$^{-1}$. Using the emission factor from the IPCC 2006 Guidelines, Syakila and Kroeze (2011) estimated that global pre-industrial $N_2O$ emission from natural soils was 7 Tg N yr$^{-1}$. Xu et al. (2017) suggested that global natural soil $N_2O$ emissions were about 6.2 Tg N yr$^{-1}$, with an uncertainty range from 4.8 to 8.1 Tg N yr$^{-1}$. Tian et al. (2019) estimated global soil $N_2O$ emissions derived from NMIP using seven process-based Terrestrial Biosphere Models (TBMs) and suggested a global soil $N_2O$ emission of 6.3±1.1 Tg N yr$^{-1}$ in the 1860s.

The total of direct agricultural emissions, other direct anthropogenic emissions, and indirect anthropogenic emissions in this study is the same as the previous estimates (Tian et al. 2020). However, the total anthropogenic emissions in this study is lower than our previous estimate (Tian et al., 2020), mainly because of the differences in perturbed fluxes from climate, $CO_2$, and land cover change. According to our new estimate derived from NMIP2, the average perturbed flux from climate, $CO_2$, and land cover change was -0.6 (-2.1-1.2) Tg N yr$^{-1}$ during 2010-2019 (Table 2). By contrast, the average perturbed flux during

2007-2016 reported by Tian et al. (2020) was 0.2 (-0.6-1.1) Tg N yr$^{-1}$, which was based on the first phase of NMIP (Tian et al. 2018). This study suggests a larger negative effect of increased $CO_2$ concentration and reduced mature forest area on $N_2O$ emissions than Tian et al. (2020). Much uncertainty exists in estimating the perturbed fluxes of atmospheric $CO_2$ and manure forest conversion as discussed in the section of uncertainties followed.

Our estimate indicates that agricultural emissions were the major drivers of the increase in anthropogenic emissions during the

past four decades, increasing from 3.0 Tg N yr$^{-1}$ in 1980 to 5.0 Tg N yr$^{-1}$ in 2020 (Figure 16). Direct agricultural emissions had a larger increase than indirect agricultural emissions (2.2 Tg N yr$^{-1}$ in 1980 to 3.9 Tg N yr$^{-1}$ in 2020 versus 0.8 Tg N yr$^{-1}$ in 1980 to 1.2 Tg N yr$^{-1}$ in 2020). Agricultural emissions contributed to 74% of total anthropogenic emissions in the 2010s, with 56% from direct agricultural emissions and 18% from indirect emissions. Non-agricultural anthropogenic emissions had a slight decreasing trend during 1980-2020 because of a higher estimate of changes in climate, $CO_2$, and land cover than

previous estimate.



**Figure 16. Changes in N$_2$O emissions from anthropogenic emissions from agricultural and non-agricultural sources during 1980-2020 (a, c). (b) and (d) show average anthropogenic emissions from different sources during 2010-2019, error bars indicate the spread between the minimum and the maximum values. Here, direct agricultural emissions include emissions from fertilizer and manure applied on agricultural soils, manure left on pasture, manure management, and aquaculture. Indirect agricultural emissions include emissions from anthropogenic nitrogen additions to inland waters, estuaries and coastal vegetation, and N deposition on land. Other anthropogenic emissions are classified as non-agricultural anthropogenic emissions.**





This study divides the global land into 18 regions and provides a more detailed regional budget than a previous study which had only 10 regions (Tian et al., 2020), thus enhancing our understanding of the $N_2O$ budget in sub-regions of North America, South America, Africa, and East Asia. In the 1980s, Europe made the largest contribution to global anthropogenic $N_2O$

emissions (12.9%), followed by Equatorial Africa (11.4%), Brazil (9.4%), China (7.7%), Russia (7.6%), and the USA (6.6%). During the study period, Europe and Russia had the largest decline in share of anthropogenic $N_2O$ emissions, from 12.9% and 7.6% in the 1980s to 8.3% and 6.1% in the 2010s, respectively. In contrast, China and South Asia had the largest increase, from 7.7% and 6.5% in the 1980s to 11.4% and 9.4% in the 2010s, respectively. In the 2010s, China (11.4%), Equatorial Africa (10.6%), South Asia (9.4%), Brazil (9.3%), Europe (8.3%) were the top five contributors to global anthropogenic $N_2O$

emissions (Figure 17).

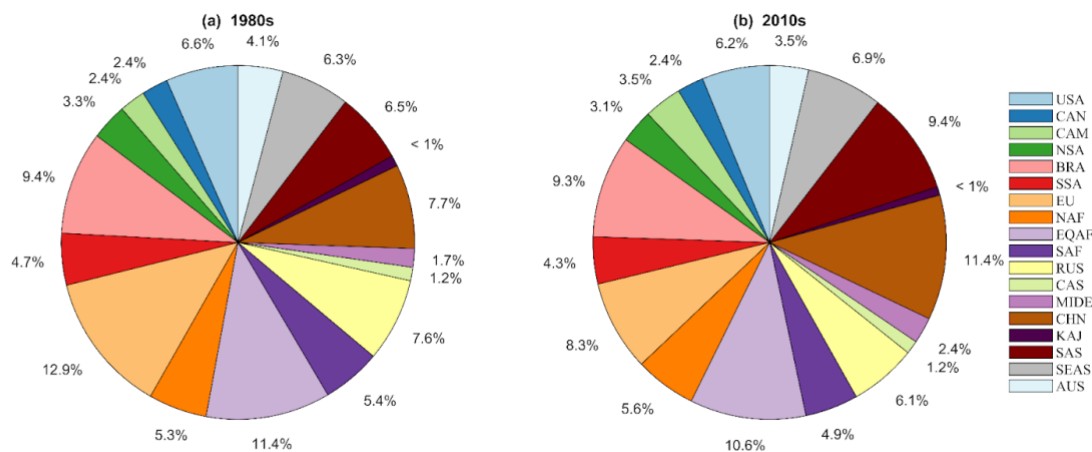

**Figure 17. Contributions of the 18 regions to global anthropogenic $N_2O$ emissions in the 1980s (a) and 2010s (b).**

Among the eighteen regions identified in this study, only Europe, Russia, and Japan and Korea had decreasing $N_2O$ emissions. Europe had the largest rate of decrease with an average of $-11.4 \times 10^{-3}$ Tg N $yr^{-2}$ during 1980-2020 (31% reduction), largely resulting from reduced emissions in fossil fuel and industry, which changed from 0.49 Tg N $yr^{-1}$ in 1980 to 0.14 Tg N $yr^{-1}$ in 2020. In addition to the large reduction of fossil fuel and industry emissions in Europe, direct agricultural emissions and indirect emissions show overall decrease trends from 0.46 and 0.16 Tg N $yr^{-1}$ in 1980 to 0.38 and 0.12 Tg N $yr^{-1}$ in 2020,

respectively. However, the decreasing trend in agricultural emissions has leveled off since the 2000s.





China and South Asia had the largest increase in $N_2O$ emissions during the study period. The rates of increase of anthropogenic emissions from China and South Asia were 18.1 x $10^{-3}$ and 14.5 x $10^{-3}$ Tg N $yr^{-2}$, respectively. The rates of increase of anthropogenic emissions from China and South Asia contributed 42.3% and 33.9% to the global anthropogenic increase rate

(0.04 Tg N $yr^{-2}$), respectively. In these two regions, direct nitrogen additions in agriculture made the largest contribution, while other direct emissions and indirect emissions also steadily increased. Our results show a significant increase in anthropogenic $N_2O$ emissions from South America, which is consistent with the previous budget (Tian et al., 2020). Moreover, we reveal that Brazil had a higher increase rate in anthropogenic $N_2O$ emissions (4.8 $\times10^{-3}$ Tg N $yr^{-2}$) than Northern South America (1.0 $\times10^{-3}$ Tg N $yr^{-2}$) and Southwest South America (0.7 $\times10^{-3}$ Tg N $yr^{-2}$) during 1980-2020, and direct emissions from agriculture

made the largest contribution. Our results suggest that Northern Africa made the largest contribution (63%) to the increase in anthropogenic $N_2O$ emissions from Africa, followed by Equatorial Africa (23%) and Southern Africa (14%). Anthropogenic $N_2O$ emissions from the USA and Canada show similar increasing rates of 1.3 $\times10^{-3}$ Tg N $yr^{-2}$ and (1.0 $\times10^{-3}$ Tg N $yr^{-2}$) during the period 1980-2020, respectively. Central America shows higher anthropogenic $N_2O$ emission increase rate (5.7 $\times10^{-3}$ Tg N $yr^{-2}$), attributing to increase in emissions from fossil fuels and industry from 0.01 Tg N $yr^{-1}$ in the 1980s to 0.16 Tg N $yr^{-1}$ in

the 2010s in Central America. The data for Mexico from EDGAR has a known problem with its estimates of $N_2O$ emissions from industry, which requires further exploration. To support countries' $N_2O$ mitigation, it is essential to accurately estimate sources and sinks of $N_2O$ at national level.

## 4.2. Sources of uncertainties and suggestions for improvements

### 4.2.1 Uncertainties in $N_2O$ emission factors

Four inventories of $N_2O$ emissions (EDGAR, FAOSTAT, GFED and UNFCCC) are integrated into the current synthesis of anthropogenic $N_2O$ emissions. These emission factor (EF)-based inventory datasets used the IPCC default EFs at regional and global scales. However, the poorly captured dependence of EFs on regional climate, management practices such as tillage, legume effect, and soil physical and biochemical conditions are key causes of the large uncertainty in the estimates of agricultural $N_2O$ emissions (Shcherbak et al., 2014; Tian et al., 2019; Lu et al., 2022), particularly for croplands where EFs

has high spatial heterogeneity (Shang et al., 2019; Wang et al., 2020). There is evidence of greater-than-linear dependence of emissions on N-input where there is an excess of N, which is not represented in inventories which assume a linear dependence on N-input (Cui et al. 2021). Higher IPCC-tier GHG inventories using the alternative EFs that are disaggregated by environmental factors and management-related factors (Buendia et al., 2019) could provide more accurate estimates, especially for regions where N input surplus is high such as Eastern China and India. Establishing national and regional $N_2O$ flux

measurement networks could improve the accuracy of EFs estimates for regions with different vegetation types and management measures. Furthermore, inventory datasets based on EF methods also suffer from large uncertainties induced by the underlying agriculture and rural data and statistics used as input, including statistics on fertilizer applications, livestock manure availability, storage and applications, and nutrient, crop and soils management.



According to the ensemble of process-based land model emissions derived from NMIP2, we estimate that the emission factor (EF) of fertilizer and manure applied on global croplands was 1.9% (1.2%-3.3%) in the 2010s, which is significantly larger than the IPCC Tier-1 default for direct emission of 1%. This higher EF derived from process-based models suggests a strong interactive effect between N additions and other global environmental changes (Table 2, Perturbed fluxes from climate, atmospheric $CO_2$, and land cover change). Figure 18 shows the spatial pattern of cropland $N_2O$ EF during the 2010s, and

highlights that the EF was high in eastern China, Southeast Asia, western Europe, and central USA where anthropogenic N inputs were high (Figure B3). Previous field experiments reported a better fit to local observations of soil $N_2O$ emissions when assuming a non-linear response to fertilizer N inputs under varied climate and soil conditions (Shcherbak et al. 2014; Wang et al. 2019). The non-linear response is likely also associated with long-term N accumulation in agricultural soils from N fertilizer use and in aquatic systems from N loads (the legacy effect) (Van Meter et al. 2016), which provides more substrate for

microbial processes (Firestone and Davidson 1989). The increasing $N_2O$ emissions estimated by process-based models (Tian et al. 2019) also suggest that recent climate change (particularly warming) may have boosted soil nitrification and denitrification processes, contributing to the growing trend in $N_2O$ emissions together with rising N additions to agricultural soils (Griffis et al. 2017; Parn et al. 2018; Smith 2017)

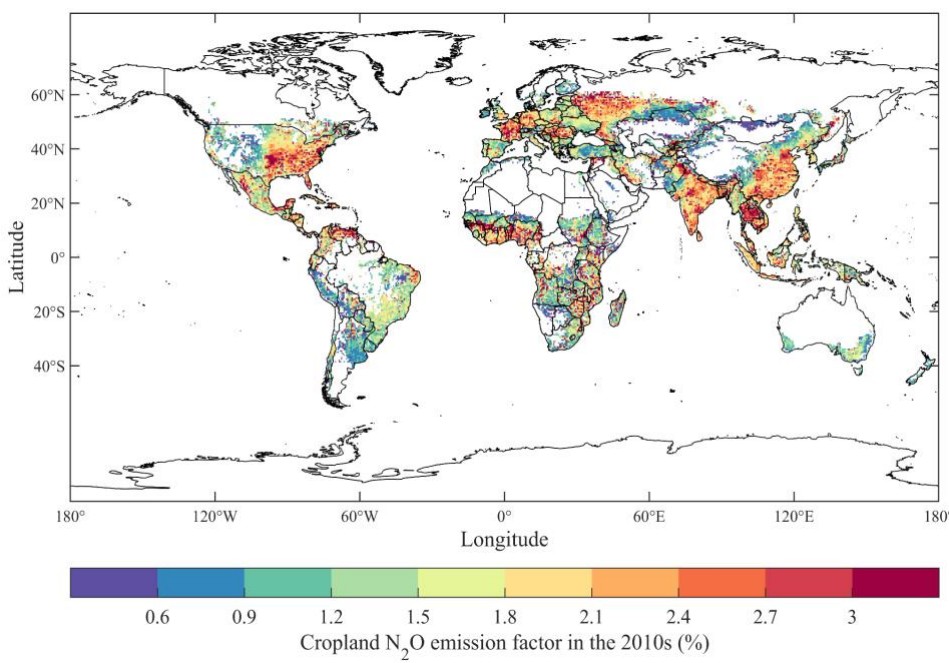


**Figure 18. Spatial pattern of the emission factor (EF) of fertilizer and manure applied on global croplands in the 2010s derived from NMIP2.**





### 4.2.2 Uncertainties in estimates of soil N₂O emissions

Both process-based land biosphere modeling and measurement-based upscaling approaches have been used to estimate global soil $N_2O$ emissions (Table 2), with large uncertainties in their estimates. As shown in Figure 19, NMIP2 models exhibit the highest uncertainties in the estimates of soil $N_2O$ emissions from tropical forests such as the Amazon Basin, the Congo Basin, and Southeast Asia, as well as in regions with high fertilizer application rate, including Eastern China, Northern India, and the US Corn Belt. A large discrepancy in natural soil emissions among NMIP2 models exists, ranging from 3.9 to 8.6 Tg N yr$^{-1}$,

which needs to be reconciled in future research.

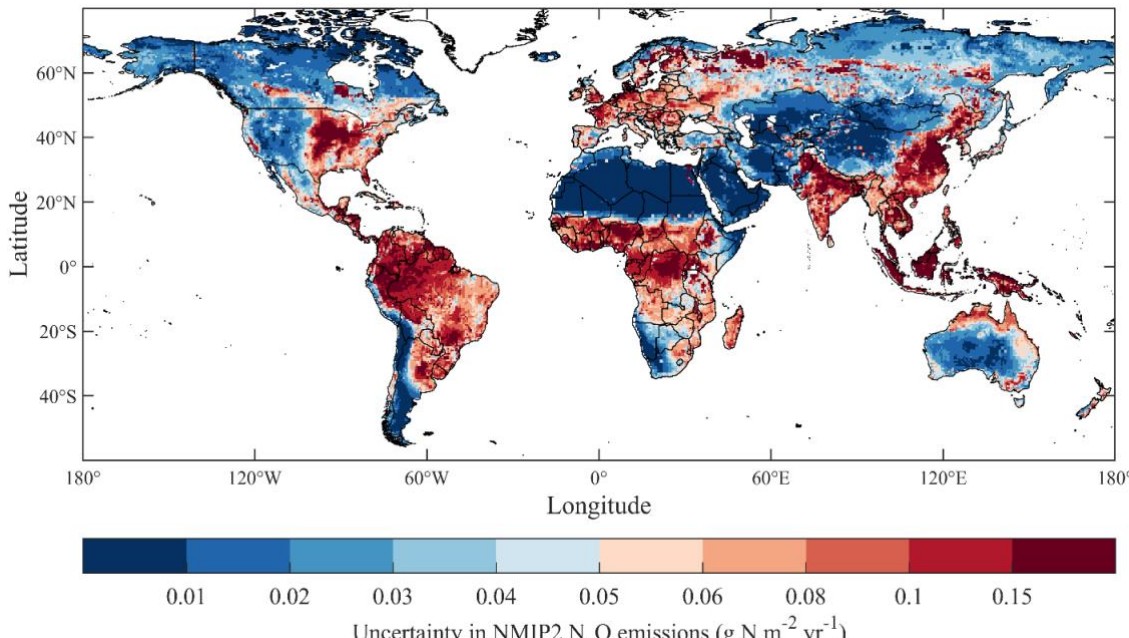

**Figure 19. Spatial distribution of uncertainty (one standard deviation) in NMIP2 estimations of soil N₂O emissions in the 2010s.**


**Uncertainties associated with NMIP2 models**: The uncertainties in process-based models primarily stem from differences in model configuration and process parameterization, as well as the missing processes and critical information (Tian et al., 2019).

First, the NMIP2 models use divergent schemes to represent the flows of reactive N through ecosystems (biological N fixation,

N deposition, N leaching, N volatilization, nitrification, and denitrification), which could result in large discrepancies in soil mineral N that serves as substrates for $N_2O$ production. Explicit representation of these processes is a critical need for enhancing model simulation accuracy.





Second, several important processes are missing in most process-based land models. Human management measures like tillage and legume cultivation can alter the physical and chemical characteristics of soil in croplands (Raji & Dörsch, 2020; Z. Yu et al., 2020), but they are not adequately represented in most NMIP2 models. Parameterizing these processes in the models is necessary to reduce uncertainty. Additionally, N addition in pasture and rangeland (e.g., livestock excreta deposition, manure, and mineral fertilizer application) constitutes an important source of global soil $N_2O$ emissions (Davidson, 2009), accounting for more than half of the global agricultural $N_2O$ emissions (Dangal et al., 2019). However, only DLEM considered these processes. The consideration of N addition in managed grasslands is an essential task for process-based models to estimate grassland soil $N_2O$ emissions accurately. Moreover, most process-based models did not explicitly consider seasonal freeze-thaw processes and the thawing of permafrost, which can emit substantial amounts of $N_2O$ (Marushchak et al., 2021; Marushchak et al., 2011; Repo et al., 2009; Voigt et al., 2017; Del Grosso et al. 2022). It is recommended to include explicit representation of permafrost physics and seasonal freeze–thaw processes in process-based models, as this would help better catch the "hot spot" and "hot moment" of soil $N_2O$ emissions in northern regions (Wagner-Riddle et al., 2017). Current process-based models also face challenges in adequately representing the fine-grained landscape structure of Arctic ecosystems (e.g., landscape elements that act as ultra-emitters of $N_2O$ like organic soil non-vegetated fractions), so integrating sub-grid information and processes into models may provide a solution for fine-grained physical-hydrological modeling.

Third, microbial nitrification and denitrification processes are regulated by multiple environmental factors, including substrate availability, precipitation, temperature, oxygen status, pH, vegetation type, and atmospheric $CO_2$ concentration (Butterbach-Bahl et al., 2013; Dijkstra et al., 2012; Li et al., 2020; Tian et al., 2019; Yin et al., 2022; Yu et al., 2022). However, there is significant divergence among NMIP2 models in their response to these factors. For example, simulated soil $N_2O$ emissions in response to N addition (i.e., fertilizer and manure N applications, and N deposition) exhibit large divergence among the participating NMIP2 models, primarily due to differences in model representation of N processes and parameterization schemes. Moreover, in contrast to our findings indicating N fertilizer application and manure additions as dominant drivers, Harris et al. (2022) identified N deposition as the primary contributor to anthropogenic $N_2O$ emissions, accounting for 41±14% of all anthropogenic emissions. These different findings highlight the complex nature of $N_2O$ emissions and the need for further research to better understand the relative contributions of different N sources. For the climatic effects on soil $N_2O$ emissions, our NMIP2 models indicate enhanced $N_2O$ emissions due to warming, consistent with findings from experiment-based studies (Smith, 1997, Cui et al., 2018; Voigt et al., 2017; Wang et al., 2017), as the denitrifying bacteria community may adapt to higher temperature (Pärn et al., 2018). Additionally, considering that microbial nitrification and denitrification are also largely controlled by soil moisture (Butterbach-Bahl et al., 2013), it is important to address the discrepancies in NMIP2 models concerning soil moisture representation, such as soil depth, root distribution, root water uptake, and water movement processes (Ostle et al., 2009; Raats, 2007, and Raoult et al., 2018).

At the global scale, although NMIP2 models show large discrepancies in the $CO_2$ effect on soil $N_2O$ emissions, most NMIP2 models show a negative effect, suggesting that enhanced plant N uptake caused by rising $CO_2$ concentration played a dominant role (Usyskin-Tonne et al., 2020; Tian et al., 2019). Nevertheless, observation-based results of the $CO_2$ effect diverge among





different ecosystem types, with some studies reporting reduced N$_2$O emissions in forests under elevated CO$_2$ (Phillips et al., 2001), while others found increased emissions in grasslands (Moser et al., 2018 and Regan et al., 2011). It should be noted that the interactions among environmental factors influencing soil N$_2$O emissions are still poorly represented in the NMIP2

models. Further targeted continuous measurements and manipulation experiments are needed to better represent the interactive effects of multiple environmental factors on N$_2$O emissions in the models to improve the simulation of complex N$_2$O dynamics. Finally, simulations targeted to explain the reconstructed increase in terrestrial N$_2$O emissions over the deglaciation and during past abrupt climate events will further help to constrain process-based models (Fischer et al., BG, 2019; Joos et al., BG, 2020).

**Land cover change/deforestation**: The two methods for estimating deforestation-induced N$_2$O changes have their limitations.

The accuracy of the empirical estimates of post-deforestation pulse N$_2$O emissions in tropical forests strongly depends on the availability of paired N$_2$O observations in deforested and nearby intact forest sites (Melillo et al., 2001; Verchot et al., 1999), which are extremely scarce. Moreover, a fixed value was adopted as the default reference N$_2$O emission rate for tropical forests to simplify computation, but it inevitably ignored the spatiotemporal heterogeneity in tropical forest N$_2$O emissions (Barthel et al., 2022). It is also noted that there were no empirical post-deforestation N$_2$O emission estimates in extra-tropical areas, as

no feasible empirical relationships between N$_2$O emissions and years after deforestation were available.  The accuracy of process-based estimates (specifically by DLEM here) could be regulated by model-specific configurations for land use change pathways. For example, in modeling tropical shift cultivation, DLEM assumed that agricultural lands newly converted from forests can only be reforested after at least 15 years to be consistent with the LUHv2 data (Ma et al., 2020). Meanwhile, treatments of different nitrogen pools (such as leaf, stem, root and litter pools) during land conversion would directly influence

the nitrogen substrate for nitrification and denitrification. The DLEM model follows the biomass allocation scheme proposed by previous studies (Houghton et al., 1983; McGuire et al., 2001), which may introduce uncertainty in varied land management practices. A bias in the LUHv2 land use change data in regions experiencing drastic land conversions could also contribute to uncertainty in deforestation induced greenhouse gas emissions, for example, in areas with large-scale plantations (Yu et al., 2022).

In addition, developing forcing datasets with high quality and high spatiotemporal resolution is also important for reducing uncertainties in simulated N$_2$O fluxes. Among various input variables, precise information regarding fertilizer and manure application (including crop-specific application rate, type, timing, and frequency) is pivotal for improving the accuracy of model simulations. However, this crucial information was not unified in NMIP2 simulations, leading to increased modeling uncertainty. To mitigate this issue, it is strongly recommended to use improved fertilizer and manure datasets that provide

detailed information on crop-specific application rate, timing and frequency to drive models in future intercomparison projects. Moreover, with the availability of additional high-precision datasets from manipulation field experiments (e.g., microbial data), we could use these datasets to constrain our models and delve deeper into the underlying mechanisms that regulate N$_2$O fluxes (e.g., the role of soil microbes) and further incorporate these mechanisms into models to reduce uncertainties.

**Uncertainties associated with measurement-based upscaling approach:** Measurement-based upscaling estimates are

subject to uncertainties due to various factors. One major reason is the limited recording of microscale variables and incomplete



quantification of local EFs related to microbial $N_2O$ production. Sampling limitations also contribute to uncertainties, as the frequency and repeatability of measurements may not fully capture the high spatiotemporal variability of $N_2O$ flux. The lack of the history of control sites further complicates the exclusion of observation data with significant legacy fluxes, thereby biasing our estimates. Additionally, gaps in global agricultural management datasets, particularly regarding fertilization details,

enlarge the prediction interval of EFs and introduce uncertainties. We then used a Monte Carlo simulation to estimate three sources of uncertainty for predicting EFs based on flux upscaling approach: i) the fixed coefficients, ii) the random coefficients, and iii) input data. The uncertainty from sampling frequency and replication is reflected in the first source, while the uncertainty from unquantified sources related to field measurements is reflected in the second source. Each of the crop-specific SRNM models was run by randomly generating the fixed and random coefficients from their fitted multivariate normal distribution,

as well as climate, soil, and other relevant factors following independent normal distributions with the mean of the value in our dataset and standard deviation of the absolute difference between the dataset used in this study and other global datasets. Fertilizer frequency was randomly selected using a Bernoulli distribution. Predicted values were calculated through 1000 iterations to construct a 95% prediction interval. The breakdown of uncertainty revealed that the random coefficients contributed the most to the estimation uncertainty, with observations showing that they explained more variance in EFs compared to fixed effects (47-74% vs. 19-35%) and contributed to the most of estimation uncertainty (Figure 20).

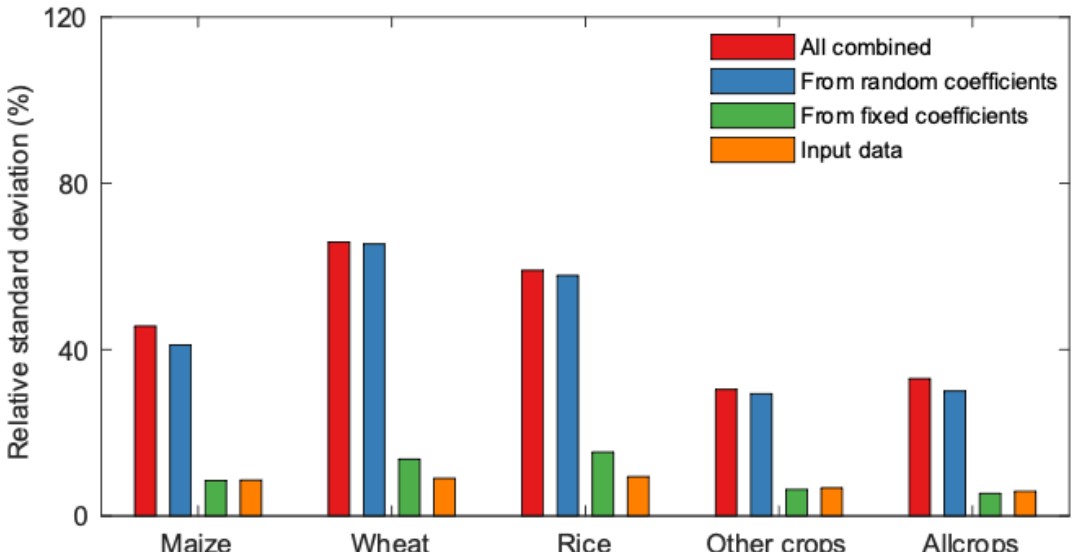

**Figure 20. Relative standard deviation in global cropland-$N_2O$ EF. The figure breaks down the uncertainty of EF per source of uncertainty (i.e., random coefficients, fixed coefficients, input data, or all combined). The uncertainty due to each source can be quantified holding the coefficients for the other sources fixed.**

To address these limitations and reduce uncertainties, concerted efforts should be made to enhance the availability of $N_2O$ observations representing diverse agroecological conditions. Meanwhile, improving the availability of high-precision datasets (e.g., microbial data), and integrating these datasets and the derived underlying mechanisms to our models could also reduce





uncertainties. Currently, most available field N$_2$O observations (see Supplementary Information) are made in Europe, the USA, and China and are scarce in most developing countries (such as Sub-Saharan Africa). Therefore, extending the global coverage of direct and indirect N$_2$O flux measurements to encompass all major agricultural land-use types and climates, land-use changes and management practices and conducting long-term high-frequency monitoring are particularly important to increase the reliability of EFs as well as upscale results from site to regional scales.

### 4.2.3 Uncertainties in estimates of ocean N$_2$O emissions

Global open ocean N$_2$O emissions derived from the ocean biogeochemistry models (Table 1) for the 2010-2019 period are estimated to be 3.5 (2.5 – 4.7) Tg N yr$^{-1}$. All models show the highest emissions associated with equatorial and coastal upwelling zones, as well as the major oxygen minimum zones (OMZs) (e.g., the Eastern Equatorial Pacific and the Arabian Sea region of the northern Indian Ocean, see Figure 21). These are regions characterized by high levels of biological productivity and higher sub-surface organic matter remineralization which results in higher N$_2$O yields in sub-oxic waters. The four participating models capture these characteristics but also show varying degrees of intensity in regional N$_2$O emissions. The models also show good agreement in representing the ocean regions of relatively low N$_2$O ocean-atmosphere fluxes (i.e., open ocean gyres where biological productivity is low).

The spatial distribution of uncertainty in ocean N$_2$O emissions among the models (Figure 21) is similar to that of the net N$_2$O ocean-atmosphere flux, with the highest uncertainties observed in the equatorial upwelling and low-oxygen waters associated with high sub-surface N$_2$O production (Babbin et al. 2020; Ganesan et al. 2020). Largest uncertainties are found in the equatorial Pacific, the Benguela upwelling region of the Atlantic, and the eastern equatorial Indian Ocean. Uncertainties in the ocean models' representation of N$_2$O fluxes result from a range of model characteristics (Zamora and Oschlies, 2014; Martinez-Rey et al. 2015; Buitenhuis et al. 2018; Battaglia and Joos, 2018; Landolfi et al. 2017; Berthet et al. 2023). These include (i) uncertainties in ocean circulation (particularly the representation of upwelling zones and the ocean circulation features (often sub-grid scale) that control the extent and intensity of oxygen-minimum zones (OMZs)); (ii) simulation of ocean organic matter productivity, export production, and mesopelagic remineralization (a driver of the sub-surface source function for N$_2$O production in models); (iii) the model biogeochemical parameterizations representing N$_2$O production and consumption from marine nitrification and denitrification processes, including their dependence on local dissolved oxygen concentrations and thresholds; and (iv) parameterization of ocean-atmosphere gas-exchange fluxes.

Model simulations of oceanic N$_2$O are closely linked to the underlying modeled oxygen distributions, as the embedded biogeochemical parameterizations for N$_2$O include the sensitivity of N$_2$O cycling processes (e.g., nitrification, denitrification) to local oxygen level (Ji et al., 2018). Significant uncertainties in modeled N$_2$O fluxes result from model biases in the representation of dissolved oxygen, especially in low-oxygen zones such as the Eastern Equatorial Pacific (Zamora and Oschlies, 2014; Martinez-Rey et al., 2015). Many ocean model simulations of dissolved oxygen display biases, especially in oxygen-minimum zones critical for N$_2$O cycling (Martinez-Rey et al., 2015). To reduce potential sources of error from model-simulated oxygen, one N$_2$O model in this analysis employs observation-based oxygen distributions when simulating ocean

N$_2$O (Buitenhuis et al., 2018). However, this approach also restricts a model's response to climate-related feedback on ocean oxygen. In addition, the models in this analysis include optimization and calibration of N$_2$O cycle parameters by incorporating constraints from ocean observations (e.g., surface and interior N$_2$O and microbially-mediated process rates) (Battaglia and Joos, 2018, Buitenhuis et al., 2018, Berthet et al., 2023). A more detailed error analysis of N$_2$O model parameters (including

uncertainty in gas-exchange fluxes) in one of the component models (Buitenhuis et al., 2018) suggests estimated uncertainties in global fluxes from biogeochemical parameter specifications of ~33%. Further, a 1,000-member ensemble with 11 parameters varied with one of the models and constrained with both surface and subsurface N$_2$O observations yields an observation-constrained standard deviation of ±36% around the median of 4.3 TgN yr$^{-1}$ (Battaglia and Joos, 2018), consistent with a recent surface pN$_2$O-based estimated of 4.2±1 TgN yr$^{-1}$ (Yang et al., 2020).

Landolfi et al. (2017) also note that uncertainties arise in current model predictions of marine N$_2$O fluxes due to the neglect of feedback from impacts of external nutrient sources and ocean acidification on marine productivity and the ocean nitrogen and oxygen cycles. Reducing uncertainties in model estimates of the evolution of ocean N$_2$O fluxes will require accounting for these impacts in the underlying biogeochemical parameterizations. In addition, due to the high sensitivity of modeled N$_2$O production/consumption rates to oxygen level in the key ocean OMZ zones, an important priority in reducing modeled ocean

N$_2$O flux uncertainties is to achieve a more accurate simulation of the ocean circulation and oxygen distribution of these regions.

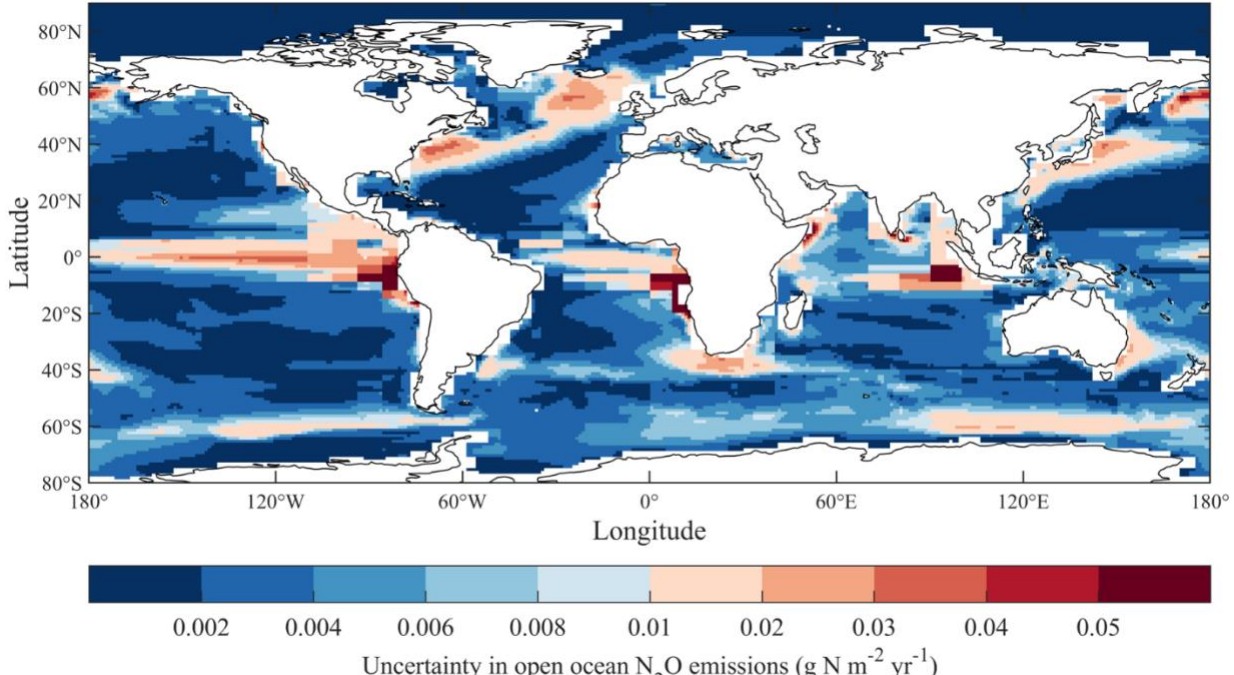

**Figure 21. Spatial uncertainty distribution (one standard deviation) in open ocean N$_2$O emissions in the 2010s. Note that the color scale in this figure is different from that in Figure 19.**



### 4.2.4 Uncertainties in emissions estimates from the continental shelves

Estimates of N$_2$O emissions vary by a factor of 2-3 in the coastal ocean (1 observation-based product and 2 models). The MEM-RF observational estimate (1.63 Tg N yr$^{-1}$, Yang et al., 2020) falls at the high end of the two high-resolution model estimates (1.39 and 0.61 Tg N yr$^{-1}$ for CNRM-0.25° and ECCO-Darwin, respectively). Shelf N$_2$O flux emissions from MEM-RF, CNRM-0.25°, and ECO-Darwin broadly agree in the main patterns and magnitude. Emission hotspots in productive, low-O$_2$ upwelling systems (e.g., eastern boundary upwellings, upwellings of the north-western Indian Ocean) appear to be underestimated by models. Lower emissions in models likely reflect the inability of models to resolve complex near-shore dynamical circulation and biogeochemical processes key to the production, transport, and evasion of N$_2$O. This includes under-resolved dynamics in upwelling systems and shallow oxygen minimum zones with high N$_2$O emissions (Resplandy et al., 2023), strong spatial gradients introduced by patterns of high production/ high remineralization and enhanced land-sea inputs of N in shallow shelves (e.g., Baltic Sea, Southeast and East Asia), sedimentary processes, and production in estuarine and coastal vegetated ecosystems, which is subsequently transported offshore. Conversely, our ability to reconstruct spatial patterns in N$_2$O air-sea fluxes from observations (MEM-RF, Yang et al., 2020), in particular along continental margins, is severely limited by the number of N$_2$O observations, which is two orders of magnitude smaller than for CO$_2$. Observations tend to be localized in regions of strong air-sea disequilibrium and might thus be biased high (e.g., Babbin et al. 2020; Ganesan et al, 2020). In addition, many coastal regions remain undersampled, further limiting the performance of MEM-RF. For instance, models point to coastal N$_2$O flux hotspots along mid-latitude western boundaries (e.g., the US east coast, the North Pacific east of Japan, the southeast coast of Australia, and the south-eastern tip of Africa) that are not diagnosed in the observational product (Resplandy et al., 2023). Furthermore, N$_2$O fluxes are highly spatially heterogeneous (scales of 1 to 100 km) due to land-ocean gradients and mesoscale and sub-mesoscale features such as eddies (Arévalo-Martínez et al., 2017, 2019; Yang et al., 2020, Grundle et al., 2017). Eddies are instrumental in setting suboxic conditions favorable for N$_2$O production, and it has been suggested that N$_2$O production weakens within eddies during their transit across the shelf and further offshore (Arévalo Martinez et al., 2016). These small-scale circulation features are important controls for N$_2$O dynamics but are poorly accounted for in data-based reconstructions and models.

This assessment provides the most up-to-date estimate of N$_2$O climatological emissions from the global shelves, but the variability of these emissions remains uncertain. Each product covers a different time period and only provides limited or missing information on seasonal fluctuations, inter-annual variability and long-term trends. For instance, only a handful of observations per year are available in most regions, providing a limited picture of seasonality, and even more limited information on interannual variability (e.g., El Nino-Southern Oscillation, Pacific Decadal Oscillation) and global longer-term trends. Disentangling such influences from limited observations alone remains a major challenge. The effects of extreme events on N$_2$O fluxes such as storms and marine heat waves are also currently not captured, and the intra-annual variability in hotspot regions such as coastal upwelling systems remains poorly constrained. Despite these limitations, data-based reconstructions





and models suggest a vigorous seasonal cycle and, potentially, important variability on interannual timescales (Yang et al., 2020, Ganesan et al., 2020). The development of a Global $N_2O$ Ocean Observation Network ($N_2O$-ON) (Bange et al., 2019; Bange, 2022) is critically needed to better resolve spatio-temporal patterns and reduce uncertainties in $N_2O$ emissions. Increasing the density of observations in regions of high $N_2O$ disequilibrium and collecting long time-series of $N_2O$ measurements will allow a better characterization of interannual changes and their dynamics. Meanwhile, algorithmic approaches that address the observational limitations should be developed and refined to extrapolate $N_2O$ measurements to global and interannual timescales, leveraging advancements made for $CO_2$ disequilibrium and flux reconstructions.

Parallel efforts based on the development of mechanistic models are also needed to strengthen our understanding of the dynamics underlying interannual $N_2O$ flux variability and to detect and attribute long-term anthropogenic effects. However, the representation of $N_2O$ processes in biogeochemical models remains limited, and very few climate models include marine emissions of $N_2O$ fluxes (only 4 out of 26 CMIP6 models considered in Séférian et al, 2020). Uncertainty persists regarding the various (micro) biological processes that drive $N_2O$ cycling in coastal waters and sediments (Bange, 2022). Current global

ocean biogeochemical models typically adopt an indirect representation of $N_2O$ production, which is diagnosed from environmental conditions (e.g., temperature) and $O_2$ consumption during remineralization of organic matter, without explicitly representing the bacterial pools and chemical reactions responsible for $N_2O$ production in suboxic waters (e.g., Aumont et al., 2015, Battaglia and Joos, 2018). In addition, key aspects of air-sea $N_2O$ exchange, such as the effects of surfactants in the sea surface microlayer (Kock et al., 2012) remain poorly understood. Finally, the interannual variability of $N_2O$ fluxes and its

attribution to climatic and anthropogenic drivers is largely unknown. Disentangling these influences will benefit from (1) interannually varying observational $N_2O$ flux reconstructions at scales fine enough to capture high emissions along continental margins; (2) statistical methods that address the limited number of observations in space and time; and (3) $N_2O$ cycle simulations with forward mechanistic models. A blueprint for this work already exists with the approaches developed by the oceanic $CO_2$ community (Gruber et al., 2022). Similar approaches would enable attribution of $N_2O$ flux changes to specific

drivers, leading to better predictability.

### 4.2.5 Uncertainties in emissions estimates from atmospheric inversions

The four atmospheric inversion frameworks show uncertainties in the estimates of $N_2O$ emissions, especially in hotspot regions such as Eastern China, India, Europe, the US Corn Belt, and Northern South America (Figure 22). The uncertainties in inversion estimates are mainly from errors in the modeled atmospheric transport, the dependence on the prior information, and

the availability of atmospheric observations. Every inversion framework in this study used a different atmospheric transport model with different horizontal and vertical resolutions (Table 1). By including estimates from multiple inversion frameworks with different modeled atmospheric transport, the systematic error can be assessed to some extent. The inversion estimates are dependent on the spatial pattern and magnitude of the prior flux estimates to an extent that is determined by the density of the observations. Using the same prior information might reduce the range in the atmospheric inversion estimates but not the

uncertainty since this depends on the spatiotemporal density of the atmospheric observations and the accuracy of the modeled

transport. The uncertainty reduction (calculated as one minus the ratio of the posterior to prior uncertainty) indicates the degree of constraint on the inversion estimates (Figure 23). It shows that the areas of South America, Africa, central and southern Asia as well as Australasia are poorly constrained by observations. The relatively sparse distribution of current $N_2O$ observation sites underscores the necessity of establishing more sites and regular aircraft profiles, especially in tropical and

sub-tropical regions, to better constrain inversion models and to further reduce the posterior uncertainty.

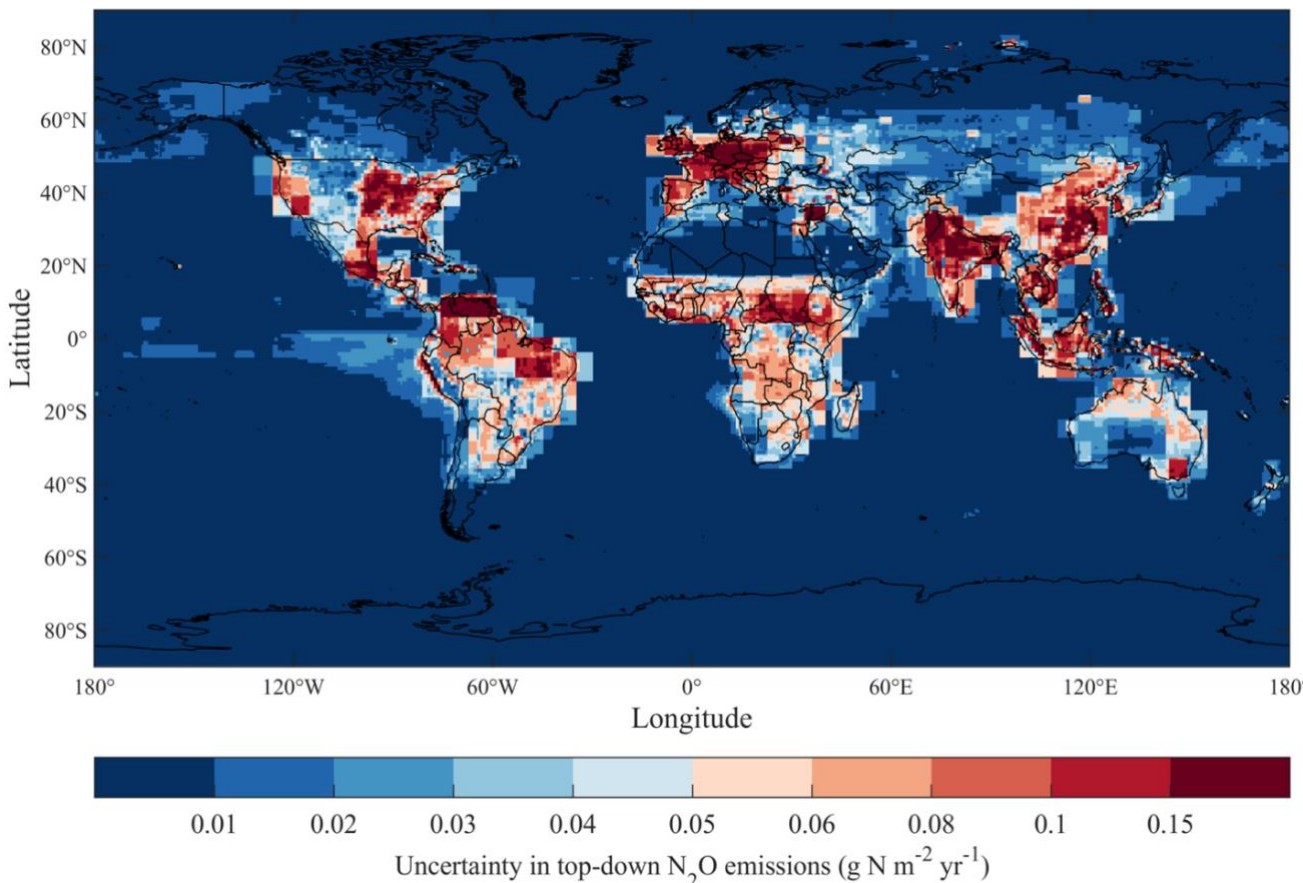

**Figure 22. Spatial distribution of posterior uncertainty (one standard deviation) in TD model estimates of $N_2O$ emissions in the 2010s.**

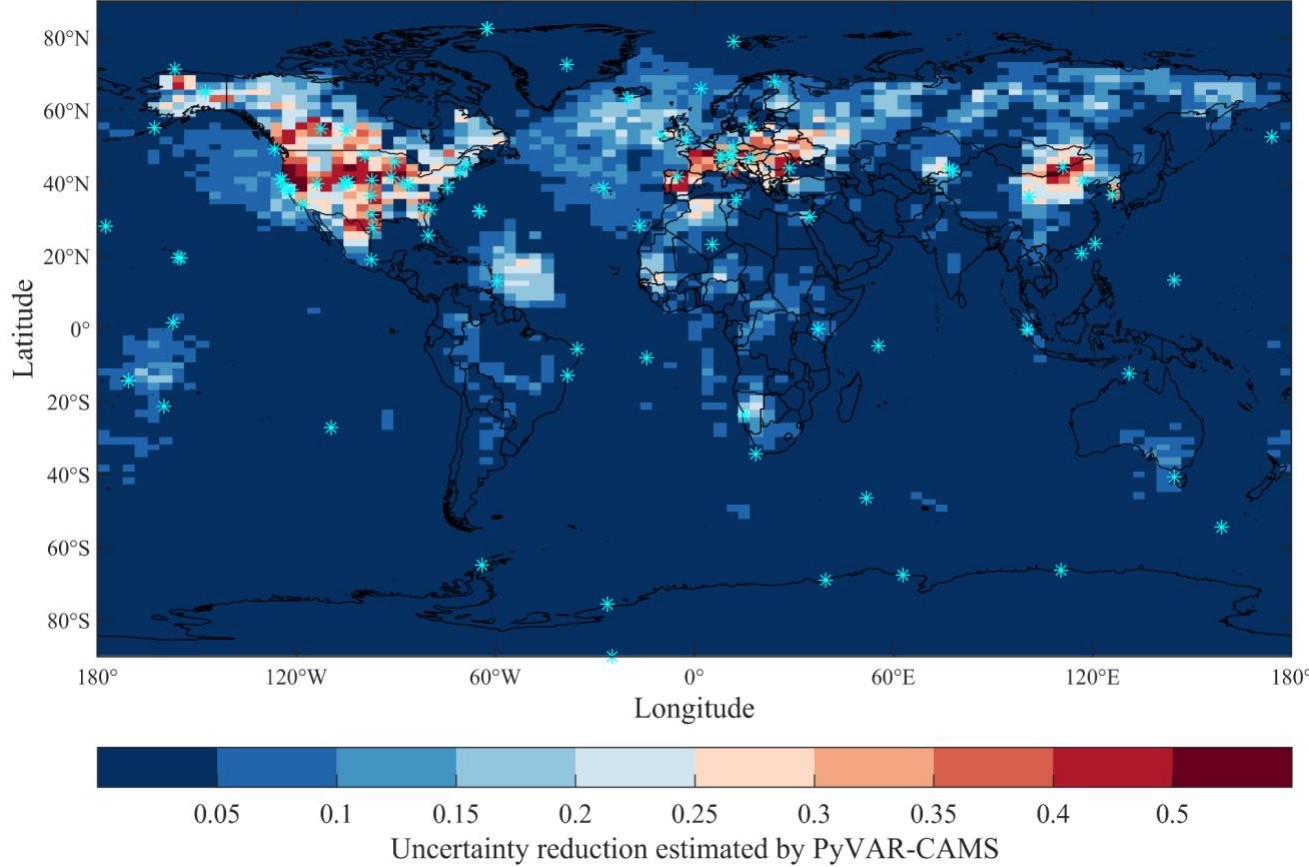

**Figure 23. Uncertainty reduction (1 - sigma_posterior/sigma_prior) from the PyVAR-CAMS inversion framework. *Atmospheric observational stations used in the inversion framework.**

### 4.2.6 Other missing fluxes

We recognize that $N_2O$ emissions contributed by termites could be a significant natural source in tropical and subtropical ecosystems (Brümmer et al., 2009; Miambi et al., 2022). The metabolic activity of microbial symbionts in the termite gut can maintain steep oxygen gradients, which facilitates nitrification and denitrification processes and the production of $N_2O$ (Brauman et al., 2015; Brune et al., 1995). Nevertheless, termites have a wide trophic diversity, and their $N_2O$ emission rates vary significantly, with some species creating emission hotspots (Brümmer et al., 2009), while others function as net sinks (Majeed et al., 2012). Feeding habits and the abundance of nitrifiers and denitrifiers in the gut are reported to be the key factors determining net $N_2O$ emission of termites (Brauman et al., 2015; Miambi et al., 2022). Termites that consume N-rich material, such as soil organic matter and fungi, exhibit high $N_2O$ production rates and emit $N_2O$ into the atmosphere, while those feeding on N-deficient wood can consume atmospheric $N_2O$ (Brauman et al., 2015). It is difficult to scale up calculations of net $N_2O$ emission by termites due to the lack of data on their abundance and biomass across global ecosystems, therefore our





understanding of the precise contribution of termites to the atmospheric N$_2$O budget on a global scale remains limited and not

considered in our analysis

## 5 Data Availability

The accompanying database includes two Excel files and 27 txt files. The two Excel files are organized into the following spreadsheets.

The Global N$_2$O Budget 1980-2020: Global emission data includes the following items:

1. Summary.
2. Bottom-up estimates: global BU N$_2$O budget from 1980 to 2020, including 20 individual sources and sinks.
3. Top-down estimates: N$_2$O emissions from land, ocean, and global during 1997-2020 estimated by the four atmospheric inversion models.
4. Atmospheric_Chemical_sink: Global atmospheric chemical sink estimated by the four atmospheric inversion models
(1997-2020) and one satellite and photolysis model (2005-2020).
5. N$_2$O_dry_mole_fraction: Monthly N$_2$O dry mole fraction and its growth rate during 2000-2020 estimated by the three observation networks.
6. Future_N$_2$O_dry_mole_fraction: the projected N$_2$O dry mole fractions from the four illustrative Representative Concentration Pathways (RCPs) in the IPCC Fifth Assessment Report (2000-2050), and the seven illustrative
Socioeconomic Pathways (SSPs) used in CMIP6 (2005-2050);

The Global N$_2$O Budget 1980-2020: Regional emission data includes the following items:

1. Summary.
2. Anthropogenic_sectors_1980_2020: N$_2$O emissions from the four anthropogenic sources for the 18 regions during 1980-2020.
3. Bottom-up_estimates: Total N$_2$O emissions from the 18 regions during 1980-2020 estimated by BU approaches.
4. Top- down estimates: N$_2$O emissions from the 18 regions during 1997-2020 estimated by the four atmospheric inversion models.
5. Decadal_mean_2010s: regional N$_2$O emissions estimated by the TD and BU approaches in the 2010s.

Global N$_2$O Budget 1980-2020: modelled gridded emission data includes the spatial patterns of N$_2$O emissions from different
sources (unit: gN/m$^2$/yr) estimated by different models as follows:

1. NMIP2: total 16 maps showing the spatial distribution of soil N$_2$O emissions, including estimates of eight process-based models participated in NMIP2 (CLASSIC, DLEM, ELM, ISAM, LPX-Bern, O-CN, ORCHIDEE, and VISIT) and two periods (the1850s and 2010s).
2. Open ocean emissions: total 4 maps showing the spatial distribution of open ocean N$_2$O emissions, including estimates
of four ocean models: Bern-3D, NEMO-PlankTOM10.2, NEMOv3.6-PISCESv2-gas, and UVic2.9.
3. Shelf emissions: total 3 maps showing the spatial distribution of continental shelves N$_2$O emissions, including estimates of three products: CNRM, ECCO, MEM-RF.
4. Top-Down estimates: total 4 maps showing the global distribution of N$_2$O emissions, including estimates of four atmospheric inversion models: GEOSChem, INVICAT, MIROC4-ACTM, and PyVAR-CAMS.

The data presented in this work can be downloaded from https://doi.org/10.18160/RQ8P-2Z4R (Tian et al. 2023).





## Appendix A: Supplementary tables

**Table A1. Comparison of terminologies used in this study and previous reports.**

| GCP Terminology (this study) | | IPCC AR6 (IPCC, 2021) | National GHG inventories (used by UNFCCC according to IPCC, 2006 and IPCC, 2019) | UNFCCC / IPCC 2006 Source sector |
|---|---|---|---|---|
| *Anthropogenic sources* | | | | |
| Direct emissions of N additions in the agricultural sector (Agriculture) | Direct soil emissions (mineral N and manure fertilization, cultivation of organic soils, and crop residue returns) | Agriculture | Direct $N_2O$ emissions from managed soils (except due to grazing animals) | part of 3C4 |
| | Manure left on pasture | | Urine and dung deposited by grazing animals | part of 3C4 |
| | Manure management | | Manure management | 2A2 |
| | Aquaculture | --- | --- | --- |
| Other direct anthropogenic sources | Fossil fuel and industry | Fossil fuel combustion and industrial processes | Energy and industrial processes | 1, 2 |
| | Waste and wastewater | Human excreta | Waste | 4C1, 4C2 4D1, 4D2 |
| | Biomass burning (from crop residue, grassland, shrubland and savannas; peat fires, tropical forests, boreal forests, and temperate forests) | Biomass and biofuel burning | Prescribed burning of savannas, field burning of agricultural residues | 3E, 3F |
| Indirect emissions from anthropogenic N additions | Inland and coastal waters (rivers, lakes, reservoirs, estuaries, and coastal zones) | Rivers, estuaries, coastal zones | Indirect emissions due to leaching and runoff | part of 3C5, 3C6 |
| | Atmospheric N deposition on land | Atmospheric deposition on land | Indirect emissions due to atmospheric deposition (of agricultural as well as other anthropogenic compounds emitted) | part of 3C5, 5A |
| | Atmospheric N deposition on ocean | Atmospheric deposition on ocean | | part of 3C5, 5A |
| Perturbed fluxes from climate/$CO_2$/land cover change | $CO_2$ effect | --- | --- | --- |
| | Climate effect | --- | --- | --- |
| | Post-deforestation pulse effect | --- | --- | --- |
| | Long-term effect of reduced mature forest area | --- | --- | --- |
| *Natural sources and sinks* | | | | |
| Natural soils baseline | | Soils under natural vegetation | --- | --- |
| Coastal and Open Ocean baseline | | Oceans | --- | --- |



| | | | |
|---|---|---|---|
| Natural (rivers, lakes, reservoirs, estuaries, and coastal vegetation) | --- | --- | --- |
| Lightning and atmospheric production | Lightning | --- | --- |
| | Atmospheric chemistry | --- | --- |
| Soil/wetland surface sink | Surface sink | --- | --- |
| Atmospheric sink | Atmospheric sink | | |





**Table A2. List of the countries used to define the 18 regions.**

| Region num. | Region name | Countries or territories |
|---|---|---|
| 1 | USA | USA with Alaska, Bermuda Islands |
| 2 | Canada | Canada |
| 3 | Central America | Anguilla, Antigua and Barbuda, Bahamas, Barbados, Belize, British Virgin Islands, Cayman Islands, Costa Rica, Cuba, Dominica, Dominican Republic, El Salvador, Guadeloupe, Guatemala, Honduras, Jamaica, Martinique, Mexico, Montserrat, Nicaragua, Panama, Puerto Rico, Saint Kitts and Nevis, Saint Lucia, Saint Vincent and the Grenadines, Turks and Caicos Islands, United States Virgin Islands |
| 4 | Brazil | Brazil |
| 5 | Northern South America | Aruba, Colombia, French Guiana, Grenada, Guyana, , Suriname , Trinidad and Tobago, Venezuela |
| 6 | Southwest South America | Argentina, Bolivia, Chile, Ecuador, Peru, Falkland Islands (Malvinas), Paraguay, Uruguay |
| 7 | Europe | Albania, Andorra, Austria, Belarus, Belgium, Belgium, Luxembourg, Bulgaria, Channel Islands, Croatia, Cyprus, Czech Republic, Denmark, Estonia, Faroe Islands, Finland, France, Germany, Gibraltar, Greece, Greenland, Hungary, Iceland, Ireland, Isle of Man, Italy, Latvia, Liechtenstein, Lithuania, Luxembourg, Malta, Montenegro, Netherlands, Norway, Poland, Portugal, Republic of Moldova, Romania, Serbia, Slovakia, Slovenia, Spain, Sweden, United Kingdom, Ukraine |
| 8 | Northern Africa | Algeria, Cabo Verde, Chad, Côte d'Ivoire, Djibouti, Egypt, Eritrea, Ethiopia, Ethiopia PDR, Gambia, Guinea, Guinea-Bissau, Libya, Mali, Mauritania, Morocco, Saint Helena Ascension and Tristan da Cunha, Sao Tome and Principe, Senegal, Somalia, Sudan former, Tunisia, Western Sahara |
| 9 | Equatorial Africa | Benin, Burkina Faso, Burundi, Cameroon, Central African Republic, Congo, Democratic Republic of the Congo, Equatorial Guinea, Gabon, Ghana, Liberia, Nigeria, Rwanda, Sierra Leone, Togo, Uganda, United Republic of Tanzania, |
| 10 | Southern Africa | Angola, Botswana, Comoros, Lesotho, Madagascar, Malawi, Mauritius, Mayotte, Mozambique, Namibia, Reunion, Seychelles, South Africa, Swaziland, Zambia, Zimbabwe |
| 11 | Russia | Russian federation |
| 12 | Central Asia | Kazakhstan, Kyrgyzstan, Tajikistan, Turkmenistan, Uzbekistan, Mongolia, |
| 13 | Middle East | Armenia, Azerbaijan, Bahrain, People's Republic of Georgia, Iran, Iraq, Israel, Jordan, Kuwait, Lebanon, Occupied Palestinian Territory, Oman, Qatar, Saudi Arabia, Syrian Arab Republic, Turkey, United Arab Emirates, Yemen |
| 14 | China | China mainland, Macao, Hong Kong, Taiwan |
| 15 | Korea and Japan | Japan, Korea, Republic of Korea |
| 16 | South Asia | Afghanistan, Bangladesh, Bhutan, India, Nepal, Pakistan, Sri Lanka |
| 17 | South East Asia | Brunei Darussalam, Cambodia, Guam, Indonesia Kiribati, Lao People's Democratic Republic, Malaysia, Maldives, Marshall Islands, Myanmar, Nauru, Northern Mariana Islands, Palau, Philippines, Singapore, Solomon Islands, Thailand, Timor-Leste, Tokelau, Viet Nam |
| 18 | Oceania | American Samoa, Australia, Cook Islands, Fiji, French Polynesia, New Caledonia, New Zealand, Niue, Norfolk Island, Pacific Islands Trust Territory, Papua New Guinea, Pitcairn Islands, Samoa, Tonga,Tuvalu,Vanuatu, Wallis and Futuna Islands |





**Table A3. The sectors in N$_2$O budget and its sources. (Sector with "*" means this sector only include maximum, mean, and minimum).**

| ID | N$_2$O budget sectors (Global scale) | Sources |
|---|---|---|
| 1 | Aquaculture | EF0.5, EF5, EF1.8 |
| 2 | Manure left on pasture | DLEM, EDGAR, FAO |
| 3 | Manure management | EDGAR |
| 4 | Direct soil emissions global | EDGAR, FAO, NMIP2/DLEM, SRNM/DLEM |
| 5 | Inland water, estuaries and coastal vegetation anthropogenic | Meta-analysis and Process-based models, EDGAR, FAO |
| 6 | N deposition on land | EDGAR/NMIP2, NMIP2 |
| 7 | CO$_2$ | CLASSIC, DLEM, ELM, ISAM, LPX-Bern, OCN, ORCHIDEE, VISIT |
| 8 | Climate | CLASSIC, DLEM, ELM, ISAM, LPX-Bern, OCN, ORCHIDEE, VISIT |
| 9 | Post deforestation pulse effect | DLEM, Book-keeping model |
| 10 | Natural soils baseline | CLASSIC, DLEM, ELM, ISAM, LPX-Bern, OCN, ORCHIDEE, VISIT |
| 11 | Open ocean | BERN, CNRM, UViC, UEA-NEMO-PlankTOM |
| 12 | N deposition on ocean[*] | Parvadha Suntharalingam et al. (2012) |
| 13 | Biomass burning | FAO, DLEM, GFED |
| 14 | Fossil fuel industry | EDGAR, EDGAR/UNFCCC |
| 15 | Waste and wastewater | EDGAR/UNFCCC |
| 16 | Inland water, estuaries and coastal vegetation natural[*] | DLEM, stochastic mechanistic model, RF model, meta-analyses-based estimates |
| 17 | Lightning and atmospheric production[*] | Schlesinger (2013) and Syakila, Kroeze, and Slomp (2010) |
| 18 | Long term reduction effect | DLEM, Book-keeping model |
| 19 | Continental shelves[*] | ECCO, CNRM, MEM-RF |





**Table A4. Simulation design of NMIP2.**

| Historical | Climate | CO₂ | Land cover | Irrigation | Ndep | Nfer | ManureN |
|---|---|---|---|---|---|---|---|
| SH0 | 1901-1920 | 1850 | 1850 | 1850 | 1850 | 1850 | 1850 |
| SH1 | • | • | • | • | • | • | • |
| SH2 | • | • | • | • | • | • | 1850 |
| SH3 | • | • | • | • | • | 1850 | • |
| SH4 | • | • | • | • | 1850 | • | • |
| SH5 | • | • | • | 1850 | • | • | • |
| SH6 | • | • | 1850 | • | • | • | • |
| SH7 | • | 1850 | • | • | • | • | • |
| SH8 | 1901-1920 | • | • | • | • | • | • |
| SH9 | 1901-1920 | 1850 | 1850 | 1850 | 1850 | • | • |
| SH10 | • | 1850 | 1850 | 1850 | 1850 | 1850 | 1850 |
| SH11 | • | • | 1850 | 1850 | • | 1850 | 1850 |
| SH12 | • | • | • | 1850 | • | 1850 | 1850 |

*Note: For historical simulations, "•" indicates the forcing during 1850-2020 is included in the simulation, "1901-1920" indicates the 20-year mean climate condition during 1901-1920 will be used over the entire simulation period, and "1850" indicates the forcing will be fixed in 1850 over the entire period. Climate data is only available from 1901; we assume the 20-yr average value between 1901 and 1920 for the years 1850-1900. N deposition is available only from 1850. Manure N is*

*available only from 1860; we assume manure N at the 1860 value for years 1850-1860. N fertilizer before 1910 was zero.*



**Table A5. Funding supporting the production of the various components of the global nitrous oxide budget in addition to the authors' supporting institutions (see also Acknowledgements).**

| Funder and grant number (where relevant) | Authors/simulations/ observations |
|---|---|
| Australian National Environmental Science Program - Climate Systems Hub | Josep G. Canadell |
| Deutsche Forschungsgemeinschaft (DFG) (grant no. SFB754/3 B1 D1807) | Angela Landolfi |
| Dutch Ministry of Education, Culture and Science through the Netherlands Earth System Science Center (NESSC) | Junjie Wang |
| European Space Agency (ESA) RECCAP2 project (grant no. ESRIN/4000123002/18/I-NB) | Philippe Ciais |
| European Union's Horizon 2020 research and innovation programme under Grant Agreement N° 101003536 (ESM2025 – Earth System Models for the Future) | Pierre Regnier, Sönke Zaehle, Nicolas Vuichard, Sarah Berthet |
| European Union's Horizon 2020 research and innovation programme under the Marie Słodowska-Curie grant agreement no. 101030750 | Luke M. Western |
| European Union's Horizon Europe Research and Innovation Programme under Grant Agreement N° 101081395 (EYE-CLIMA) | Glen P. Peters |
| EYE-CLIMA, a project funded under the European Union's Horizon Europe Research and Innovation programme under grant agreement number 101081395 | Wilfried Winiwarter |
| French state aid, managed by ANR under the "Investissements d'avenir" programme (ANR-16-CONV-0003) | Ronny Lauerwald |
| Hatch Act (Accession Number IDA01722) through the USDA National Institute of Food and Agriculture | Daniele Tonina |
| Hutchinson Postdoctoral Fellowship from the Yale Institute for Biospheric Studies at Yale University | Judith A. Rosentreter |
| Member countries to FAOSTAT through the FAO's Regular Budget. | Francesco N. Tubiello |
| MIROC4-ACTM from the Environment Research and Technology Development Fund (SII-8; grant no. JP-MEERF21S20800) and the Arctic Challenge for Sustainability phase II (ArCS-II; grant no. JP- MXD1420318865) project | Prabir K. Patra |
| National Natural Science Foundation of China (grant no. 42107393) | Minpeng Hu |
| National Natural Science Foundation of China (grant no. 42225102; 41977082) | Feng Zhou |



| | |
|---|---|
| Natural Environment Research Council through its grants to the UK National Centre for Earth Observation (NCEO; NERC grant numbers NE/R016518/1 and NE/N018079/1) | Chris Wilson |
| Swiss National Science Foundation (200020_200511) | Fortunat Joos, Aurich Jeltsch-Thoemmes, Qing Sun |
| U.S. Department of Energy through the Reducing Uncertainties in Biogeochemical Interactions through Synthesis andComputation Scientific Focus Area (RUBISCO SFA) project | Qing Zhu |
| U.S. National Science foundation (grant no. 1903722) | Hanqin Tian, Shufen Pan, Chaoqun Lu |
| U.S. National Science Foundation (grant no. OCE-1847687). | Daniele Bianchi |
| US Department of Agriculture CBG (grant no. TENX12899) | Hanqin Tian |
| US National Science Foundation (grant no. 1922687) | Shufen Pan |
| | |
| Computing Resources | |
| Computational resources from the Expanse system at the San Diego Supercomputer Center through allocation TG-OCE170017 from the Advanced Cyber infrastructure Coordination Ecosystem: Services and Support (ACCESS) program, which is supported by National Science Foundation grants 2138259, 2138286, 2138307, 2137603, and 2138296. | Daniele Bianchi |
| Computing resources from LSCE | Rona Thompson |
| Computing Resources from Auburn University and Boston College | Hanqin Tian, Shufen Pan |
| | |
| Support for atmospheric observations | |
| Copernicus Atmosphere Monitoring Service (https://atmosphere.copernicus.eu/), implemented by ECMWF on behalf of the European Commission | Rona Thompson |
| CSIRO for long-term support for the operation and maintenance of CSIRO GASLAB and flaks network, the Australian Bureau of Meteorology, Australian Institute of Marine Science, Australian Antarctic Division, NOAA USA, and Environment & Climate Change Canada | CSIRO flask network, Paul B. Krummel |



| | |
|---|---|
| NOAA's Climate Program Office under the Atmospheric Chemistry Carbon Cycle and Climate (AC4) theme. | NOAA observational network, Xin Lan, Geoffrey Dutton |
| U.S. NASA Upper Atmospheric Research Program in the United States with grants NNX07AE89G and NNX16AC98G and 80NSSC21K1369 to MIT and NNX07AF09G, NNX07AE87G, NNX16AC96G, NNX16AC97G and 80NSSC21K1210 and 80NSSC21K1201 to SIO. NASA award to MIT with sub-award to University of Bristol for Mace Head and Barbados (80NSSC21K1369). NASA award to MIT with sub-award to CSIRO for Cape Grim (80NSSC21K1369). U.K. Department for Energy Security & Industrial Strategy (BEIS) (contract 1028/06/2015). U.S. NOAA (contract 1305M319CNRMJ0028). | AGAGE flask network, Jens Mühle |

**Appendix B: Supplementary figures**

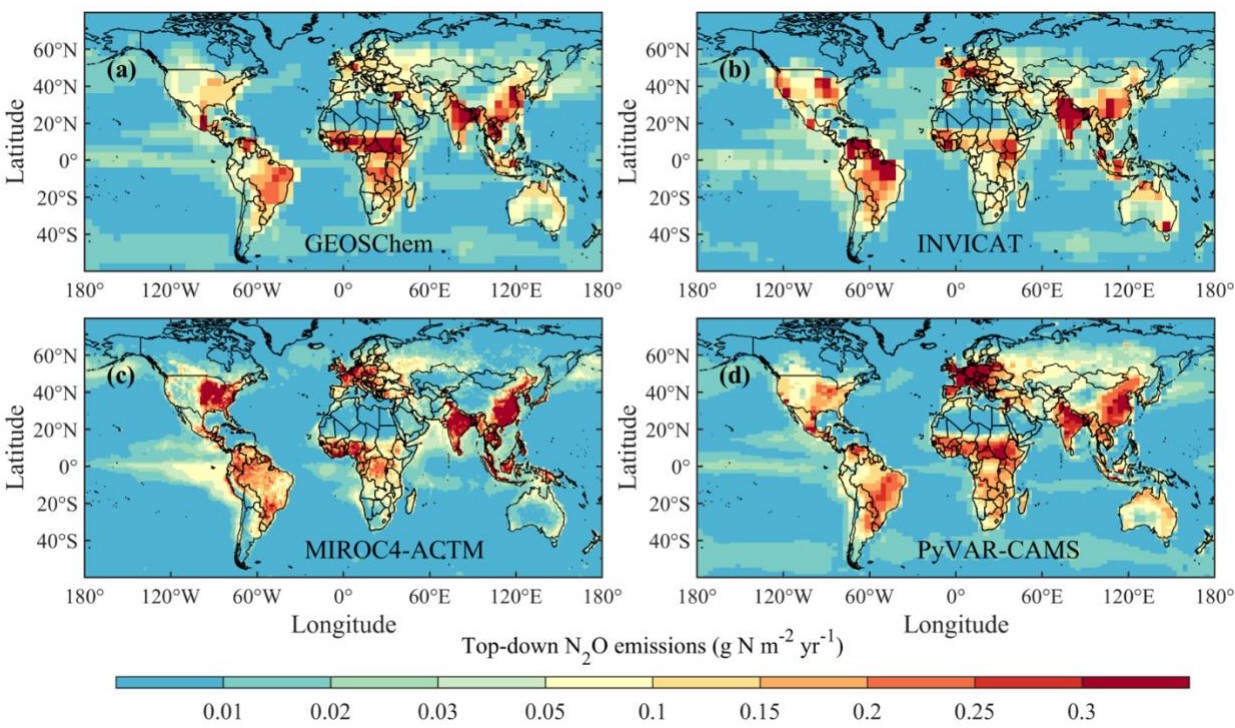

**Figure B1. Spatial distribution of global N$_2$O emissions in the 2010s estimated by different atmospheric inversion frameworks (top-down approach).**
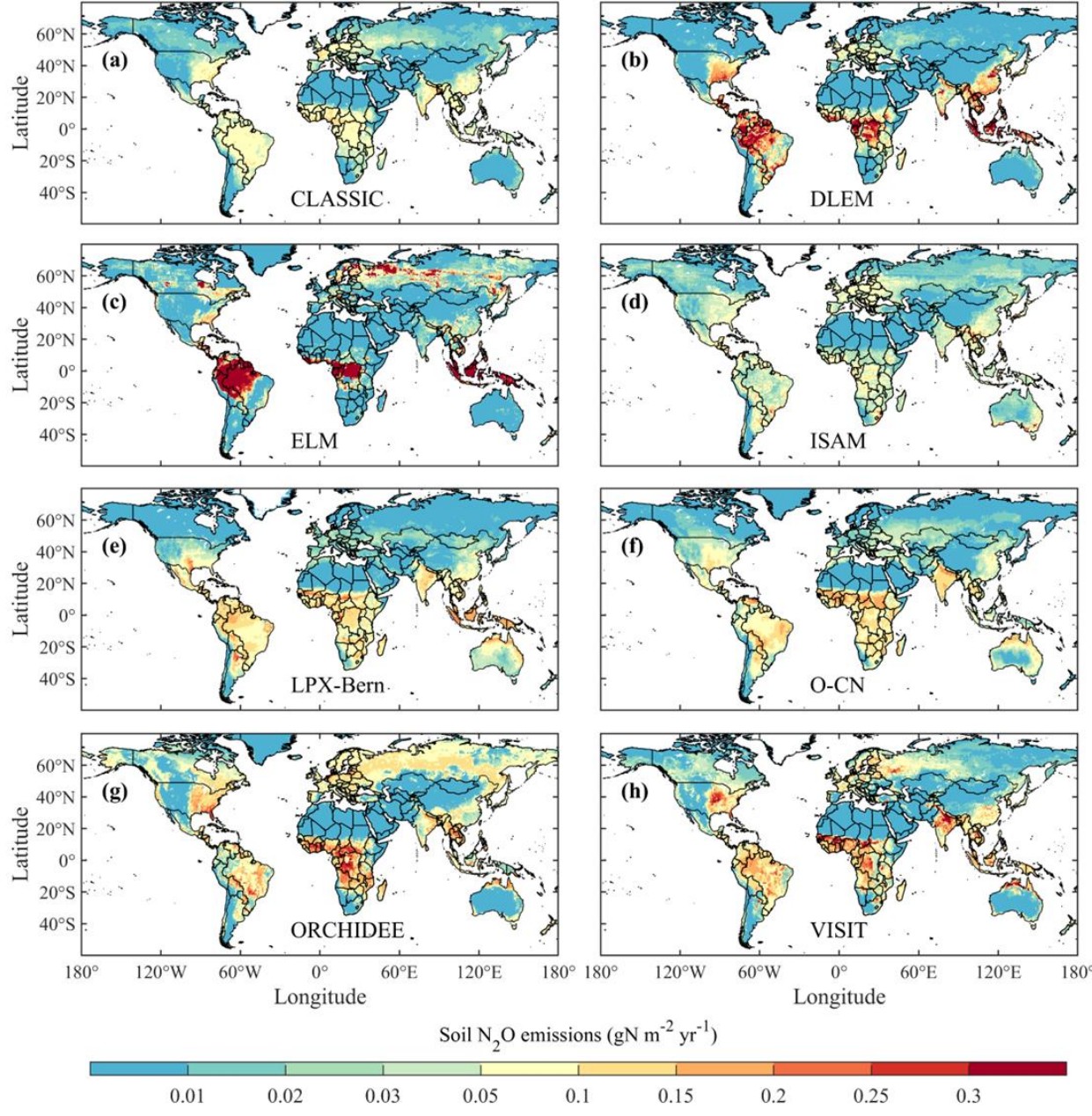

**Figure B2. Spatial distribution of pre-industrial (1850s) soil N$_2$O emissions estimated by different NMIP2 terrestrial biosphere models.**





**Figure B3. Spatial-temporal changes in fertilizer N and manure N applications and atmospheric N deposition to global terrestrial ecosystems derived from HaNi data set (Tian et al. 2022), which were used to drive NMIP2 terrestrial biosphere models.**





**Figure B4. Spatial distribution of soil N₂O emissions during 2010-2019 estimated by NMIP2 terrestrial biosphere models.**



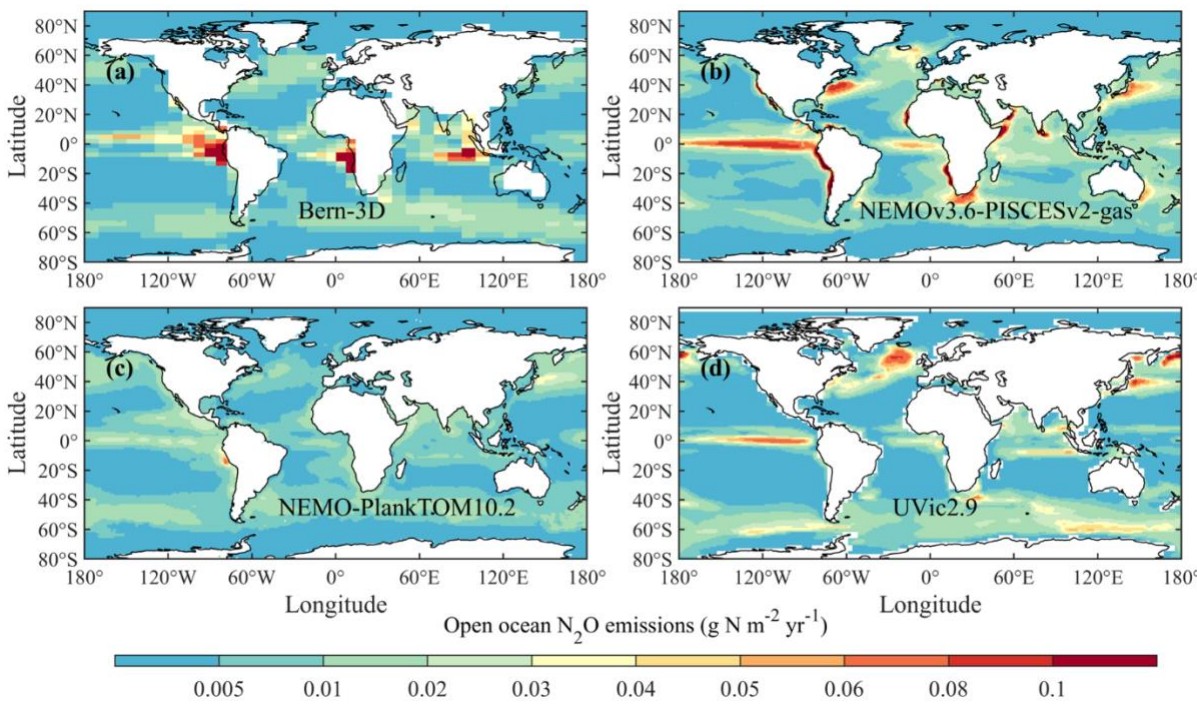


**Figure B5. Spatial distribution of N₂O emissions from open oceans during 2010-2019 estimated by different ocean biogeochemistry models/Earth System models.**




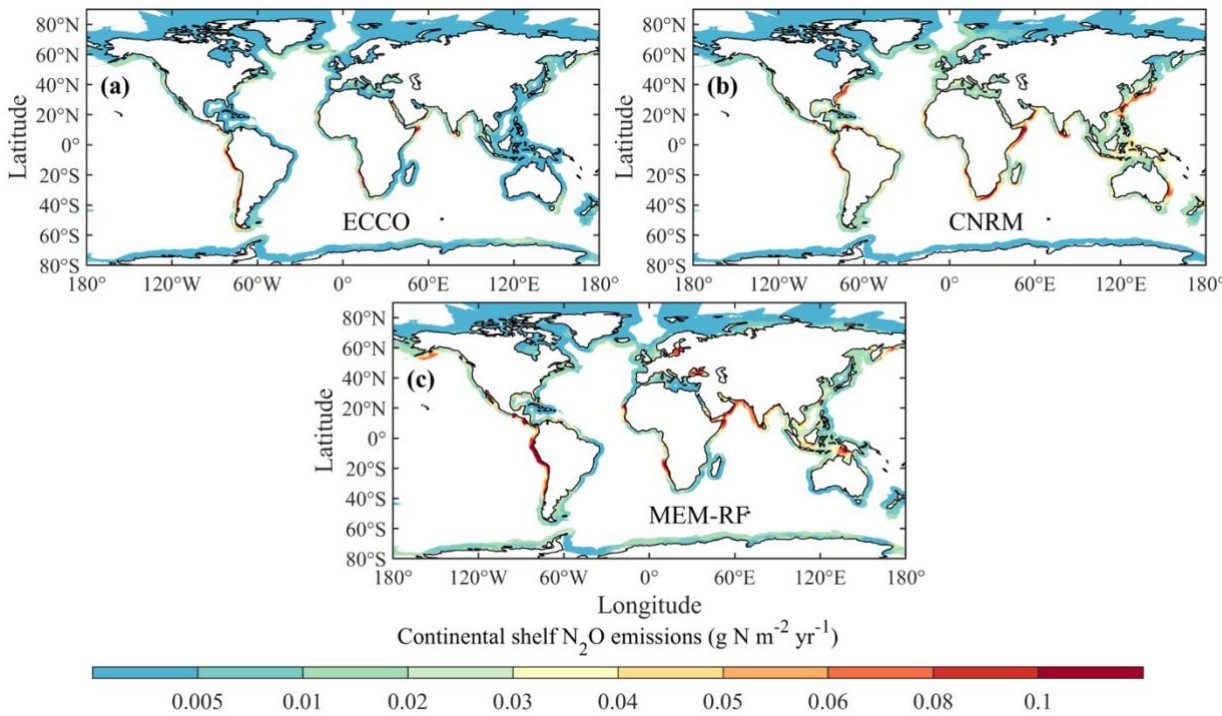

**Figure B6.** N₂O emission from continental shelves as estimated by three methods.



**Supplement**

The supplement related to this article is available online at: https://doi.org/ ************.

**Author contributions**

H.T., R.L.T., and J.G.C. designed and coordinated the study. H.T., N.P., S.P., Y.L., R.L.T., P.S., P.R. gathered the BU and TD data sets and performed the post-processing, analysis and synthesis. H.T., N.P., R.L.T., J.G.C., P.S., P.R., E.A.D., M.J.P., P.C., M.M., S.P., W.W., S.Z., F.Z., and R.B.J. wrote the paper. R.L.T. led atmospheric inversions teaming with P.K.P., K.C.W., D.B.M. and C.W.; H.T. led land biosphere modeling teaming with N.P., S.P., S.Z., A.I, A.K.J., F.J., S.K.G., C.L., H.S., Q.S., and Q.Z.; P.S. led ocean biogeochemical modeling teaming with E.B., A.L., S.B., A.J.-T. and F.J; P.R. led the synthesis of LAOC (Land-Aquatic Ocean Continuum) teaming with R.L.,T.M., Y.Y., M.H.,,P.R., J.R., L.R., M.M., S.B., H.B., D.B., and H.T.; J.W. and L.B. provided data of $N_2O$ flux from aquaculture. G.R.W. and J.Y. provided data of $N_2O$ emissions from biomass burning. F.Z. provided cropland $N_2O$ flux data from a statistical model and field observations. M.M., F.N.T. and W.W. provided $N_2O$ inventory data. M.J.P. provided data of stratospheric and tropospheric sinks. G.P. provided RCP and SSP scenarios data and analysis. X.L. and G.D. provided a global $N_2O$ monitoring dataset of NOAA/ESRL GMD. J.M. and L.M.W. provided a global $N_2O$ monitoring dataset of AGAGE stations. P.K. provided a global $N_2O$ monitoring dataset of CSIRO. All coauthors reviewed and commented on the manuscript.

**Competing interests**

At least one of the (co-)authors is a member of the editorial board of *Earth System Science Data*.

**Acknowledgements**

This paper is the result of a collaborative international effort under the umbrella of the Global Carbon Project (a project of Future Earth and a research partner of the World Climate Research Programme) in partnership with International Nitrogen Initiative (INI). We acknowledge all the people and institutions who provided the data used in the global nitrous oxide budget as well as the institutions funding parts of this effort (see Table A5). We acknowledge the modelling groups for making their simulations available for this analysis. H.T. and S.P. acknowledges computational and administrative support from Schiller Institute for Integrated Science and Society at Boston College, and International Center for Climate and Global Change Research at Auburn University. J.G.C. thanks the Australian National Environmental Science Program - Climate Systems Hub for supporting the GHG budget activities of the Global Carbon Project (GCP), including the Global and regional $N_2O$ Budgets work. We are grateful to the EDGAR team (M. Crippa, D. Guizzardi, E. Schaaf, M. Muntean, E. Solazzo, F. Pagani and M. Banja) for the work needed to publish the EDGARv7.0 Global Greenhouse Gas Emissions dataset (https://edgar.jrc.ec.europa.eu/dataset_ghg70). AKJ thanks Shijie Shu and Tzu-Shun Lin for their involvement in developing



and analyzing the ISAM model products used here. Giulia Conchedda, Griffiths Obli-Layrea and Nathan Wanner contributed
significant efforts to the generation of fertilizer, livestock and soil nutrient data that underlie FAO's estimates of $N_2O$ data.

**Financial support**

Please see a full list of funders in the Appendix (Table A5).



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
