# Peer review of "Global Nitrous Oxide Budget 1980-2020"

_Earth System Science Data, 2023_

## Referee Comment (RC2)

Review ESSD-2023-401, Global N2O budget

Remarkable compilation of many N-relevant processes. Hard to review, particularly because they have many experts in their author list. Definite merit for ESSD, but needs key improvements.

Basic approach: For important N-molecule, one needs to pay attention to a wide variety of sources, sinks, and biochemical reactions. This paper represents best efforts. They itemize important sources, sinks, and reactions. They draw conclusions by source and by regions. They list abundant uncertainties but rarely in numerical terms. Instead, authors focus on geographic patterns/biases of these uncertainties. They recommend that uncertainties get resolved ("reconciled") in future observations and from future models.

For examples: Line 1173: "large uncertainty in the estimates of agricultural $N_2O$ emissions"; Line 1209: "large discrepancy in natural soil emissions among NMIP2 models exists"; Line 1391: "variability of these emissions remains uncertain" (referring to emissions from continental shelf regions); etc.

They miss (fail to address) test-able hypothesis: that uncertainties remain so large that conclusions prove speculative (at best). Reader finds zero discussion of cumulative numeric uncertainty. In very practical terms, this reader suggests that uncertainties mapped geographically in Figure 21 overwhelm any regional signals authors might hope to identify in Figure 13.

In GCB, reader immediately (abstract) learns that those authors present all data with consistent specified uncertainty of $\pm 1$ sigma. In global methane budget, reader confronts ranges (min/max) but those authors write (explicitly, in second paragraph of Introduction): uncertainties in regional emissions may reach 40%–60% (of global mean). For N2O, where obs come directly from same sources as for CH4, where microbial processes intervene in both sources and sinks, and where models have similar (plus, additional?) weaknesses as for other chemical or transport models, one expects similar quantitative uncertainty info? Should readers assume that we know N2O concentrations and fluxes with better-than-CH4 uncertainty, or with worse? Reader here gets no hints. This particular reader assumes 'worse'! Not because authors have done a poor job (instead, they seem to have done a very good job) but because N2O represents a more difficult, complicated reactive molecule than e.g. CO2 or CH4. If the target remains more elusive, authors must demonstrate and exercise BETTER methods to identify and quantify. Disparaging statement above (e.g. that uncertainties remain so large so as to preclude conclusions on specific sources or assigned to specific regions) represents a clear refutable hypothesis. Unfortunately, authors never pose nor address such a hypothesis.

Validation remains conceptually and quantitatively difficult for these global budgets. Other ESSD products showed good success by using 'leave-one-out' techniques. Because these authors report multiple atmospheric data sources, multiple model outcomes, etc., they might choose to pursue similar leave-one-out strategies? Or, and this would prove relevant to uncertainty issues, authors feel reluctant to identify one obs data set or one model as 'reference'?

I identify many small but necessary changes below. Until, however, readers gain a complete quantitative discursion on uncertainty, I doubt that suggestions that follow pertain. Manuscript needs serious overhaul; small fixes unimportant on that scale.

Line 84: "accumulating in the atmosphere since the pre-industrial period". No evidence presented here. Most data in this paper start in 1980. No citation for statement, in next sentence, that N2O concentrations "from 270 parts per billion (ppb) in 1750". How do we know

pre-1980 N2O concentrations? In other publications, GCP considers 1750 as start of industrial period? 'Pre-industrial' seems unsupportable and too vague. I know what authors intend here but many ESSD readers will not?

Line 92: if fluxes increased by 40% over 1980 to 2020 but concentrations increased only 25% (from 1750 to 2022, line 85) then something must also have changed in sinks? E.g. Fig 1 shows at least three sinks (downward arrows, including one massive downward arrow); atmospheric concentrations must represent some balance of these processes? Not clear here, nor elsewhere in thus manuscript. Authors job to compile and present best data (good on them) but also to explain basic balance / imbalance of global N2O budget? This reader might understand subtle differences here but many readers will not? Somewhere, in abstract or exec summary, readers need to find concise summary?

Lines 107 to 116: Good summary here! Compares recent work (1980 to 2020) to ice core records. No mention of 'industrial' or 'pre-industrial'. Revise abstract in light of what you have here? Also no mention of stratospheric processes (O3 impact) or sinks but these processes emerge later? Or, somewhere, a sentence that this budget focuses (necessarily) primarily on tropospheric processes?

Line 110: "It" You mean 'these'? Or, 'these concentrations'? Eliminate this sentence, because you do not need two successive mentions of ice core records?

Here, you use and cite units in fluxes of ppb per year. In abstract and next (emissions) paragraph you instead use Tg N per year. Settle on most useful or most appropriate set of units? Or, include a table that helps readers quickly convert?

Line 195: "most emitted"? Most often emitted? Most emitted by net (atmospheric) concentration or by flux? Most reactive? Need slight clarification here.

Line 198: "mole fractions have increased by more than 25% since the pre-industrial era, from 270 parts per billion (ppb) in 1750 to 336 ppb in 2022" But, figures here only show data since 1980. If increase since pre-industrial values is true (as I accept), reader needs a citation or source for such data?

Line 200: detailed and well-referenced sentence starting 'The 20th century rate' renders the previous sentence moot; reader does not need to see both. ESSD/Copernicus impose some punctuation standard for '20th century'?

Line 204: "growth rate of atmospheric $N_2O$, the mean annual growth" need changed punctuation here, e.g. growth rate of atmospheric $N_2O$: the mean annual growth.

Line 217: "Reducing $N_2O$ emissions is a required net-zero greenhouse gas (GHG) emissions and the recovery of stratospheric ozone" Something missing here? Required to meet GHG targets, and to allow (or foster) recovery?

Line 218: Complete "$N_2O$ mitigation measures"? I think Pier's paper pointed out that remaining CO2 targets/budgets for 2C disappeared into the 'noise' of other GHG mitigation efforts (e.g. for N2O) but did not address N2O mitigation impacts directly?

Line 221: "Implementing $N_2O$ mitigation" you already said this in previous sentence. This sentence seems redundant?

Line 223: Nitrification and denitrification might both impact N2O production but, with one a source and one a sink, they can't both "contribute". Awkward phasing for most readers.

Line 225 and following: good list but punctuation should change to semi-colon between each of 21 factors? Proof readers will know.

Line 225 and following: check Fig 3 to ensure tight correlation with list here, by exact terms and directions of arrows? I think I counted 21 fluxes in Fig 3 but with uncertainty about whether to count bidirectional arrows as one or two terms?

Line 262: But, no red arrow (indirect anthropogenic impact) from coast oceans box of Fig 3?

Line 266: Good list but, strictly speaking, these should not fit in the category of terrestrial natural ecosystems?

Line 284: "multiple BU (BU) and TD (TD) methods" something missing or awkward here?

Line 302: "all possible" but readers just learned that you had to ignore termite sources for lack of data? Perhaps all 'plausible'. Or, all 'quantifiable'? Change wording to reflect availability of reliable data? Lists and categories that follow seem reasonable and well-documented. Later (Line 1443) authors devote an entire paragraph to "missing fluxes". "All possible" remains confusing and/or inappropriate.

Line 316: very important if slightly confusing paragraph. Put this in a table, instead? Ala Table 1 in GCB? Fluxes, change rates of fluxes, atmospheric concentrations: too much for reader to remember without a reference table?

Line 341: 'are' rather than "is"?

Line 364: to " to develop" and "quantified" in same sentence? Need some attention to tense here?

Please ensure to define bottom-up (BU) and top-down (TD) once and only once, then attend to all subsequent uses of abbreviations to ensure coherence. E.g. to this reader, text in legend to Figure 4 (line 379) seems confusing?

Here, confusion threatens overall merit of this work. By this point reader has confronted 21 types of N2O fluxes, recasting of those types into six broader categories, definition of units (fluxes, change rates of fluxes, concentrations), parsing across geographical regions (including to a few specific countries), and - finally (!?) elucidation by 31 (more if one counts same model run at two different spatial resolutions) inventories and global, regional or process models. Huge effort by authors to compile all this! Please keep readers well-informed and cognizant of which source (or sink) estimates apply to which categories. The category 'Shelf' for example, which authors intend as one depth-limited region of perimeter oceans, remains confusing as used. Authors will know best what they need to report and how but, unfortunately, present parsing and arrangement implies perfunctory approach while authors prefer to project entire effort as careful and complex. Some better way to convey complexity and uncertainty? Not clear for me. I plead for better overall arrangement or at least an ongoing outline to help readers? Table 1 represents a comprehensive list, without reliability designation? Figure 4 complicates when it might clarify?

Line 415: Copernicus publisher adheres to standard mechanism to handle 'submitted' references. Not this one, unfortunately; authors and editors need to correct.

Line 417: Which "observation-based analysis"?

Line 423: percent of what?

Line 424: "Shelf processes" in Table 1 actually represent "continental shelves" as specified here?

Line 434: If authors already listed all pertinent (hi-res) ocean biogeochemistry models in Table 1, does reader need a second list here? Again, this reader wishes for some clarification: unable to assign priority to any one model, authors have chosen to use them all, to use a mean, to use a median? How do uncertainties from individual models penetrate into overall global estimates?

Line 455: First reports on deriving uncertainty info from source materials? Reader of ESSD needs more of the same?

Line 465: More useful to list FAO general reports first, then to deal with FAOSTAT specifics on fire types after? E.g. helpful to readers to change order of last and next-to-last sentences?

Line 470 and following: In this section authors need to include continental shelves in a more-general 'coastal' category? Needs explicit mention/explanation?

Line 481: not clear where "56%" comes from?

Line 484: "low" or "act as a sink". Should reader assume you included these sources/sinks or ignored them as insignificant?

Line 486: "SH1-SH7" and line 487 "SH1-SH8" From supplement reader learns that SH1 etc. represent set-up parameters for models involved in NMIP2 but readers need that informations sooner, e.g. here?

Line 491: "book-keeping" approach may account for deforestation/reforestation but you lost readers on broader issue of overall issue of land-cover changes on indirect (perturbation?) emissions?

Line 495 and following: Not clear what authors concluded here: should they use older estimate for NH3 oxidation and lightning production or do they ignore these processes as "small" and inconsistent or unquantifiable?

Line 515: ppb to Tg conversion factor buried here, should appear more prominently in a 'units' table?

Line 517 and following: Did authors use $\pm$ 1.4% uncertainty or IPCC AR5 uncertainty? One applies to concentrations and other to concentration changes? Not clear to this reader.

Line 524: another example of a process (tropospheric loss) too small to appear in overall N2O budget?

Line 533: 'is' rather than "as"?

Line 548: additional uncertainties introduced by these interpolation or re-gridding steps?

Line 553: Back at line 319, we read "Unless specified, uncertainties are reported in brackets as minimum and maximum values of all estimates". Data presented here (e.g. 315.8 ("315.5-316.2) ppb in 2000 to 335.9 (335.6-336.1) ppb in 2022") follow this convention? Values in parentheses represent min and max? But, each source (e.g. NOAA, CSIRO) will have gone to lot of effort to

identify uncertainties of their N2O measurements, reported not as $\pm$ min/max. Min/max tells readers very little about distributions or uncertainties? Please can authors adopt, and adhere consistently to, better more reliable more informative uncertainties?

Line 556: "was" implies singular but this sentence refers two (plural) years, 2020 and 2021?

Line 556: "30% higher than the average value in the decade of the 2010s" Not sure that readers can confirm this information from Figure 2? Inset of Figure 2, which purports to show annual growth rate, has no uncertainties, no demarcation of decades, nothing to help readers follow (or, dispute) authors' conclusions. Help, please.

Line 563, Figure 5: No uncertainties in obs nor in model outcomes?

Line 571: "with large uncertainties (Figure 6)." What uncertainties? $\pm$ min/max as above? Something different here? No information! Nothing in panels or figure legend?

Line 579, Figure 6: Except for panel C (Other direct anthropogenic emissions), all uncertainties (if shown by color ranges) appear to increase, 1980 to 2020? Not a positive report? Should readers assume that capability to construct N2O budget decreases, because uncertainties increase? Not what this reviewer would have expected as measurements and models all improve? If end of 2020 total of 6.7, with range of 3.3 (minimum or ?) to 10.9 (maximum or ?), how can reader trust anything that follows about sectors or regions? If authors expect readers to accept time-dependent changes in N2O sources or sinks, those readers will need to trust authors' handling of received as well as generated uncertainties. No evidence provided here.

Line 583 and following, sections on agriculture, other direct, etc.: Lots of work here, compiling and reporting data from trusted sources NIMP2, EDGAR, FAOSTAT, etc., but not one mention of uncertainties. Do all these sources produce 'perfect' data? I know, and readers will know better, that each source spends a great deal of time and effort to identify and report uncertainties. All of that effort and info lost here? Not one error bar or uncertainty envelop in any of these figures? Authors need to provide readers a basis to trust these conclusions, to respect authors' good efforts, but nothing presented here provokes nor supports such respect. Fossil fuel N2O emissions have remained unchanged for four decades? Weakness in reporting? True? How can any reader know? We need to trust authors to provide explanation with documentation and uncertainties! Instead, nothing provided!

Line 633, "both DLEM and book-keeping approach suggested increasing uncertainties in post-deforestation pulse effect". Really? Increasing uncertainties? Expressed as min/max, 95CI, per cent of total, $\pm$ x sigma, what? If authors want readers to accept this contention about perturbation fluxes, those readers will want to have seen consistent approach to uncertainties up to this point and will need more details here.

Lines 648 and following: Reader would like to accept that natural N2O fluxes have not changed over four decades but a) this reader doubts that contention and b) authors have given no indication of their skill or knowledge to back up such contentions.

Line 650 and following: do authors now (or, again) present minimum and maximum values? Doubtful, but no information given to help readers decide? Readers would like to accept authors' contentions on all these natural N2O fluxes but, without uncertainty information and specifications, how can we?

Line 685, TD estimates: Inversions, as published, include substantial uncertainties. As presented (plotted) here, however, readers get no indication of authors expertise at assimilating

and assessing TD info. No error bars on any plot, can't be true? What valid signals emerge from what noise? No sense of that provided in this section.

Line 778, Figure 13: Total N2O emissions (panel A), look nothing like panel A of Figure 6 (total anthropogenic). Total here represents Fig 6 plus non-varying natural total of 12 (Line 651)? 7 plus 12 gives 19, perhaps within uncertainty (?) of this Figure? If this reader has made too-simple assumptions or additions, authors have not given sufficient information to prevent my errors? What do uncertainties shown here represent? Cumulative for BU and TD? From where? If we can't understand panel A, why should any reader give any credence to panels b to t? If I calculate instead from numbers given in Table 2 (again, with min and max?), I get 18 BU or 17 TD total N2O sources, minus stratospheric sink of 13 or total atmospheric sink of 14 to get a residual in atmosphere of 3 to 5? Compared to 6 as provided authors? Sorry, does not compute correctly. In notes to Table 2 one reads that uncertainties follow AR5 with detailed notes provided in Supplement. Sorry again, this reader will not comb through supplement to elucidate uncertainty info. Many readers will simply give up at this point.

Lines 790 to 1084, regional extrapolations. More hard work by authors but basically bogus without uncertainties or some mechanism to validate?

---

## Author Comment (AC2)

**Responses to reviewers' comments on "Global Nitrous Oxide Budget 1980–2020" (manuscript number essd-2023-401)**

We would like to thank the reviewers for their thoughtful and insightful comments. The manuscript has been revised accordingly, and our point-by-point responses in blue color are provided below, and our new/modified texts in the revised manuscript are indicated in red color.

Reviewer 3:

This paper is an ambitious and detailed compilation of information across many data sources and different modeling approaches. As such, it will provide a useful reference for the scientific community, including policy makers.

Response: We thank the reviewer for the positive comments!

I have listed my most major comments below.
The paper does not always clearly distinguish between results that are highly speculative and purely model based (e.g., responses to CO2 and climate change) and those that have a more solid grounding in data (e.g., responses to fertilizer and manure inputs). For example, lines 123-131 are stated as though they are facts. They should be qualified with, "according to BU estimates," as is appropriately done for the presentation of model results in the next paragraph starting on line 132.

Response: Thank you for your insightful suggestion! We acknowledge the importance of distinctly differentiating between speculative, model-based results and those grounded in data. We have added "According to BU estimates," to the revised manuscript.

It also would be useful to include a graphical depiction of the relative uncertainty of different budget terms in Figure 1, e.g., perhaps with solid arrows for more robustly known fluxes and more faded arrows for speculative fluxes. (The current arrows have more faded colors that transition to more solid colors but it is not clear what this transition represents.)

Response: Thanks for the suggestion. We have tried to implement it, but the final result was not satisfactory from a visual point of view. We hope that the fact each flux value comes with a clear range, as an indication of its uncertainty, provides all the information necessary for the reader to know that not all fluxes are known the same.

Line 217 "Reducing N2O emissions is a required net-zero greenhouse gas (GHG) emission and the recovery of stratospheric ozone." First, this sentence does not make sense grammatically. Second and more importantly, the concept of "net zero" is not appropriate for N2O. Earth has always been, and always will be, a net natural source of N2O, which is then destroyed photochemically in the stratosphere. "Net zero" is mentioned again on line 298 so line 217 does not seem to be just a typo. I bring this up because I have seen essays arguing for "N2O neutrality" as a feasible and desirable policy goal, to the point that spraying chemicals on natural landscapes is considered as a way to stop nitrification. The idea that humans should be actively trying to stop natural N2O emissions seems likely to have potentially bad unintended

consequences. The concept that "net zero" is not logical for N2O (which is not analogous to CO2) should be clearly communicated to policy makers, who may not be familiar with Earth's natural biogeochemical cycles.

Response: Thanks for pointing this out! We agree with the reviewer on that Earth surface is a net natural source of $N_2O$ and $N_2O$ neutrality is not a feasible and desirable policy goal. We have revised the sentences to avoid misleading readers and policymakers:

 "Reducing $N_2O$ emissions will contribute to the mitigation of global warming and the recovery of stratospheric ozone (Jackson et al., 2019)."

 "contribute to the global stocktake of the Paris Agreement to track progress towards national determined contributions."

There is a confusing switching back and forth between 3 alternative time frames: 1997-2020, 1980-2020 and 2010-2019. Please state clearly somewhere early in the methods why these 3 time frames were chosen and why each is significant.

Response: "We focus on $N_2O$ fluxes and their change rates during three periods: 1997-2020, 1980-2020 and 2010-2019. 1980-2020 is the entire study period, we report temporal variations in BU estimates of $N_2O$ emissions from different sources to depict the overall trends of these fluxes. 1997-2020 is the overlapping period of BU and TD approaches, we compare BU and TD estimates during this period to exam their consistency. 2010-2019 is the most recent decade, we report the magnitudes of emissions from different sources to show their latest status and relative importance." We have added these statements to the revised manuscript.

The frequent reporting of rates of increase in TgN/yr-2 units is not intuitively meaningful and arguably not mathematically correct. Strictly speaking, it assumes that the source can be fit with a parabolic (t^2) parabolic dependence on time, which is not obviously the case for many regions, based on figure 14. At minimum please explain how these rates of rates of increase were calculated and why there is so much emphasis on reporting them.

Response: We are sorry for the unclear statement. As stated in Section 2.1, annual $N_2O$ fluxes are expressed in teragrams of $N_2O$-N per year: Tg $N_2O$-N $yr^{-1}$ (Tg N $yr^{-1}$), and the change rates in $N_2O$ fluxes are expressed in the unit of Tg $N_2O$-N $yr^{-2}$ (Tg N $yr^{-2}$). Therefore, rates of increase or decrease reported in this paper are the first derivatives of annual $N_2O$ fluxes, rather than the second derivatives. They represent the average change rate
We have revised the sentence to avoid confusion:

"In this study, $N_2O$ fluxes are expressed in teragrams of $N_2O$-N per year: 1 Tg $N_2O$-N $yr^{-1}$ (1 Tg N $yr^{-1}$) $=10^{12}$ g $N_2O$-N $yr^{-1}=1.57\times10^{12}$ g $N_2O$ $yr^{-1}$, with change rates in $N_2O$ fluxes expressed in the unit of Tg $N_2O$-N $yr^{-2}$ (Tg N $yr^{-2}$) which represent the first derivative of annual $N_2O$ fluxes calculated by the linear regression method."

Below is a list of more detailed minor comments.

Line 84. It would be useful to provide an estimate of N2O's relative contribution to enhanced GH forcing (e.g., 6% or whatever the latest value is).

Response: Thanks for the suggestion. We have added the following sentence into the revised manuscript: "According to Forster et al (2023), $N_2O$'s relative contribution to the total enhanced effective radiative forcing of greenhouse gases was 6.4% for 1750-2022."

Line 109. It's odd to mention 10% here when the abstract cited "nearly 25%" from the preindustrial, indicating that 15% of the rise was prior to 1980. This isn't wrong necessarily but it sends a confusing message.

Response: Thanks for the suggestion! We have revised this sentence according to your suggestion:
"The tropospheric $N_2O$ mole fractions, precisely measured at a global network of stations, increased from 301 parts per billion (ppb) in 1980 to 333 ppb in 2020 and 336 ppb in 2022."

Line 110. What does "it" refer to?

Response: We are sorry for the unclear statement. "It" refers to the tropospheric $N_2O$ mole fraction in 2022. We have revised this sentence:

"The tropospheric $N_2O$ mole fraction in 2022 is higher than at any time in the last 800,000 years."

Line 113. I suggest to delete "with a substantially lower resolution" because it is confusing.

Response: Thanks for the suggestion, and we have deleted it according to your suggestion.

Lines 123-131. This paragraph should state explicitly that these results (i.e., trends in different sources) are based on BU approaches, since TD approaches cannot generally distinguish individual sources. As such, the BU results are in large part based on speculative model-based estimates. The paragraph states them as though they are known facts.

Response: Thanks for your insightful suggestion! We have added "According to BU estimates," in the manuscript.

Lines 132-144. This paragraph seems to overlap considerably with the previous paragraph (lines 117-131). Is the previous paragraph covering the period 1997-2020 (or 1980-2020?), while the current paragraph covers 2010-2019? Please distinguish the 2 paragraphs more clearly and consider deleting one of them unless there is a compelling reason to distinguish the last 10 years from the last 23-40 years.

Response: Line 117-131 focuses on reporting the temporal variations in $N_2O$ emissions over 1980-2020 (the entire study period) and compare the trends in BU and TD estimates during their overlapping period (1997-2020). Line 132-144 shows the

magnitudes of emissions from different sources and their relative importance in the most recent decade. Moreover, line 117-131 focuses on the total emissions or emissions from the five big categories, while line 132-144 also reports emissions from the 21 identified natural and anthropogenic sources which contains more detailed information. In summary, these two paragraphs have different emphases. Therefore, we think it is better to keep both paragraphs.

Line 149. "since" is confusing. Do you mean "by"?

Response: Yes! We have corrected it.

Line 151. Is this a second derivative (Tg N/yr-2)? I think this will be confusing to many readers and not intuitively meaningful. I suggest to present as the rate of growth in TgN/yr in the 1980s contrasted with the higher rate of growth in the most recent decade.

Response: We are sorry for the unclear statement. As stated in Section 2.1, $N_2O$ fluxes are expressed in teragrams of $N_2O$-N per year: Tg $N_2O$-N $yr^{-1}$ (Tg N $yr^{-1}$). Therefore, it is the first derivative of annual $N_2O$ fluxes, rather than the second derivative. We have revised the sentence to avoid confusion:

"In this study, $N_2O$ fluxes are expressed in teragrams of $N_2O$-N per year: 1 Tg $N_2O$-N $yr^{-1}$ (1 Tg N $yr^{-1}$) $=10^{12}$ g $N_2O$-N $yr^{-1}=1.57\times10^{12}$ g $N_2O$ $yr^{-1}$, with change rates in $N_2O$ fluxes expressed in the unit of Tg $N_2O$-N $yr^{-2}$ (Tg N $yr^{-2}$) which represent the first derivative of annual $N_2O$ fluxes calculated by the linear regression method."

Line 169. What is "manure forest conversion"?

Response: We are sorry for the spelling error. It should be "mature forest conversion". We have corrected it in the manuscript.

Line 198. 66/270 = 24.4%, which is not "more than 25%".

Response: We are sorry for the calculation error. We have corrected it in the manuscript.

"Atmospheric $N_2O$ mole fractions have increased by more than 24% since the pre-industrial era"

Line 203. Please update the NOAA reference.
Lan, X., E.J. Dlugokencky, J.W. Mund, A.M. Crotwell, M.J. Crotwell, E. Moglia, M. Madronich, D. Neff and K.W. Thoning (2022). Atmospheric Nitrous Oxide Dry Air Mole Fractions from the NOAA GML Carbon Cycle Cooperative Global Air Sampling Network, 1997-2021, Version: 2022-11-21, *https://doi.org/10.15138/53g1-x417*.

Response: Thank you! We have updated the NOAA reference.

Line 207. Please clarify that this means since the period of observations began. As written, it seems to imply that the growth rate was higher prior to 1980.

Response: Yes, this means since the period of observations began. We have deleted "since 1980" to avoid confusion.

Line 217. What does "is a required net-zero GHG emission" mean?
Response: We are sorry for the grammatical error. We have revised the sentences as follows:

 "Reducing $N_2O$ emissions will contribute to the mitigation of global warming and the recovery of stratospheric ozone (Jackson et al., 2019)."

Line 223. This suggests that up to 44% of N2O is produced abiotically. This doesn't seem right. Furthermore the cited 56-70% seems at odds with Figure 1, in which only 2.3/18.1 TgN (12.7%) is from a non-microbial source.

Response: Thank you for the suggestion. This sentence has been revised:
"Nitrification and denitrification are the two key microbial processes controlling $N_2O$ production, making the largest contribution to global $N_2O$ emissions;"

Line 248. Grammar note: missing "a" before small.

Response: Thanks for pointing out this grammar error. We have revised accordingly.

Substance note: the effect on the N2O source from OMZs in the ocean will not necessarily be small, but perhaps this sentence refers to the fact that N2O contributes a relatively small part of GHG warming (even if the expanding OMZ effect on N2O emissions is large).

Response: Yes, we acknowledge that the effect on the $N_2O$ emissions from OMZs in the ocean may be large. This sentence refers to the fact that ocean $N_2O$ emission contributes a relatively small part of GHG warming.

Figure 3. Please delete page number 275.

Response: We have deleted it.

Line 298. Again, it is not possible or even desirable to achieve net zero emissions of N2O. To do so would involve severe disruptions to Earth's natural nitrogen cycle. This needs to be made clear to policy makers who are not familiar with biogeochemistry.

Response: Thanks for pointing this out! We agree with the reviewer on that Earth surface is a net natural source of $N_2O$ and $N_2O$ neutrality is not a feasible and desirable policy goal. We have deleted "and the ultimate goal of achieving net-zero GHG emissions" to avoid misleading readers and policymakers.

Line 355. It is debatable whether models are accurately capturing nitrification, denitrification, and other key processes (e.g., Nevison, C., Goodale, C., P. Hess, W.R. Wieder, J. Vira and P.M. Groffman (2022). Nitrification, and denitrification in the Community Land Model compared to observations at Hubbard Brook Forest. Ecological Applications. https://doi.org/10.1002/eap.2530.)

Response: Thanks for pointing this out! We acknowledge that it is debatable whether process-based models can accurately capture nitrification, denitrification, and other key processes. We revised the sentence as follows:

"they are capable of modelling the key processes affecting $N_2O$ production and emission such as autotrophic nitrification, denitrification, plant nitrogen uptake, ammonia volatilization, nitrate leaching, soil thermal and hydrological processes, although their accuracy in representing these processes needs further improvement;"

p.15, Table 1. Please specify whether the shelf products represent natural or anthropogenic emissions or both.

Response: We categorized shelf emissions into the natural emissions category. We have specified it in the Table 1.

p.16, Table 1. Please indicate what type of emissions are modeled with these "other" approaches.

Response: Thank you for the suggestion! We have added the type of emission modeled by SRNM, bookkeeping method, and IMAGE-GNM in Table 1.

Line 419. That this part of the natural flux was ASSUMED to be constant should be acknowledged in the abstract around line 130, which states that natural sources were relatively constant, as though this were a result.

Response: Thank you for this suggestion. Among all sources, natural emissions from shelves, inland waters, and lightning and atmospheric production were assumed to be constant during 1980-2020. We have added this sentence to the manuscript:

"Among all sources, natural emissions from shelves, inland waters, and lightning and atmospheric production were assumed to be constant during 1980-2020. According to BU approaches, the total natural emissions from these sources were 1.7 (0.9-3.0) Tg N $yr^{-1}$."

Line 423. 44% of what?

Response: 44% of the total $N_2O$ emissions from inland waters. We are sorry for the unclear statement. The results in Yao et al. (2020) suggested that 56% of the total $N_2O$ emissions from rivers, reservoirs, estuaries and lakes were attributed to anthropogenic N additions, and the resting 44% of the total $N_2O$ emissions were from natural sources. To avoid confusion, this sentence has been revised as:

"Using this approach, we estimated that $N_2O$ emissions from natural sources of rivers, reservoirs, lakes and estuaries accounted for 44% (36%−52%) of the total emissions from inland waters."

Line 492. Please comment here on whether any data exist to evaluate these model predictions. This seems like a highly speculative flux to include in the budget, with a less solid grounding in data and observations than, e.g., direct agricultural emissions from fertilizer or manure.

Response: We acknowledge that $N_2O$ fluxes from climate/$CO_2$/land cover change are highly uncertain. Although data from control experiments exist at several sites, no global or regional level data exists to evaluate model predictions.

Line 505. Is this in the stratosphere or the troposphere?

Response: This is in the stratosphere. We are sorry for the unclear statement. We have revised the sentence:

"There is also $N_2O$ production from N2 +O(1D), which amounts to about 2% of the atmospheric source in the stratosphere (Estupiñán et al. 2002)."

Line 587. Why such a small increase in manure management when the other 3 sources increased by 50% or more during the same period?

Response: Emissions from agriculture-related activities in EDGAR and FAOSTAT, including direct and indirect $N_2O$ emissions and manure management are estimated based on the IPCC methodology. IPCC coefficients for manure management are skewed towards developed countries, where animal numbers have stayed rather constant overall—and diminished in Europe for instance. Conversely, manure left on pasture is applied to all countries and hence heavily influenced by increasing animal trends in the rest of the world. For manure management sector, the technology penetration in some countries might offset the increases in emissions associated to the increases in number of heads.

Line 626. What is EDGAR/NMIP2? I do not see it described above in the methods.

Response: Sorry, it should be NMIP2/EDGAR v7.0, we have corrected it. We define "NMIP2/EDGAR v7.0" in section 2.4.4: "EDGAR v7.0 provided estimates of indirect emissions from both agricultural and non-agricultural sectors, however, here, we sum the ensemble mean of NMIP2 estimates of indirect emissions from agricultural sectors with indirect emissions from non-agricultural sector of EDGAR v7.0 (i.e., NMIP2/EDGAR v7.0) to represent N deposition induced soil emissions from both agricultural and non-agricultural sectors."

Line 700. Perhaps comment on whether the trend in the posterior fluxes differed substantially from the assumed trend in the prior emissions.

Response: Thanks for the suggestion. We have revised accordingly.

Figure 13. Please label TD and BU in the figure legend and/or describe in the caption.

Response: We are sorry for the unclear statement. We have added more statements in the figure caption:

"The blue lines represent the mean $N_2O$ emission from bottom-up methods and the shaded areas show minimum and maximum estimates; the red lines represent the mean $N_2O$ emission from top-down methods and the shaded areas show minimum and maximum estimates."

Line 798. Similar to my comment in the executive summary, why are 1997-2020 and 1980-2020 used as alternative historical periods?

Response: 1980-2020 is the entire study period, we report temporal variations in BU estimates of $N_2O$ emissions from different sources to depict the overall trends of these fluxes. 1997-2020 is the overlapping period of BU and TD approaches, we compare BU and TD estimates during this period to exam their consistency. We have added these statements to the revised manuscript.

Line 898. Was this decrease due mainly to a reduction in fertilizer use from 1980 to 2020? If so, readers might be interested to know if the reduction was this achieved due to deliberate mitigation strategies or rather to the collapse of the Soviet Union?

Response: This decrease was mainly caused by a reduction in fertilizer use after the collapse of the Soviet Union (Tian et al., 2022). We have revised this sentence:

"Direct agricultural emissions and indirect emissions show overall decrease trends from 0.46 and 0.16 Tg N $yr^{-1}$ in 1980 to 0.38 and 0.12 Tg N $yr^{-1}$ in 2020, respectively, mainly due to a reduction in fertilizer use after the collapse of the Soviet Union (Tian et al., 2022)."

Line 998 and elsewhere. Again, the use of the second derivative is confusing.

Response: We are sorry for the unclear statement. They are first derivatives of annual $N_2O$ fluxes, rather than second derivatives. We have revised the description of units of $N_2O$ fluxes to avoid confusion:

"In this study, $N_2O$ fluxes are expressed in teragrams of $N_2O$-N per year: 1 Tg $N_2O$-N $yr^{-1}$ (1 Tg N $yr^{-1}$) $=10^{12}$ g $N_2O$-N $yr^{-1}=1.57\times10^{12}$ g $N_2O$ $yr^{-1}$, with change rates in $N_2O$ fluxes expressed in the unit of Tg $N_2O$-N $yr^{-2}$ (Tg N $yr^{-2}$) which represent the first derivative of annual $N_2O$ fluxes calculated by the linear regression method."

Line 1112. Again, what is manure forest conversion?

Response: We are sorry for the spelling error. It should be "mature forest conversion". We have corrected it in the manuscript.

**References:**
Estupinán, E. G., et al. "Investigation of $N_2O$ production from 266 and 532 nm laser flash photolysis of $O_3/N_2/O_2$ mixtures." *The Journal of Physical Chemistry A* 106.24 (2002): 5880-5890.

Forster, Piers M., et al. "Indicators of Global Climate Change 2022: annual update of large-scale indicators of the state of the climate system and human influence." *Earth System Science Data* 15.6 (2023): 2295-2327.

Tian, H., et al. "History of anthropogenic Nitrogen inputs (HaNi) to the terrestrial biosphere: a 5 arcmin resolution annual dataset from 1860 to 2019." *Earth System*

*Science Data* 14.10 (2022): 4551-4568.

Yao, Y., et al. "Increased global nitrous oxide emissions from streams and rivers in the Anthropocene." *Nature Climate Change* 10.2 (2020): 138-142.

---

## Author Comment (AC3)

**Responses to reviewers' comments on "Global Nitrous Oxide Budget 1980–2020" (manuscript number essd-2023-401)**

We would like to thank the reviewers for their thoughtful and insightful comments. The manuscript has been revised accordingly, and our point-by-point responses in blue color are provided below, and our new/modified texts in the revised manuscript are indicated in red color.

**Response to Reviewer 1:**

The paper is very comprehensive and provides the most complete and accurate N2O budget published to date. Estimates for almost all known sources/sinks are included and disaggregated spatially and temporally. One of the main strengths is various bottom up and different top-down inversion methods are used and it is encouraging that the estimates are mostly consistent. The information presented is very useful and anticipate that this work will be well cited. Some relatively minor points of clarification/suggestions for improvement:

Response: We thank the reviewer for the positive comments!

Lines 131-132 are #s based on BU or TD?

Response: We are sorry for the unclear statement. These numbers are based on BU approaches. We have revised the sentence to make it more clear to readers:
"Unlike anthropogenic emissions, global natural land and ocean $N_2O$ emissions were relatively stable. According to the BU approaches, the total amount of global natural $N_2O$ emissions fluctuated between 11.5 and 11.9 Tg yr$^{-1}$ during 1980-2020."

Line 143 and elsewhere manure should be mature

Response: Thanks for pointing out the spelling error. We have corrected this error.
Figure 3, why no anthropogenic source for coastal?

Response: This was an omission from our side and a red arrow from the coastal waters box back to the atmosphere has now been added to the revised figure. This change is consistent with the text that accounts for aquaculture (part of it being coastal) as a direct anthropogenic source and coastal emissions induced by N leaching as an indirect anthropogenic source.

Lines 447 and 1170 were FAOSTAT emission factors for N additions based on the 2006 guidelines or the 2019 refinement?

Response: FAOSTAT emission factors for N additions are based on the 2006 guidelines. We have added this statement to P#L##.

Line 481 does this mean that 56% of N inputs were assumed to be anthropogenic and consequently 56% of total N2O from this source is anthropogenic?

Response: We are sorry for the unclear statement. The results in Yao et al. (2020) suggested that 56% of the total $N_2O$ emissions from rivers, reservoirs, estuaries and

lakes was attributed to anthropogenic N additions. Empirical methods (empirical models and meta-analysis) adopted this ratio to calculate long-term average anthropogenic $N_2O$ emissions from inland waters, consistent with Tian et al. (2020). This sentence has been revised as:

"The anthropogenic emission from inland freshwaters estimated by Yao et al. (2020) considered annual N inputs and other environmental factors (i.e., climate, elevated $CO_2$, and land cover change). The results in Yao et al. (2020) suggested that 56% of the total $N_2O$ emissions from rivers, reservoirs, estuaries and lakes was attributed to anthropogenic N additions. Empirical methods (empirical models and meta-analysis) adopted this ratio to calculate long-term average anthropogenic $N_2O$ emissions from inland waters, consistent with Tian et al. (2020)."

Line 489 define bookkeeping approach; is it the same as mass balance?

Response: Thanks for the reviewer's comment. The bookkeeping approach is not built on the principle of mass balance. "In the original bookkeeping model developed by Houghton et al. (1983), land conversion and the affected carbon pools are tracked each year. The initial values of carbon pools are set for each type of land use. Annual changes of carbon pools in areas affected by land use change or some land management practices (like wood harvest and fire management) are prescribed in the model using response curves, which are usually a function of the age of the newly converted land use. These response curves are specific for each type of land cover type and land use change and do not include the effects of environmental changes (Houghton and Castanho 2023). For each age cohort, it either gains carbon (afforestation or reforestation) or loses carbon (deforestation) until its carbon pools reach a new stable state (the response curve converges). Here different from the original bookkeeping model calculating carbon fluxes through tracking changes in vegetation or soil pools, the response curves directly tracking annual $N_2O$ emissions after deforestation, which are also a function of the age of newly converted land use, were developed in our bookkeeping method (The details refer to Supplementary Information SI-9)." The above statements have been added to the text for better explanation of our bookkeeping method.

Line 503 does the 5 Tg N refer to NOx?

Response: Yes, this refers to NOx. We have revised the sentence:
 "we assume an effective emission factor of 1% (de Klein et al. 2006) and using the median estimate of 5 Tg N $yr^{-1}$ of NOx,".

Figure 13 state that blue is BU and red TD

Response: We are sorry for the unclear statement. We have added more statements in the figure caption:
"The blue lines represent the mean $N_2O$ emission from bottom-up methods and the shaded areas show minimum and maximum estimates; the red lines represent the mean $N_2O$ emission from top-down methods and the shaded areas show minimum and maximum estimates."

Line 795 replace The sections followed with The following sections
Response: Thanks for the suggestion, we have revised accordingly.

Figure 13 KAJ seems low. Perhaps this is related to Figure 15, the green bar for TD shows a net sink; this does not seem correct, please double check.

Response: The figures on the right Y axis of the KAJ subfigure in Figure 15 were incorrect, we have revised the KAJ subfigure. In the revised figure, TD shows a net $N_2O$ source.

Figure 14 does ensemble include BU and TD?

Response: This figure only shows the estimates of BU approaches because TD approaches are not able to quantify the contributions of different sources. We have revised the caption to make it more clear to the readers:

"Figure 14. Ensembles of regional anthropogenic $N_2O$ emissions over the period 1980–2020 estimated by BU approaches."

Figure 15 I think (blue) should be (red) and (yellow) should be (green)

Response: We have double checked the colors of the bars in Figure 15. There is no mismatch between the colors and the items they represent.

Figure 16b why is non-ag error bar so large? 16d what are A B C D E and why such large bars for A?

Response: The error bar of A in Figure 16(d) is large due to the accumulation of large uncertainties in $CO_2$ effect, climate effect, and post-deforestation effect (Table 2, Figure 10). The error bar of non-agricultural emissions in Figure 16b is large because it is the sum of uncertainties in eight items: $CO_2$ effect, climate effect, post-deforestation effect, long-term effect of reduced mature forest area, emissions from nitrogen deposition on ocean, fossil fuels and industry, waste and waste water, and biomass burning. Among these items, $CO_2$ effect, climate effect, post-deforestation effect, and emissions from biomass burning have large uncertainties.

A-E in Figure 16d represent perturbed $N_2O$ fluxes from climate/$CO_2$/land cover change, emissions from nitrogen deposition on ocean, emissions from fossil fuels and industry, emissions from waste and waste water, and emissions from biomass burning, respectively. We have added more statements in the figure caption:

"A-E in Figure 16d represent perturbed $N_2O$ fluxes from climate/$CO_2$/land cover change, emissions from nitrogen deposition on ocean, emissions from fossil fuels and industry, emissions from waste and waste water, and emissions from biomass burning, respectively."

Line 1177 mentions higher tier estimates. In this context, mention that the USA uses a Tier 3 approach for most agricultural soils.

Response: Thanks for the suggestion. We have revised this sentence as follows:
"… especially for regions where N input surplus is high such as Eastern China, India, and the USA. For example, the U.S. national inventory uses a Tier 3 modelling approach (Del Grosso et al., 2022)."

Supplement line 15 says 6 models accounted for manure N but line 69 says 5 models

Response: We are sorry for the inconsistent descriptions. Only five models include manure N: DLEM, ISAM, O-CN, ORCHIDEE, and VISIT. We have corrected the number in supplementary information line 15.

Line 87 equation is missing

Response: Sorry for missing the equation. We have added the missing equation.

Line 126 use more descriptive text than intense. Perhaps state if microbes or plant roots have 1st shot at available N

Response: Thanks for the suggestion. We have added more descriptions of nutrient competition in the ELM model.

"The competition of those limited resources is represented by consumer–substrate networks, therefore, the uptake of nutrient substrate by each consumer is dependent on the relative competitiveness of one consumer over the others. Nutrient consumers' competitiveness is parametrized with kinetic parameters (Zhu et al., 2016). As a result, neither plan nor soil microbes get the first priority to access nutrient substrates."

Lines 294 and 303 - 2006 guidelines or 2019 refinement?

Response: FAOSTAT emission factors for N additions are based on the 2006 guidelines. We have added this statement to the revised manuscript.

**References:**
Del Grosso, Stephen J., et al. "A gap in nitrous oxide emission reporting complicates long-term climate mitigation." Proceedings of the National Academy of Sciences 119.31 (2022): e2200354119.

Houghton, R. A., J. E. Hobbie, J. M. Melillo, B. Moore, B. J. Peterson, G. R. Shaver & G. M. Woodwell (1983) Changes in the Carbon Content of Terrestrial Biota and Soils between 1860 and 1980: A Net Release of CO'2 to the Atmosphere. Ecological Monographs, 53, 235-262.

Houghton, R. A. & A. Castanho (2023) Annual emissions of carbon from land use, land-use change, and forestry from 1850 to 2020. Earth Syst. Sci. Data, 15, 2025-2054.

Tian, H., et al. "A comprehensive quantification of global nitrous oxide sources and sinks." *Nature* 586.7828 (2020): 248-256.

Yao, Y., et al. "Increased global nitrous oxide emissions from streams and rivers in the Anthropocene." *Nature Climate Change* 10.2 (2020): 138-142.

Zhu, Q., et al. "Multiple soil nutrient competition between plants, microbes, and mineral surfaces: model development, parameterization, and example applications in several tropical forests." *Biogeosciences* 13.1 (2016): 341-363.

---

## Author Comment (AC4)

**Responses to reviewers' comments on "Global Nitrous Oxide Budget 1980–2020" (manuscript number essd-2023-401)**

We would like to thank the reviewers for their thoughtful and insightful comments. The manuscript has been revised accordingly, and our point-by-point responses in blue color are provided below, and our new/modified texts in the revised manuscript are indicated in red color.

**Reviewer 2:**

Review ESSD-2023-401, Global N2O budget

Remarkable compilation of many N-relevant processes. Hard to review, particularly because they have many experts in their author list. Definite merit for ESSD, but needs key improvements. Basic approach: For important N-molecule, one needs to pay attention to a wide variety of sources, sinks, and biochemical reactions. This paper represents best efforts. They itemize important sources, sinks, and reactions. They draw conclusions by source and by regions. They list abundant uncertainties but rarely in numerical terms. Instead, authors focus on geographic patterns/biases of these uncertainties. They recommend that uncertainties get resolved ("reconciled") in future observations and from future models.
For examples: Line 1173: "large uncertainty in the estimates of agricultural N2O emissions"; Line 1209: "large discrepancy in natural soil emissions among NMIP2 models exists"; Line 1391: "variability of these emissions remains uncertain" (referring to emissions from continental shelf regions); etc.
They miss (fail to address) test-able hypothesis: that uncertainties remain so large that conclusions prove speculative (at best). Reader finds zero discussion of cumulative numeric uncertainty. In very practical terms, this reader suggests that uncertainties mapped geographically in Figure 21 overwhelm any regional signals authors might hope to identify in Figure 13.

Response: Our study. has developed the most comprehensive assessments possible given the observations and modeling capability available to date. Following the IPCC approach, we build robustness of what we know on the global N$_2$O budget and its component fluxes by bringing multiple and partially/fully independent lines of evidence to constrain temporal and spatial fluxes. The diversity of data and approaches used in this effort has limitations in the way we can treat numerically uncertainties, and so we ensure we provide at all times range values for each budget flux as the best expression of uncertainty we can provide. As it is done in other GCP global budgets, agreed protocols and community acceptance of best practices and data dictates what is used for this assessment to ensure the highest quality, and not a simple review of everything that has been published.

In GCB, reader immediately (abstract) learns that those authors present all data with consistent specified uncertainty of +1 sigma. In global methane budget, reader confronts ranges (min/max) but those authors write (explicitly, in second paragraph of Introduction): uncertainties in regional emissions may reach 40%–60% (of global mean). For N2O, where obs come directly from same sources as for CH4, where microbial processes intervene in both sources and sinks, and where models have similar (plus, additional?) weaknesses as for other chemical or transport models, one expects

similar quantitative uncertainty info? Should readers assume that we know N2O concentrations and fluxes with better-than-CH4 uncertainty, or with worse? Reader here gets no hints. This particular reader assumes 'worse'! Not because authors have done a poor job (instead, they seem to have done a very good job) but because N2O represents a more difficult, complicated reactive molecule than e.g. CO2 or CH4. If the target remains more elusive, authors must demonstrate and exercise BETTER methods to identify and quantify. Disparaging statement above (e.g. that uncertainties remain so large so as to preclude conclusions on specific sources or assigned to specific regions) represents a clear refutable hypothesis. Unfortunately, authors never pose nor address such a hypothesis. Validation remains conceptually and quantitatively difficult for these global budgets. Other ESSD products showed good success by using 'leave-one-out' techniques. Because these authors report multiple atmospheric data sources, multiple model outcomes, etc., they might choose to pursue similar leave-one-out strategies? Or, and this would prove relevant to uncertainty issues, authors feel reluctant to identify one obs data set or one model as 'reference'?

Response: As the reviewer is aware, the Global Carbon Budget provides a much more mature, and process and datasets and modeling are more mature and established. Separate assessments are available for $CO_2$, $CH_4$ and now also for $N_2O$, that are characterized by the respective properties, emission sources and processes typical for the respective gas. It is not the aim of either of these activities to compare against each other, but rather to point out the distinctive conditions. The present manuscript on $N_2O$ uses a large range of available sources to provide an overall assessment, to provide results of fluxes into and out of the atmosphere, with the ranges derived from the differences from individual sources. Each individual approach may provide their own uncertainty analysis, based on the respective input parameters. As a guidance to the scientific community, we understand that providing the ranges will be much more useful than attempting a quantitative analysis of uncertainties, which may be a topic of a separate piece of work. Identifying where the uncertainties are well constrained or very large can be viewed as a success of our effort, which we hope will guide future research.

(Just for your information, in our opinion, the geographic uncertainties of N2O vs. CH4 budgets are about the same because we rely on similar bottom-up and top-down data constraints. For differences among sectors, we would say that we are more confident about N2O sources and sinks, because agriculture is clearly the dominant source and the stratosphere is clearly the dominant sink. In contrast, CH4 has multiple important sources in the range of 10-30% of the total. For CH4, tropospheric OH is the dominant sink, but the methanotrophic sink could still be bigger that currently estimated, whereas it is very unlikely that soils represent more than a tiny sink for N2O. We don't think that it is appropriate to devote space in the manuscript to compare N2O and CH4 uncertainties, but we suppose we could make these arguments in our response to the reviewer's comments).

I identify many small but necessary changes below. Until, however, readers gain a complete quantitative discursion on uncertainty, I doubt that suggestions that follow pertain. Manuscript needs serious overhaul; small fixes unimportant on that scale.

Line 84: "accumulating in the atmosphere since the pre-industrial period". No evidence

presented here. Most data in this paper start in 1980. No citation for statement, in next sentence, that N2O concentrations "from 270 parts per billion (ppb) in 1750". How do we know pre-1980 N2O concentrations? In other publications, GCP considers 1750 as start of industrial period? 'Pre-industrial' seems unsupportable and too vague. I know what authors intend here but many ESSD readers will not?

Response: We have revised the first two sentences in the abstract to avoid the picky argument regarding pre-industrial period and to be more precise regarding troposphere vs whole atmosphere: "Nitrous oxide ($N_2O$) is a long-lived potent greenhouse gas and stratospheric ozone-depleting substance, which continues to accumulate in the atmosphere. The mole fraction of tropospheric $N_2O$ has …" . For $N_2O$ concentration in 1750, we have added the following citation to the revised manuscript.
Reference :
Macfarling Meure, Cecelia, et al. "Law Dome $CO_2$, $CH_4$ and $N_2O$ ice core records extended to 2000 years BP." *Geophysical Research Letters* 33.14 (2006).

Line 92: if fluxes increased by 40% over 1980 to 2020 but concentrations increased only 25% (from 1750 to 2022, line 85) then something must also have changed in sinks? E.g. Fig 1 shows at least three sinks (downward arrows, including one massive downward arrow); atmospheric concentrations must represent some balance of these processes? Not clear here, nor elsewhere in thus manuscript. Authors job to compile and present best data (good on them) but also to explain basic balance / imbalance of global N2O budget? This reader might understand subtle differences here but many readers will not? Somewhere, in abstract or exec summary, readers need to find concise summary?

Response: $N_2O$ emission reported here represents the annual amount of $N_2O$ emitted to the atmosphere (concept of flux, unit: Tg N/yr). Atmospheric $N_2O$ concentration is directly proportional to atmospheric $N_2O$ burden (concept of pool, unit: Tg N). Since the initial atmospheric $N_2O$ burden (pool) is not zero, it doesn't change proportionally with $N_2O$ flux. We agree with the reviewer that the stratsopheric sink has increased as the concentration of N2O has increased, because it is concentration-dependent. Prather's papers have discussed this point and cited in this manuscript (Prather et al. 2023). That explains how fluxes can increase more than concentrations, because the concentration-dependent sink has also increased somewhat.

Lines 107 to 116: Good summary here! Compares recent work (1980 to 2020) to ice core records. No mention of 'industrial' or 'pre-industrial'. Revise abstract in light of what you have here? Also no mention of stratospheric processes (O3 impact) or sinks but these processes emerge later? Or, somewhere, a sentence that this budget focuses (necessarily) primarily on tropospheric processes?

Response: Thanks for the suggestion. We have revised the sentences as follows: "..land cover change. Ice core data show relatively constant tropospheric $N_2O$ mixing ratio over the past two millennia (Canadell et al., 2021; MacFarling Meure et al., 2006; Fischer et al., 2019), from about 270 ppb in 1750 to well above 300 ppb. The tropospheric N2O mole fractions,, .." Regarding the stratosphere, we would refer the reviewer to line 157 to 160.

Line 110: "It" You mean 'these'? Or, 'these concentrations'? Eliminate this sentence,

because you do not need two successive mentions of ice core records?

Response: We are sorry for the unclear statement. "It" refers to the tropospheric $N_2O$ mole fraction in 2022. We have revised this sentence:
"The tropospheric $N_2O$ mole fraction in 2022 is higher than at any time in the last 800,000 years."

We mentioned the ice core records two times because the first sentence focuses on the tropospheric $N_2O$ mole fraction, and the second focuses on the growth rate of atmospheric $N_2O$ mole fraction. We'd like to express that both the current $N_2O$ concentration level and its growth rate are unprecedented in the last 800,000 years.
Here, you use and cite units in fluxes of ppb per year. In abstract and next (emissions) paragraph you instead use Tg N per year. Settle on most useful or most appropriate set of units? Or, include a table that helps readers quickly convert?

Response: As stated in the manuscript, "ppb" is the most appropriate unit for atmospheric $N_2O$ mole fraction, and "ppb yr$^{-1}$" is the unit for annual change rate of atmospheric $N_2O$ mole fraction. "Tg Nyr$^{-1}$" is the most appropriate unit for $N_2O$ fluxes. We added one sentence stating the converting factor from "ppb yr$^{-1}$" to "Tg Nyr$^{-1}$" in the revised manuscript: "The conversion factor from the unit "ppb yr$^{-1}$" to the unit "Tg Nyr$^{-1}$" is 4.79 Tg N ppb$^{-1}$ (Prather, et al., 2012)."

Line 195: "most emitted"? Most often emitted? Most emitted by net (atmospheric) concentration or by flux? Most reactive? Need slight clarification here.

Response: Sorry for the unclear statements. We have revised the sentence as follows:
"which is the most important depleting substance of stratospheric ozone (World Meteorological Organization, 2022)"

Line 198: "mole fractions have increased by more than 25% since the pre-industrial era, from 270 parts per billion (ppb) in 1750 to 336 ppb in 2022" But, figures here only show data since 1980. If increase since pre-industrial values is true (as I accept), reader needs a citation or source for such data?

Response: Thanks for the suggestion. We have added the citation to the revised manuscript.
Reference :
Macfarling Meure, Cecelia, et al. "Law Dome $CO_2$, $CH_4$ and $N_2O$ ice core records extended to 2000 years BP." *Geophysical Research Letters* 33.14 (2006).

Line 200: detailed and well-referenced sentence starting 'The 20th century rate' renders the previous sentence moot; reader does not need to see both. ESSD/Copernicus impose some punctuation standard for '20th century'?

Response: We have revised the sentences as follows: "The increase rate of atmospheric $N_2O$ in the 20th century is unprecedented over the past 20,000 years".
We think it's better to keep both sentences, because the first sentence focuses on the atmospheric $N_2O$ mole fraction, and the second focuses on the growth rate of atmospheric $N_2O$ mole fraction. We'd like to express that both the current $N_2O$

concentration level and its growth rate are unprecedented in the last 800,000 years.

Line 204: "growth rate of atmospheric N2O, the mean annual growth" need changed punctuation here, e.g. growth rate of atmospheric N2O: the mean annual growth.

Response: Thanks for the suggestion. We have revised accordingly.

Line 217: "Reducing N2O emissions is a required net-zero greenhouse gas (GHG) emissions and the recovery of stratospheric ozone" Something missing here? Required to meet GHG targets, and to allow (or foster) recovery?

Response: We are sorry for the grammatical error. We have revised the sentences as follows:

 "Reducing $N_2O$ emissions will contribute to the mitigation of global warming and the recovery of stratospheric ozone (Jackson et al., 2019)."

Line 218: Complete "N2O mitigation measures"? I think Pier's paper pointed out that remaining CO2 targets/budgets for 2C disappeared into the 'noise' of other GHG mitigation efforts (e.g. for N2O) but did not address N2O mitigation impacts directly?

Response: We have improved and make more complete statement based on a recent paper reviewing the needs for non-$CO_2$ greenhouse gas emissions reductions compatible with a number of temperature stabilization targets based on IPCC AR6 data (https://dhttps://doi.org/10.1038/s43247-023-01168-8). The revised sentences are as follows:
"Significant reductions of $N_2O$ emissions are required along with net $CO_2$-emissions to stabilize the global climate system. For pathways consistent with the remaining carbon budget of 1.5°C, 1.7°C and 2°C stabilization, global $N_2O$ emissions need to be reduced by 22%, 18% and 11 %, respectively, by 2050 (Rogelj and Lamboll 2024)."

Line 221: "Implementing N2O mitigation" you already said this in previous sentence. This sentence seems redundant?

Response: We have revised the sentence as follows: "All in all, implementing $N_2O$ mitigation will contribute to achieving a set of United Nations Sustainable Development Goals (United Nations, 2016)."

Line 223: Nitrification and denitrification might both impact N2O production but, with one a source and one a sink, they can't both "contribute". Awkward phasing for most readers.

Response: $N_2O$ can be produced and emitted in both nitrification and denitrification processes (Butterbach-Bahl et al., 2013; Gruber & James, 2008; Kuypers et al., 2018; Frestons and Davidson 1989). We have added the references to the revised manuscript.

Line 225 and following: good list but punctuation should change to semi-colon between each of 21 factors? Proof readers will know.

Response: We have revised accordingly.

Line 225 and following: check Fig 3 to ensure tight correlation with list here, by exact terms and directions of arrows? I think I counted 21 fluxes in Fig 3 but with uncertainty about whether to count bidirectional arrows as one or two terms?

Response: We have revised Fig3 by adding a red arrow from the coastal waters box back to the atmosphere. Each bidirectional arrow corresponds to one term and indicates that this term can either positively or negatively affect terrestrial $N_2O$ emissions.

Line 262: But, no red arrow (indirect anthropogenic impact) from coast oceans box of Fig 3?

Response: This was an omission from our side and a red arrow from the coastal waters box back to the atmosphere has now been added to the revised figure. This change is consistent with the text that accounts for aquaculture (part of it being coastal) as a direct anthropogenic source and coastal emissions induced by N leaching as an indirect anthropogenic source.

Line 266: Good list but, strictly speaking, these should not fit in the category of terrestrial natural ecosystems?

Response: Sorry for the incorrect statement, these fluxes are in the category of "anthropogenic fluxes". We have changed "terrestrial natural ecosystems" to "terrestrial ecosystems".

Line 284: "multiple BU (BU) and TD (TD) methods" something missing or awkward here?

Response: Thanks for pointing this out. We have deleted "(BU)" and "(TD)".

Line 302: "all possible" but readers just learned that you had to ignore termite sources for lack of data? Perhaps all 'plausible'. Or, all 'quantifiable'? Change wording to reflect availability of reliable data? Lists and categories that follow seem reasonable and well-documented. Later (Line 1443) authors devote an entire paragraph to "missing fluxes". "All possible" remains confusing and/or inappropriate.

Response: Thanks for pointing this out. We have changed "all possible" into "all quantifiable".

Line 316: very important if slightly confusing paragraph. Put this in a table, instead? Ala Table 1 in GCB? Fluxes, change rates of fluxes, atmospheric concentrations: too much for reader to remember without a reference table?

Response: Thanks for the suggestion. We have added the unit table to the supplementary material.

Line 341: 'are' rather than "is"?

Response: We used "is" because "Which" refers to "the total $N_2O$ emission".

Line 364: to " to develop" and "quantified" in same sentence? Need some attention to tense here?

Response: Thanks for pointing this out. We have changed "quantified" into "to develop".

Please ensure to define bottom-up (BU) and top-down (TD) once and only once, then attend to all subsequent uses of abbreviations to ensure coherence. E.g. to this reader, text in legend to Figure 4 (line 379) seems confusing?

Response: Thanks for the suggestion. We have revised this sentence as follows:
 "We use both BU and TD approaches, including 20 BU and four TD estimates of $N_2O$ fluxes from land and oceans."

Here, confusion threatens overall merit of this work. By this point reader has confronted 21 types of N2O fluxes, recasting of those types into six broader categories, definition of units (fluxes, change rates of fluxes, concentrations), parsing across geographical regions (including to a few specific countries), and - finally (!?) elucidation by 31 (more if one counts same model run at two different spatial resolutions) inventories and global, regional or process models. Huge effort by authors to compile all this! Please keep readers well-informed and cognizant of which source (or sink) estimates apply to which categories. The category 'Shelf' for example, which authors intend as one depth-limited region of perimeter oceans, remains confusing as used. Authors will know best what they need to report and how but, unfortunately, present parsing and arrangement implies perfunctory approach while authors prefer to project entire effort as careful and complex. Some better way to convey complexity and uncertainty? Not clear for me. I plead for better overall arrangement or at least an ongoing outline to help readers? Table 1 represents a comprehensive list, without reliability designation? Figure 4 complicates when it might clarify?

Response: We appreciate this comment by the reviewer, which highlights the complexity and completeness of this assessment. The reviewer very well identifies the challenges we were confronted with when compiling the multiple sets of data. We have been largely able to sort out the multiple approaches. Without seriously misinterpreting the individual approaches, it proved impossible to fully harmonize the terms used differently, and to fully bring into agreement the respective system boundaries used. Although it is not clear how to reorganize such a volume of information, as the reviewer acknowledges, we have highlighted in the text Figures 1 and 2 (overall budget and infographic) and Table 3 (more detail and numbers for the same), which were developed to guide the reader thru the multiple fluxes and numbers. A good reference to return to when the reader needs to place the information in context. Likewise, Figures 13, 14 and 15 were developed to guide the reader through the regional information. We think this guidance will assist the reader in navigating the paper. We are convinced - as also indicated by the reviewer - that the present manuscript provides a useful compilation of the available information.

Line 415: Copernicus publisher adheres to standard mechanism to handle 'submitted' references. Not this one, unfortunately; authors and editors need to correct.

Response: We have updated this reference:

Li, Ya, et al. "Increased nitrous oxide emissions from global lakes and reservoirs since the pre-industrial era." *Nature Communications* 15.1 (2024): 942.

Line 417: Which "observation-based analysis"?

Response: We have revised the sentence to avoid confusion:
"The analysis of Rosentreter et al. (2023) is observation-based and includes the contribution of coastal vegetated ecosystems,"

Line 423: percent of what?

Response: 44% of the total $N_2O$ emissions from inland waters. We are sorry for the unclear statement. The results in Yao et al. (2020) suggested that 56% of the total $N_2O$ emissions from rivers, reservoirs, estuaries and lakes were attributed to anthropogenic N additions, and the resting 44% of the total $N_2O$ emissions were from natural sources. To avoid confusion, this sentence has been revised as:

"Using this approach, we estimated that $N_2O$ emissions from natural sources of rivers, reservoirs, lakes and estuaries accounted for 44% (36%−52%) of the total emissions from inland waters."

Line 424: "Shelf processes" in Table 1 actually represent "continental shelves" as specified here?

Response: Yes. We have changed "Shelf products" in Table 1 to "Continental shelf products" to avoid confusion.

Line 434: If authors already listed all pertinent (hi-res) ocean biogeochemistry models in Table1, does reader need a second list here? Again, this reader wishes for some clarification: unable to assign priority to any one model, authors have chosen to use them all, to use a mean, to use a median? How do uncertainties from individual models penetrate into overall global estimates?

Response: In this paper we look at an ensemble of ocean biogeochemistry models. Therefore we use the ensemble mean of models as the "best estimate" and use the range of ocean model results (min/max) as an estimate for the uncertainty. Using an ensemble can give a better indication of the uncertainty compared to the uncertainty from a single ocean biogeochemistry model. That's why we do not provide uncertainty bars for each inversion result. Here we listed the names of ocean models to follow the convention that had been used in the section above on the land models - which had listed model names and main references

Line 455: First reports on deriving uncertainty info from source materials? Reader of ESSD needs more of the same?

Response: Thanks for the suggestion. We have added more descriptions of uncertainties in FAOSTAT, EDGAR, and NMIP2 estimates to Section 4.2.1.

Line 465: More useful to list FAO general reports first, then to deal with FAOSTAT specifics on fire types after? E.g. helpful to readers to change order of last and next-tolast sentences?

Response: Thanks for your suggestion! We have changed the order of these two sentences according to your suggestion.

Line 470 and following: In this section authors need to include continental shelves in a more general 'coastal' category? Needs explicit mention/explanation?

Response: Here we include continental shelves in a more general "ocean" category which includes both open ocean and continental shelves. We have revised the sentence as follows:

"The emission from 'N deposition on ocean' was provided by Suntharalingam et al. (2012) which includes emission from both open oceans and continental shelves,"

Line 481: not clear where "56%" comes from?

Response: We are sorry for the unclear statement. The results in Yao et al. (2020) suggested that 56% of the total $N_2O$ emissions from rivers, reservoirs, estuaries and lakes was attributed to anthropogenic N additions. Empirical methods (empirical models and meta-analysis) adopted this ratio to calculate long-term average anthropogenic $N_2O$ emissions from inland waters, consistent with Tian et al. (2020). This sentence has been revised as follows:

"The anthropogenic emission from inland freshwaters estimated by Yao et al. (2020) considered annual N inputs and other environmental factors (i.e., climate, elevated $CO_2$, and land cover change). The results in Yao et al. (2020) suggested that 56% of the total $N_2O$ emissions from rivers, reservoirs, estuaries and lakes was attributed to anthropogenic N additions. Empirical methods (empirical models and meta-analysis) adopted this ratio to calculate long-term average anthropogenic $N_2O$ emissions from inland waters, consistent with Tian et al. (2020)."

Line 484: "low" or "act as a sink". Should reader assume you included these sources/sinks or ignored them as insignificant?

Response: We have deleted this sentence to avoid confusion.

Line 486: "SH1-SH7" and line 487 "SH1-SH8" From supplement reader learns that SH1 etc. represent set-up parameters for models involved in NMIP2 but readers need that informations sooner, e.g. here?

Response: We have moved the "Table A4. Simulation design of NMIP2." To "Table 2. Simulation design of NMIP2"

Line 491: "book-keeping" approach may account for deforestration/reforestration but you lost readers on broader issue of overall issue of land-cover changes on indirect (perturbation?) emissions?

Response: The reviewer is correct that bookkeeping methods could not account for indirect effects of environmental changes or emissions from some management

practices (e.g., fertilizer application) after land use change (e.g., climate effects). However, these effects are considered in other $N_2O$ emission sectors through the NMIP2 experiments. Such a use of the bookkeeping approach in our study exactly follows the methodology adopted in global carbon accounting (Friedlingstein et al. 2022).

Line 495 and following: Not clear what authors concluded here: should they use older estimate for NH3 oxidation and lightning production or do they ignore these processes as "small" and inconsistent or unquantifiable?

Response: We are sorry for the unclear statement and calculation error. For $NH_3$ source of $N_2O$: we used the mean value of two estimates : 0.4 Tg N $yr^{-1}$ (Kohlmann and Poppe, 1999) and 0.6 (0.3-1.1) Tg N $yr^{-1}$ (Dentener and Crutzen, 1994), the mean value is 0.5 Tg N $yr^{-1}$ and the range is 0.3-1.1 Tg N $yr^{-1}$. For $N_2O$ emission from lightning production, the estimate is 0.05 (0.02-0.09) Tg N $yr^{-1}$. Therefore, for $N_2O$ emissions from lightning and atmospheric production, the estimate is 0.55 (0.32-1.19) Tg N $yr^{-1}$. We also have revised the description and updated the number in the global $N_2O$ budget table (Table 3 in the revised manuscript).

Line 515: ppb to Tg conversion factor buried here, should appear more prominently in a 'units' table?

Response: We have added one 'unit' table and added the statement of conversion factor to the notes of Table 3.

Line 517 and following: Did authors use + 1.4% uncertainty or IPCC AR5 uncertainty? One applies to concentrations and other to concentration changes? Not clear to this reader.

Response: Sorry for the unclear statement. We have revised the sentences as follows:
 "Combining uncertainties in measuring the annual mean surface mole fraction, which are <1 ppb (Dlugokencky et al., 1994), with those of converting surface mole fractions to a global mean abundance, we estimate a ±1.4 % uncertainty in the absolute burden (Prather et al., 2012).  The uncertainty in the ppb-to-Tg conversion does not affect the trend uncertainty.  This uncertainty is estimated to be ±0.2 ppb or ±1 Tg N between any two years over any recent period, based on the combined NOAA and AGAGE record of surface $N_2O$ taken from Table 2.1 of the IPCC AR5 (Hartmann et al., 2013).  Thus, the uncertainty in the burden change between two decades (e.g., 2000s to 2010s) is bounded by ±1 Tg N (<0.1 %)."

Line 524: another example of a process (tropospheric loss) too small to appear in overall N2O budget?

Response: Yes, tropospheric chemical loss occurs at a very low rate, thus we did not include this item in the budget.

Line 533: 'is' rather than "as"?

Response: Should be "was" rather than "is". Sorry for the grammatical error, we have corrected it.

Line 548: additional uncertainties introduced by these interpolation or re-gridding steps?

Response: This interpolation doesn't affect the global statistics. Sorry for the grammatical error, we have corrected it.

Line 553: Back at line 319, we read "Unless specified, uncertainties are reported in brackets as minimum and maximum values of all estimates". Data presented here (e.g. 315.8 ("315.5-316.2) ppb in 2000 to 335.9 (335.6-336.1) ppb in 2022") follow this convention? Values in parentheses represent min and max? But, each source (e.g. NOAA, CSIRO) will have gone to lot of effort to identify uncertainties of their N2O measurements, reported not as + min/max. Min/max tells readers very little about distributions or uncertainties? Please can authors adopt, and adhere consistently to, better more reliable more informative uncertainties?

Response: Yes, uncertainties in the atmospheric $N_2O$ mole fractions also follow this convention, using minimum and maximum to represent uncertainty. We acknowledge that min/max can't tell much information about the distribution of uncertainty. However, considering that many of the identified $N_2O$ sources only have 2 or 3 products/estimates, we can't get much information about the distribution of uncertainty or just simply assume the normal distribution of uncertainties. Considering these facts, using minimum and maximum to represent uncertainty is the best choice for our study.

Line 556: "was" implies singular but this sentence refers two (plural) years, 2020 and 2021?

Response: Sorry for the grammatical error. We have corrected it.

Line 556: "30% higher than the average value in the decade of the 2010s" Not sure that readers can confirm this information from Figure 2? Inset of Figure 2, which purports to show annual growth rate, has no uncertainties, no demarcation of decades, nothing to help readers follow (or, dispute) authors' conclusions. Help, please.

Response: The inset of previous Figure2 shows the growth rate of $N_2O$ dry mole fraction at a monthly resolution. We have revised it into the annual resolution and added dash lines to indicate the levels of $N_2O$ growth rate in the 2000s and 2010s.

Line 563, Figure 5: No uncertainties in obs nor in model outcomes?

Response: We have added a table in supplementary material (Table SI-3) reporting the uncertainties in observed atmospheric $N_2O$ concentrations and the future predictions.

Line 571: "with large uncertainties (Figure 6)." What uncertainties? + min/max as above?
Something different here? No information! Nothing in panels or figure legend?

Response: Yes, we used min/max to represent uncertainty. We have deleted ", with large uncertainties" to avoid confusion. We have also added the following statements in the caption of Figure 6"

"For each sub-figure, the line represents the mean $N_2O$ emission of different estimates, and the shaded area shows minimum and maximum estimates."

Line 579, Figure 6: Except for panel C (Other direct anthropogenic emissions), all uncertainties (if shown by color ranges) appear to increase, 1980 to 2020? Not a positive report? Should readers assume that capability to construct N2O budget decreases, because uncertainties increase? Not what this reviewer would have expected as measurements and models all improve? If end of 2020 total of 6.7, with range of 3.3 (minimum or ?) to 10.9 (maximum or ?), how can reader trust anything that follows about sectors or regions? If authors expect readers to accept time-dependent changes in N2O sources or sinks, those readers will need to trust authors' handling of received as well as generated uncertainties. No evidence provided here.

Response: We acknowledge that the range of maximum-minimum estimates increased for most sectors (panel B, D, E), however, this is mainly caused by the increase in the overall magnitude of emissions. For the relative uncertainty ((maximum-minimum)/mean), direct emission from agriculture and indirect emission (panel B and D) only show slight increase, other direct emission (panel C) shows significant decreasing trend. The increase in the estimated ranges particularly increases reflects the increasing in the number of multiple sources of emission data. Our analysis has followed the principle that a larger member ensemble, in this case of estimates coming from diverse approaches, provides a more robust and higher confidence mean value, than using a smaller number of data sources, but with the downside of having larger ranges. This approach is consistent with the IPCC Guidance for Consistent Treatment of Uncertainties in its assessment reports, which we have followed in our analysis. We think this is best approach available to us at present given the diversity of data, and we acknowledge the need for improvements in future the way uncertainties are expressed in complex data syntheses.

Line 583 and following, sections on agriculture, other direct, etc.: Lots of work here, compiling and reporting data from trusted sources NIMP2, EDGAR, FAOSTAT, etc., but not one mention of uncertainties. Do all these sources produce 'perfect' data? I know, and readers will know better, that each source spends a great deal of time and effort to identify and report uncertainties. All of that effort and info lost here? Not one error bar or uncertainty envelop in any of these figures? Authors need to provide readers a basis to trust these conclusions, to respect authors' good efforts, but nothing presented here provokes nor supports such respect. Fossil fuel N2O emissions have remained unchanged for four decades? Weakness in reporting? True? How can any reader know? We need to trust authors to provide explanation with documentation and uncertainties! Instead, nothing provided!

Response: Uncertainties in estimates of FAOSTAT, EDGAR, and NMIP2 are as follows: (1) FAOSTAT: from Tubiello et al. (2013): Fig. 3 pg. 7, uncertainty in FAOSTAT $N_2O$ emissions is ~ 60% across typology. In fact, it is asymmetrical, following 2006 GL values and IPCC uncertainty formulae, with umin ~-30% and umax ~90%; (2) EDGAR: the global uncertainties in EDGAR for $N_2O$ are provided in Solazzo et al. (2021) (Table 6) for all sectors. The uncertainty in $N_2O$ emissions ranges for Energy between 113.3% and 113.3%, for IPPU between 15.7% and 12.4%, for agriculture between 301.7% and 224.9%, for Waste between 202.6% and 159.0%, and

for Other sectors between 111.8% and 111.8%. We would like to highlight the fact that N2O emissions from agriculture in EDGAR are very uncertain. (3) NMIP2: we also calculated the uncertainty in NMIP2 estimates of direct agricultural emissions, the maximum estimate is about 60% higher than the ensemble mean, and the minimum estimate is about 40% lower than the ensemble mean. We have added the uncertainties of FAOSTAT, EDGAR, and NMIP2 into Section 4.2.1 and 4.2.2.

In the manuscript, we reported that N2O emissions from fossil fuel and industry (including emissions from fossil fuel combustion and industry) only slightly increased during 1980-2020, mainly because that the increase in N2O emissions from fossil fuel combustion was largely offset by the decrease in N2O emissions from industry (see the following figure), including emissions from chemical processes, solvents and products use as described in EDGAR database (Solazzo et al. 2021). N2O emissions from fossil fuel combustion significantly increased during the past four decades.

[Figure]

Line 633, "both DLEM and book-keeping approach suggested increasing uncertainties in postdeforestation pulse effect". Really? Increasing uncertainties? Expressed as min/max, 95CI, per cent of total, + x sigma, what? If authors want readers to accept this contention about perturbation fluxes, those readers will want to have seen consistent approach to uncertainties up to this point and will need more details here.

Response: We apologize for the inaccurate description. The text has been revised as "The spread between different estimates (DLEM and the bookkeeping method) on post-deforestation pulse effect increased from the 1980s to the 2010s". Such a spread increase features the discrepancies of the two methods in accounting for the long-term changes in N2O fluxes in deforested areas.

Lines 648 and following: Reader would like to accept that natural N2O fluxes have not changed over four decades but a) this reader doubts that contention and b) authors have given no indication of their skill or knowledge to back up such contentions.

Response: Sorry for the unclear statement. Among all sources, natural emissions from shelves, inland waters, and lightning and atmospheric production were assumed to be constant during 1980-2020. According to BU approaches, the total natural emissions from these sources were 1.7 (0.9-3.0) Tg N yr$^{-1}$. Emissions from other natural sources including soils and open oceans kept relatively steady throughout the study period 1980-2020, with mean estimates fluctuating between 9.9-10.3 Tg N yr$^{-1}$ (minimum estimates: 6.2-7.1 Tg N yr$^{-1}$; maximum estimates: 12.8-13.6 Tg N yr$^{-1}$). We have revised Section 3.2.2 as follows:

"Emissions from natural soils and open oceans kept relatively steady throughout the study period 1980-2020, with mean estimates fluctuating between 9.9-10.3 Tg N yr$^{-1}$ (minimum estimates: 6.2-7.1 Tg N yr$^{-1}$; maximum estimates: 12.8-13.6 Tg N yr$^{-1}$). Natural emissions from all other sources including shelves, inland waters, and lightning and atmospheric production were assumed to be constant during 1980-2020. According to BU approaches, the total natural emissions from these sources were 1.7 (0.9-3.0) Tg N yr$^{-1}$. The mean value of global N$_2$O emissions from all the above-mentioned sources fluctuated between 11.5-11.9 TgN yr$^{-1}$, with an average of 11.7 TgN yr$^{-1}$. Global natural N$_2$O emissions have a large uncertainty, with the maximum estimates (15.8-16.6 TgN yr$^{-1}$) roughly double the minimum estimates (7.0-8.0 TgN yr$^{-1}$)."

Line 650 and following: do authors now (or, again) present minimum and maximum values? Doubtful, but no information given to help readers decide? Readers would like to accept authors' contentions on all these natural N2O fluxes but, without uncertainty information and specifications, how can we?

Response: Yes, as stated in section 2.1: "uncertainties are reported as minimum and maximum values of all estimates". In section 3.2.2, we have reported the minimum and maximum values of estimates for emissions from different natural sources.

Line 685, TD estimates: Inversions, as published, include substantial uncertainties. As presented (plotted) here, however, readers get no indication of authors expertise at assimilating and assessing TD info. No error bars on any plot, can't be true? What valid signals emerge from what noise? No sense of that provided in this section.

Response: In this paper we look at an ensemble of inversions and use the range of inversion results as an estimate for the uncertainty. Using an ensemble can give a better indication of the uncertainty (due to atmospheric transport and to the specific parameters used in the inversion, e.g. choice of observation and prior uncertainties) compared to the uncertainty from a single inversion framework. That's why we do not provide uncertainty bars for each inversion result. Also, not all these inversion frameworks provide uncertainties.

Line 778, Figure 13: Total N2O emissions (panel A), look nothing like panel A of Figure 6 (total anthropogenic). Total here represents Fig 6 plus non-varying natural total of 12 (Line 651)? 7 plus 12 gives 19, perhaps within uncertainty (?) of this Figure? If this reader has made too simple assumptions or additions, authors have not given sufficient information to prevent my errors? What do uncertainties shown here represent? Cumulative for BU and TD? From where? If we can't understand panel A, why should any reader give any credence to panels b to t? If I calculate instead from numbers given

in Table 2 (again, with min and max?), I get 18 BU or 17 TD total N2O sources, minus stratospheric sink of 13 or total atmospheric sink of 14 to get a residual in atmosphere of 3 to 5? Compared to 6 as provided authors? Sorry, does not compute correctly. In notes to Table 2 one reads that uncertainties follow AR5 with detailed notes provided in Supplement. Sorry again, this reader will not comb through supplement to elucidate uncertainty info. Many readers will simply give up at this point.

Response: We have double checked Figure 6 and Figure 13 (a), there is no calculation error. Figure 13 (a) looks different from Figure 6, mainly because it (total emission) includes natural emission (mean: 11.6 Tg N $yr^{-1}$, min-max: 7.2-15.9 Tg N $yr^{-1}$), which makes uncertainty range larger. For (a)-(t), "the blue lines represent the mean $N_2O$ emission from bottom-up methods and the shaded areas show minimum and maximum estimates; the red lines represent the mean $N_2O$ emission from top-down methods and the shaded areas show minimum and maximum estimates." We added the above description to the caption of Figure 13 to avoid confusion.

Lines 790 to 1084, regional extrapolations. More hard work by authors but basically bogus without uncertainties or some mechanism to validate?

Response: From Line 790 to 1084, we provide the uncertainties of most fluxes and their change rates (minimum and maximum). It's not objective to describe our regional analyses as "without uncertainties". Since it's not feasible to directly measure or observe $N_2O$ fluxes at regional level, we are unable to really validate the estimates of fluxes reported in our paper. Therefore, we compare the top-down estimates with bottom-up estimates to give readers the information of reliability of our reported numbers. This paper synthesizes state-of-the-art $N_2O$ flux products to estimate global and regional $N_2O$ budget, we don't think it's appropriate to say our results "bogus".

References:
Butterbach-Bahl, Klaus, et al. "Nitrous oxide emissions from soils: how well do we understand the processes and their controls?." *Philosophical Transactions of the Royal Society B: Biological Sciences* 368.1621 (2013): 20130122.

Firestone, M. K. and Davidson, E. A.: Microbiological basis of NO and N2O production and consumption in soil. In Exchange of trace gases between terrestrial ecosystems and the atmosphere [Andreae, M.O. and Schimel, D.S. (eds)] John Wiley & Sons, New York, pp. 7–21, 1989.

Dentener, Frank J., and Paul J. Crutzen. "A three-dimensional model of the global ammonia cycle." Journal of Atmospheric Chemistry 19 (1994): 331-369.

Fischer, Hubertus, et al. "N 2 O changes from the Last Glacial Maximum to the preindustrial–Part 1: Quantitative reconstruction of terrestrial and marine emissions using N 2 O stable isotopes in ice cores." *Biogeosciences* 16.20 (2019): 3997-4021.

Friedlingstein, P., M. O'Sullivan, M. W. Jones, R. M. Andrew, L. Gregor, J. Hauck, C. Le Quéré, I. T. Luijkx, A. Olsen, G. P. Peters, W. Peters, J. Pongratz, C. Schwingshackl, S. Sitch, J. G. Canadell, P. Ciais, R. B. Jackson, S. R. Alin, R. Alkama, A. Arneth, V. K. Arora, N. R. Bates, M. Becker, N. Bellouin, H. C. Bittig, L. Bopp, F. Chevallier, L. P. Chini, M. Cronin, W. Evans, S. Falk, R. A. Feely, T. Gasser, M. Gehlen, T. Gkritzalis,

L. Gloege, G. Grassi, N. Gruber, Ö. Gürses, I. Harris, M. Hefner, R. A. Houghton, G. C. Hurtt, Y. Iida, T. Ilyina, A. K. Jain, A. Jersild, K. Kadono, E. Kato, D. Kennedy, K. Klein Goldewijk, J. Knauer, J. I. Korsbakken, P. Landschützer, N. Lefèvre, K. Lindsay, J. Liu, Z. Liu, G. Marland, N. Mayot, M. J. McGrath, N. Metzl, N. M. Monacci, D. R. Munro, S. I. Nakaoka, Y. Niwa, K. O'Brien, T. Ono, P. I. Palmer, N. Pan, D. Pierrot, K. Pocock, B. Poulter, L. Resplandy, E. Robertson, C. Rödenbeck, C. Rodriguez, T. M. Rosan, J. Schwinger, R. Séférian, J. D. Shutler, I. Skjelvan, T. Steinhoff, Q. Sun, A. J. Sutton, C. Sweeney, S. Takao, T. Tanhua, P. P. Tans, X. Tian, H. Tian, B. Tilbrook, H. Tsujino, F. Tubiello, G. R. van der Werf, A. P. Walker, R. Wanninkhof, C. Whitehead, A. Willstrand Wranne, R. Wright, et al. (2022) Global Carbon Budget 2022. Earth Syst. Sci. Data, 14, 4811-4900.

Gruber, Nicolas, and James N. Galloway. "An Earth-system perspective of the global nitrogen cycle." *Nature* 451.7176 (2008): 293-296.

Kohlmann, J-P., and D. Poppe. "The tropospheric gas-phase degradation of NH3 and its impact on the formation of N2O and NOx." *Journal of atmospheric chemistry* 32 (1999): 397-415.

Kuypers, Marcel MM, Hannah K. Marchant, and Boran Kartal. "The microbial nitrogen-cycling network." Nature Reviews Microbiology 16.5 (2018): 263-276.

Li, Ya, et al. "Increased nitrous oxide emissions from global lakes and reservoirs since the pre-industrial era." *Nature Communications* 15.1 (2024): 942.

Macfarling Meure, Cecelia, et al. "Law Dome $CO_2$, $CH_4$ and $N_2O$ ice core records extended to 2000 years BP." Geophysical Research Letters 33.14 (2006).

Rogelj, Joeri, and Robin D. Lamboll. "Substantial reductions in non-CO2 greenhouse gas emissions reductions implied by IPCC estimates of the remaining carbon budget." *Communications Earth & Environment* 5.1 (2024): 35.

Solazzo, Efisio, et al. "Uncertainties in the Emissions Database for Global Atmospheric Research (EDGAR) emission inventory of greenhouse gases." *Atmospheric Chemistry and Physics* 21.7 (2021): 5655-5683.

Tubiello, Francesco N., et al. "The FAOSTAT database of greenhouse gas emissions from agriculture." Environmental Research Letters 8.1 (2013): 015009.

---

## Referee Report (RR1)

Direct N2O "precisely measured at a global network of stations". Uncertainties known and specified by those sources (e.g CSIRO, NOAA); they spend lots of effort to ensure precision. Not reported here! Would affect e.g. Figure 5!

This reader accepts authors' claims of massive unique effort to gather and report N2O information, to "explore the relative temporal and spatial importance of multiple sources and sinks". Not their job, they correctly claim, to construct an encompassing end-to-end uncertainty budget. But, if this group of experts can not or will not undertake such an effort, who will? My review concedes the effort required but remains dismayed by large unspecified unquantified uncertainties. My challenge / hypothesis stands: "uncertainties remain so large so as to preclude conclusions on specific sources or assigned to specific regions". Again, who, if not this group, will help us sort such issues?

In opening paragraphs, for example, authors switch between ppm without plus/minus to decadal emissions rates (likewise without min/max designation) to BU source information with min/max. Set aside weaknesses of min/max; why do readers encounter them in some places but miss them completely in others?

Format and labels of figures remain incomplete. Abbreviations applied in Fig 3, for example, remain inconsistent with text and incomplete.

Line 358 delineates data into "three" periods: "1997-2020, 1980-2020 and 2010-2019." But, at line 320, reader learned that "budgets cover the decades of 1980-89, 1990-99, 2000-09, 2010-2019,". What?

Line 362: authors claim to "show"; this reader accepts only that they have made best estimates. Term 'estimate' frequently used in following paragraphs. A bit of modesty in describing the outcome could have perverse effect of highlighting massive efforts!

Line 1323: something wrong here as EDGAR min and max for N emissions due to energy both equate to 113.3%? Or, next line, to 111.8% and 111.8%. Not my job to chase down this reference! Authors need to ensure correct extraction and citation!!! 300% errors? Sentence following, assigning EDGAR as "very uncertain" seems redundant as well as erroneous? EDGAR folks would disagree? All due to erroneous EFs? No means to estimate reliability of national or regional or global EFs? Fig 18 shows EFs ranging from <1 to >3 for productive agricultural regions across the globe. This reader maintains that Fig 18 invalidates prior effort(s) at regional assignments. Likewise for Figs 19 & 21?

Supplement includes only min-max? (If additional info on statistical uncertainties remains buried in individual descriptions of biogeography models or

atmospheric inversions this reviewer did not find them.) I want to credit authors for adding uncertainty info, but because I did not look at prior Supplement I can not tell.

I admire massive effort to assemble N2O info from so many disparate sources. If efforts to delineate composite uncertainties have not kept up, that remains mostly a challenge to future research.

---

## Author Response (AR2)

**point-by-point response to the reviews.**

**Response to the Reviewer #1**

The authors have responded carefully and thoroughly to the original reviews. I have just a few more comments, focused on the beginning sections of the manuscript.

Response: We would like to thank the reviewer for the careful review and positive comments. The manuscript has been revised accordingly, and our point-by-point responses in blue color are provided below, and our new/modified texts in the revised manuscript are indicated in red color.

Line 99. Should "biochemical" be "biogeochemical"?

Response: Thanks for the suggestion, we have revised accordingly.

Lines 112-114. "The observed atmospheric N2O concentrations in recent years have exceeded projected levels under all scenarios in the Coupled Model Intercomparison Project Phase 6 (CMIP6), underscoring the urgency to reduce anthropogenic N2O emissions." The word "urgency" seems overstated, given that N2O contributes a relatively small 6.4% of anthropogenic radiative forcing, as stated above on line 96. I recommend changing "urgency to reduce" to something like "importance of reducing." There could be many reasons why CMIP6 underestimated the rate of growth. It is therefore perhaps not appropriate to create alarm on that basis, especially without context or explanation of how far in excess the actual concentration is compared to the CMIP6 projected concentration.

Response: Thanks for the suggestion, we have revised accordingly.

Line 121-122. There are some missing words in this sentence, which I have inserted as capitals, "Ice core data show A relatively constant tropospheric N2O mixing ratio over the past two millennia (Canadell et al., 2021; MacFarling Meure et al., 2006; Fischer et al., 2019), FOLLOWED BY AN INCREASE from about 270 ppb in 1750 to well above 300 ppb."

Response: Thanks for the suggestion, we have revised accordingly.

Line 133. "BU approaches estimated…" would be better than "BU approaches showed…"

Response: Thanks for the suggestion, we have revised accordingly.

Lines 237-238. "Reducing N2O emissions will contribute to the mitigation of global warming and the recovery of stratospheric ozone (Jackson et al., 2019)." This would be a good place to include the important note that stratospheric NOx from N2O has offset (rather than exacerbated) halogen-catalyzed stratospheric ozone loss through various buffering reactions, e.g., the formation of halogen reservoir species like ClONO2. NOx also buffers HOx-catalyzed ozone loss. As a result, increased stratospheric NOx due to rising levels of N2O can lead to incremental

stratospheric O3 loss but is unlikely to cause catastrophic ozone loss the way that anthropogenic halogens did (e.g., Wennberg et al., 1994; Nevison et al., 1999; Ravishankara et al., 2009).

Response: We have added one sentence according to your suggestion, line 239-243: It is noted that although increased stratospheric $NO_x$ due to rising levels of $N_2O$ can lead to incremental stratospheric $O_3$ loss but is unlikely to cause catastrophic ozone loss the way that anthropogenic halogens did, because stratospheric $NO_x$ from $N_2O$ has offset halogen-catalyzed stratospheric ozone loss through various buffering reactions, e.g., the formation of halogen reservoir species like $ClONO_2$ (Wennberg et al., 1994; Nevison et al., 1999; Ravishankara et al., 2009).

Lines 238-241 "Significant reductions of N2O emissions are required along with net CO2-emissions to stabilize the global climate."

This sentence is much improved over the original. However, the subsequent sentence seems to make the debatable assumption that all GHGs should be cut proportionally. This is not necessarily true if some GHGs (e.g., methane) are easier and less societally disruptive to cut than others or if they are shorter-lived, such that their reduction will more effectively reduce radiative forcing. Since anthropogenic N2O emissions are fundamentally related to food production and the livelihood of farmers, I would encourage a carefully worded statement here. One suggestion would be to add a qualifying clause such as, "For pathways consistent with the remaining carbon budget of 1.5°C, 1.7°C and 2°C stabilization, and assuming that all GHGs should be cut in equal proportion to their contribution to anthropogenic radiative forcing, global N2O emissions need to be reduced by 22%, 18% and 11 %, respectively*, by 2050 (Rogelj and Lamboll, 2024). *Note that respectively was misspelled.

Response: Thank you! We have added "and assuming that all GHGs should be cut in equal proportion to their contribution to anthropogenic radiative forcing," to line 245-246 according to your suggestion.

Line 361. The 3 different time frames, which were quite confusing in the original manuscript, are better explained with the addition of this paragraph. However, the choice of 2010-2019 is still confusing since the full period of the study is 1980-2020. It would seem more logical to use 2011-2020 as the last 10 years. I would suggest switching from 2010-2019 to 2011-2020 if this is not too difficult to do. However, this change is not absolutely necessary if it is too burdensome to implement.

Response: Thanks for the suggestion! We chose 2010-2019 because our study is designed to report $N_2O$ budget for the four decades: 1980s, 1990s, 2000s, and 2010s, and we use the year 2020 to represent the most recent status. Therefore, we prefer keeping the choice of 2010-2019, rather than changing to 2011-2020.

References

Wennberg, P,.O., et al., 1994. Removal of stratospheric O3 by radicals: In situ measurements of OH, HO2, NO, NO2, ClO and BrO, Science, 266, 398-404.

Nevison, C.D, S. Solomon and R.S. Gao, 1999. Buffering interactions in the modeled response of stratospheric O3 to increased NOx and HOx, J. Geophys. Res., 104 (D3), 3741-3754.

Ravishankara, A., Daniel, J. & Portmann, R. 2009. The dominant ozone-depleting substance emitted in the 21st century. Science 326, 123-125.

**Response to the Reviewer #2**

We would like to thank the reviewer for the careful review and insightful comments. The manuscript has been revised accordingly, and our point-by-point responses in blue color are provided below, and our new/modified texts in the revised manuscript are indicated in red color.

Direct N2O "precisely measured at a global network of stations". Uncertainties known and specified by those sources (e.g CSIRO, NOAA); they spend lots of effort to ensure precision. Not reported here! Would affect e.g. Figure 5!

Response: We acknowledge that there are uncertainties in measurements from the observation networks (AGAGE, NOAA and CSIRO), and we have reported the uncertainties in Figure 2. We didn't plot all the three measurements in Figure 5 because the differences in the three lines are hard to distinguish and the uncertainty information has already been reported in Figure 2.

This reader accepts authors' claims of massive unique effort to gather and report N2O information, to "explore the relative temporal and spatial importance of multiple sources and sinks". Not their job, they correctly claim, to construct an encompassing end-to-end uncertainty budget. But, if this group of experts can not or will not undertake such an effort, who will? My review concedes the effort required but remains dismayed by large unspecified unquantified uncertainties. My challenge / hypothesis stands: "uncertainties remain so large so as to preclude conclusions on specific sources or assigned to specific regions". Again, who, if not this group, will help us sort such issues?

Response: Thanks for your comments! We acknowledge that our current $N_2O$ budget still has large uncertainties although we have used a large range of available sources to provide an overall assessment. However, our study identifies where the uncertainties are well constrained or very large, which can be viewed as a success of our effort. We also acknowledge that constructing an encompassing end-to-end uncertainty budget will guide future research. We will make more efforts to attempt an improved quantitative analysis of uncertainties in the next round of the $N_2O$ budget.

In opening paragraphs, for example, authors switch between ppm without plus/minus to decadal emissions rates (likewise without min/max designation) to BU source information with min/max. Set aside weaknesses of min/max; why do readers encounter them in some places but miss them completely in others?

Response: We did not use the unit "ppm" in our study. We used plus/minus to report the magnitude of uncertainties in atmospheric burden in Section 2.5.1. Here, our aim was to report uncertainty itself, so we didn't use the mean (minimum-maximum) as in other places. We think it is clear for readers, and will not cause misunderstanding.

Format and labels of figures remain incomplete. Abbreviations applied in Fig 3, for example, remain inconsistent with text and incomplete.

Response: Sorry for the errors. We have added abbreviations of the identified $N_2O$ sources and sinks shown in the main text and corrected the inconsistent descriptions. We also revised Figure 3.

Line 358 delineates data into "three" periods: "1997-2020, 1980-2020 and 2010-2019." But, at line 320, reader learned that "budgets cover the decades of 1980-89, 1990-99, 2000-09, 2010-2019,". What?

Response: Our study period is 1980-2020, the statement "budgets cover the decades of 1980-89, 1990-99, 2000-09, 2010-2019 .." is not contradictory with the statement that "We focus on $N_2O$ fluxes and their change rates during three periods: 1997-2020, 1980-2020 and 2010-2019".

Line 362: authors claim to "show"; this reader accepts only that they have made best estimates. Term 'estimate' frequently used in following paragraphs. A bit of modesty in describing the outcome could have perverse effect of highlighting massive efforts!

Response: Thanks for the suggestion! We have revised the sentence as follows: "we report the magnitudes of emissions from different sources to give best estimates of their latest status and relative importance."

Line 1323: something wrong here as EDGAR min and max for N emissions due to energy both equate to 113.3%? Or, next line, to 111.8% and 111.8%. Not my job to chase down this reference! Authors need to ensure correct extraction and citation!!! 300% errors? Sentence following, assigning EDGAR as "very uncertain" seems redundant as well as erroneous? EDGAR folks would disagree? All due to erroneous EFs? No means to estimate reliability of national or regional or global EFs? Fig 18 shows EFs ranging from <1 to >3 for productive agricultural regions across the globe. This reader maintains that Fig 18 invalidates prior effort(s) at regional assignments. Likewise for Figs 19 & 21?

Response: The emission factor shown in Figure 18 is derived from NMIP2 models whose estimates were integrated in this study. This figure reflects the phenomenon that emission factor is spatially heterogeneous, which is reported in previous studies (Cui et al., 2021; Harris et al., 2022). There is no direct linkage between spatial heterogeneity of emission factor and the reliability of estimates of emissions, therefore, we don't agree with the opinion that Fig 18 invalidates our prior effort(s) at regional assignments. Figures 19 and 21 show the uncertainties in estimates of $N_2O$ emissions from soils and ocean, identifying where the uncertainties are well constrained or very large. It can be viewed as a success of our effort, which will inform future research.

This manuscript doesn't implement additional uncertainty analysis for EDGAR database. These uncertainty estimates are directly taken from Solazzo et al. (2021). We modified the statements as follows: "The uncertainties for EDGAR $N_2O$ emissions estimated by *Solazzo et al.* (2021) are based primarily on the uncertainties in emissions factors and activity data statistics from the *IPCC* (2006). Globally, these emissions are accurate within an interval of $\pm113$ for energy, -12% to +16% for industrial processes and product use, -225 to +302 for agriculture, -159% to 203% for waste and $\pm112\%$ for others; the most uncertain emissions are those related to $N_2O$ from waste and agriculture."

Supplement includes only min-max? (If additional info on statistical uncertainties remains buried in individual descriptions of biogeography models or atmospheric inversions this reviewer did not find them.) I want to credit authors for adding uncertainty info, but because I did not look at prior Supplement I can not tell.

Response: In supplementary material, we listed atmospheric $N_2O$ dry mole fraction measured by the three observing networks, and the corresponding minimum and maximum values, as well as minimum and maximum estimates of future projections of atmospheric $N_2O$ dry mole fraction. We reported the minimum and maximum values as an indication of the uncertainty, consistent with the main text.

I admire massive effort to assemble N2O info from so many disparate sources. If efforts to delineate composite uncertainties have not kept up, that remains mostly a challenge to future research.

Response: Thanks for your insightful comments! We acknowledge that delineating composite uncertainties in $N_2O$ budget will guide future research. We will make more efforts on an improved quantitative analysis of uncertainties in the next round of the $N_2O$ budget.

**Reference:**

Cui X, Zhou F, Ciais P, et al. Global mapping of crop-specific emission factors highlights hotspots of nitrous oxide mitigation[J]. Nature Food, 2021, 2(11): 886-893.

Harris E, Yu L, Wang Y P, et al. Warming and redistribution of nitrogen inputs drive an increase in terrestrial nitrous oxide emission factor[J]. Nature Communications, 2022, 13(1): 4310.